# Timely TGFβ signalling inhibition induces notochord

Tiago Rito[1✉], Ashley R. G. Libby[1], Madeleine Demuth[1], Marie-Charlotte Domart[1], Jake Cornwall-Scoones[1] & James Briscoe[1✉]

The formation of the vertebrate body involves the coordinated production of trunk tissues from progenitors located in the posterior of the embryo. Although in vitro models using pluripotent stem cells replicate aspects of this process[1–10], they lack crucial components, most notably the notochord—a defining feature of chordates that patterns surrounding tissues[11]. Consequently, cell types dependent on notochord signals are absent from current models of human trunk formation. Here we performed single-cell transcriptomic analysis of chick embryos to map molecularly distinct progenitor populations and their spatial organization. Guided by this map, we investigated how differentiating human pluripotent stem cells develop a stereotypical spatial organization of trunk cell types. We found that YAP inactivation in conjunction with FGF-mediated MAPK signalling facilitated WNT pathway activation and induced expression of TBXT (also known as BRA). In addition, timely inhibition of WNT-induced NODAL and BMP signalling regulated the proportions of different tissue types, including notochordal cells. This enabled us to create a three-dimensional model of human trunk development that undergoes morphogenetic movements, producing elongated structures with a notochord and ventral neural and mesodermal tissues. Our findings provide insights into the mechanisms underlying vertebrate notochord formation and establish a more comprehensive in vitro model of human trunk development. This paves the way for future studies of tissue patterning in a physiologically relevant environment.

The formation of the vertebrate body axis is an evolutionarily conserved process that requires coordinated generation of multiple cell types from a population of progenitors in the caudal embryo[12]. The node, a midline structure, has a central role in this process[13]. It secretes factors to organize forming trunk tissues[14] and later gives rise to the notochord[15,16], the mesodermal rod that provides mechanical and signalling cues to the embryo, and the floor plate, the ventral midline domain of the neural tube that patterns neural tissue. Various signalling pathways, including WNT, BMP, NODAL and FGF[17], have been implicated in this process, but how these produce and organize the cell types necessary to form the body is not clear.

In vitro models of stem cell differentiation aimed to mimic signalling conditions around the node have been developed including gastruloids[1,3–5], axially elongating organoids[9,10], trunk-like structures[2], spinal cord organoids[6], somitoids[7], and axioloids[8]. Although these models generate varying amounts of neural, endodermal and mesodermal tissue, they often lack key components, most notably the notochord and its dependent tissues such as the floor plate. This raises the possibility that notochord specification has specific signalling requirements not met in current models.

## Single-cell analysis of developing trunk

To investigate the emergence of trunk tissue, including neural tube, somites and notochord, we analysed the transcriptome of single cells from caudal regions of chick embryos, approximately 5–6 h apart, with somite (S) numbers 4S, 7S, 10S and 13S (Hamburger–Hamilton stages 8–11 (HH8–HH11); Fig. 1a). These stages encompassed the induction of HOXB9 and HOXC9 expression (Extended Data Fig. 1a) and complement existing data from earlier developmental stages (HH4–HH7)[18,19], and from tailbud and anterior portions of the embryo[20,21]. Cell types were defined by Louvain clustering and marker gene expression (Fig. 1b and Supplementary Fig. 1a). The proportions of cell types were stable across stages (Supplementary Fig. 1b–d).

Analysis of gene expression identified cells of the expected tissues. Expression of MESP1, SNAI2 and MSGN1 was used to demarcate the primitive streak and early nascent mesoderm. TBX6 defined paraxial and presomitic mesoderm, the latter with additional MEOX1 co-expression (Fig. 1c left and Supplementary Fig. 1e). Co-expression of MESP1 and RIPPLY2 identified cells at the so-called wavefront, differentiating into the first somite (Fig. 1c left (MESP1+MEOX1+ in purple) and Supplementary Fig. 1a (cluster 23)). Closely associated with the transition to paraxial mesoderm (PXM) was ADAMTS18 (Supplementary Fig. 1e). This secreted metalloprotease cleaves fibronectin[22] and promotes cell locomotion and an epithelial-to-mesenchymal transition.

Lateral plate mesoderm was identifiable by HAND1, HAND2, GATA2, GATA4, CFC1, BMP4 and EVX1 expression, and it included an OSR1+ population, TBX20+TNNT2+ cardiac mesoderm and PAX2+LHX1+MNX1+ intermediate mesoderm (Fig. 1c right and Supplementary Fig. 1f).

[1]The Francis Crick Institute, London, UK. ✉e-mail: tiago.rito@crick.ac.uk; james.briscoe@crick.ac.uk

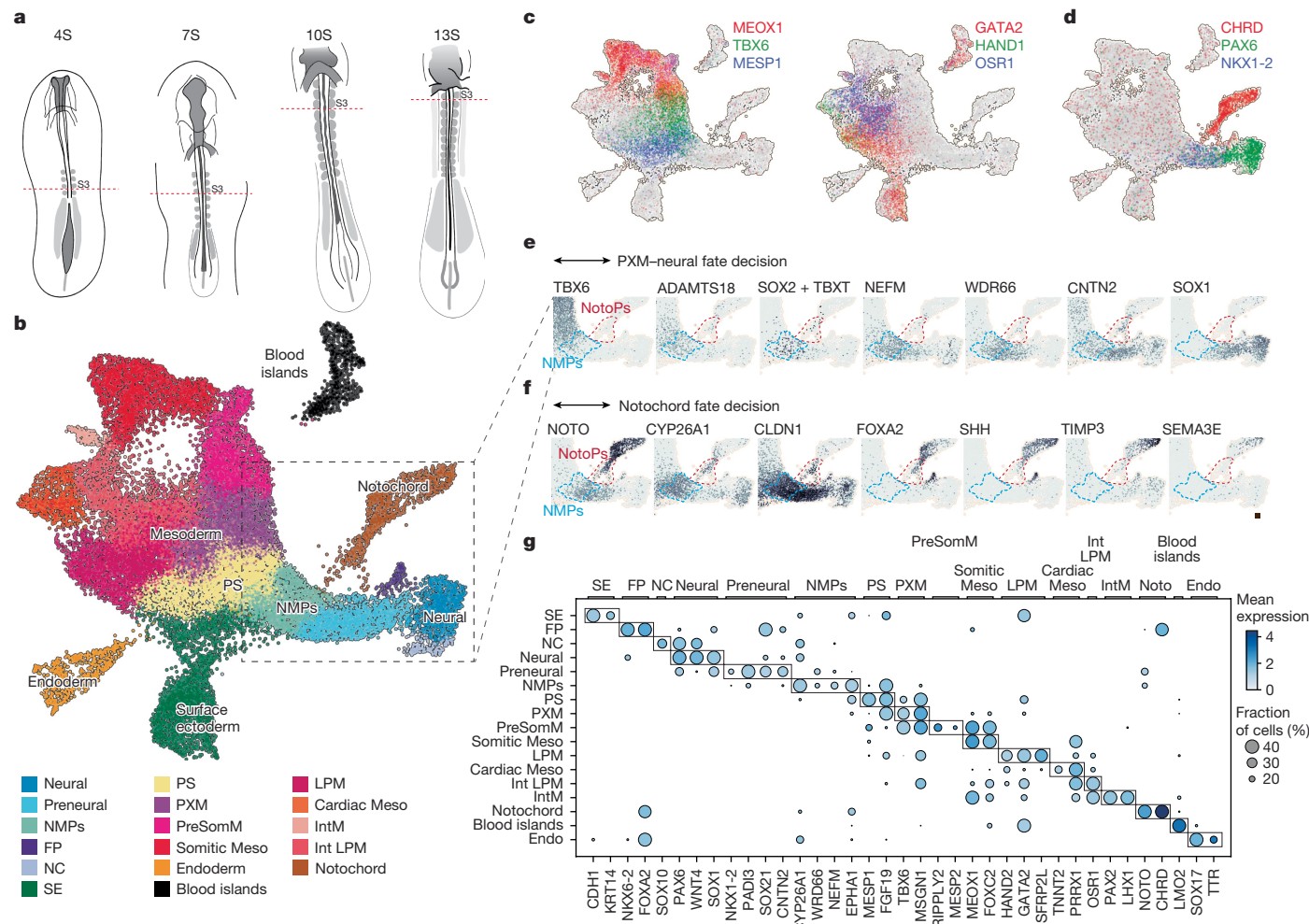

**Fig. 1 | Single-cell transcriptomics of chick trunk development. a**, Summary of chicken embryo dissections to generate scRNA-seq datasets. Tissue caudal to the third somite pair in embryos with 4S, 7S, 10S and 13S (HH8–HH11) was dissociated and sequenced. **b**, 2D embedding (uniform manifold approximation and projection) of single-cell data highlighting the different cell populations present during early vertebrate trunk formation. The plot shows 27,379 single cells from all developmental stages. **c,d**, Embedding of chick trunk (4S–13S) coloured by expression of different marker genes of the paraxial and lateral plate mesodermal (LPM) compartment (**c**) and of notochord (Noto), neural, endoderm (Endo) and surface ectoderm (SE) (**d**). **e,f**, Embedding showing the expression of several genes in the PXM, NMPs and neural cell populations (**e**) and the NMPs and notochord (**f**) cell populations. Scale is linear and shows a maximum count of 4. **g**, Overview of gene expression levels and percentage of cells positive for marker gene expression used to distinguish cell types present in the trunk. FP, floorplate; IntM, intermediate mesoderm; NC, neural crest; PreSomM, presomitic mesoderm.

Notochord cells expressed TBXT, NOTO, SHH and CHRD (Fig. 1d and Supplementary Fig. 1g). SHH also marked the floor plate and SOX17+ endodermal cells, and the latter included a MNX1+HOXB1+ subcluster of the pancreatic lineage (Supplementary Fig. 1f,g).

A CDH1+ surface ectoderm population was composed of more mature surface ectoderm expressing KRT14, KRT17, TFAP2A, DLX5, EPHA1 and WNT6 and a group of cells co-expressing MESP1 and CDH1 (ref. 23, Extended Data Fig. 1b and Supplementary Fig. 2a). PAX6-expressing neural cells (Fig. 1d) included ventral (NKX6-1+NKX6-2+) and dorsal (PAX3+MSX1+) progenitors and SOX10-expressing neural crest cells (Supplementary Fig. 2b). In addition, a population of preneural cells expressing NKX1-2 (also known as SAX1) was evident[24,25]. The data revealed previously uncharacterized markers of this population, such as PADI3 and CA2, and CNTN2, a cell adhesion and recognition molecule from the immunoglobulin superfamily. Comparison with a single-cell study of early anterior neural induction in the chick[21] identified genes in common, including *ZIC2* (early preneural) and *MAFA* (late preneural and neural) as well as others such as *GLI2*, *RFX3*, *ZNF423* and *TAF1A* that were more broadly expressed at these stages (Supplementary Fig. 2c).

We identified transcriptomic signatures of progenitor populations involved in generating trunk tissue. Two populations stood out: neuromesodermal progenitors (NMPs) and putative notochord progenitors. NMPs comprised SOX2+TBXT+ cells, typically used to identify these progenitors[17,26] (Fig. 1e and Extended Data Fig. 1c), as well as other proteins previously associated with NMPs, such as NKX1-2 (ref. 27), EPHA1 (ref. 28), CDX2 and CYP26A1 (ref. 29). Low levels of NOTO were detected in this population in agreement with previous observations[30]. Expression of CLDN1 and F2RL1 (also known as PAR2; Extended Data Fig. 1c) was also present in these cells, emphasizing the epithelial nature of NMPs[31]. Additionally, NEFM, an intermediate filament, and WDR66 (ENSGALG00000004365 in chick), a cilia and flagella-associated protein, were expressed in NMPs.

By contrast, NOTO+CHRD+ notochord cells contained at least two distinct populations, one of which expressed FOXA2, CDX2, CYP26A1, CLDN1, CNTN2 and GNOT2 that we considered to correspond to the node (that is, the cell population around the median pit; Fig. 1f, NotoPs). The second seemed to be more mature notochord and expressed SHH, NOG, LEFTY and also TIMP3, a metallopeptidase inhibitor, and SEMA3C

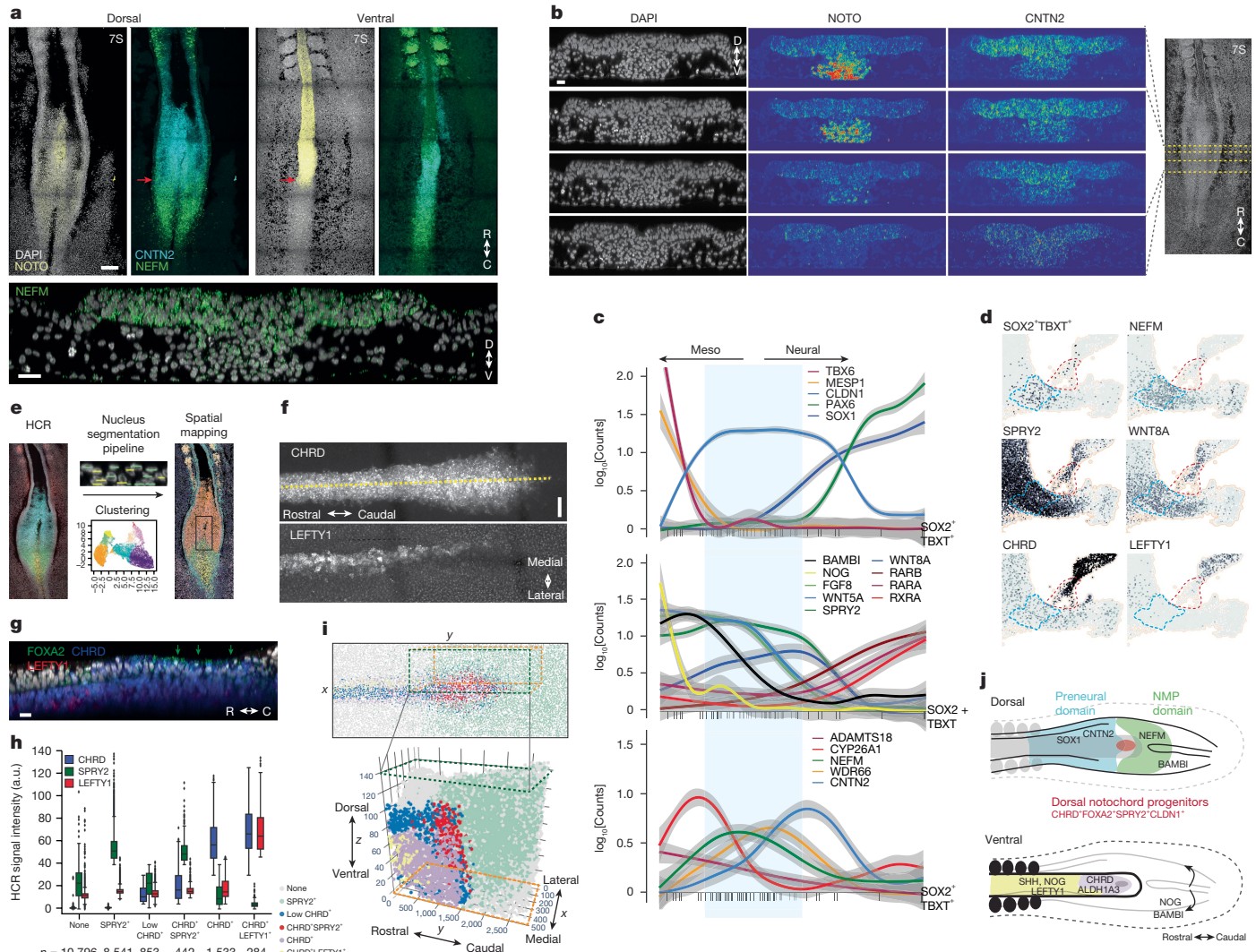

**Fig. 2 | Axial progenitors delimited by cytoskeletal components and TGFβ family inhibitors. a**, Multiplexed RNA fluorescence HCR image of a 7S chick embryo stained for NOTO in yellow (red arrows mark caudal limit), NEFM in green and CNTN2 in cyan. R, rostral; C, caudal; D, dorsal; V, ventral. **b**, Optical section of an HCR image showing dorso-ventral segregation of NOTO and CNTN2 caudal of the median pit. **c**, Transcriptional similarity ordering (pseudotime analysis) of chick single cells from PXM to NMPs and neural clusters showing the expression of CNTN2 and NEFM relative to that of other lineage and signalling genes. Data are shown as 95% confidence interval around a smooth spline. Vertical black lines at the bottom of the plots show cells co-expressing SOX2 and TBXT. **d**, Embedding detail with gene expression of TGFβ family inhibitors, SPRY2 and WNT8A. NotoPs, red outlined region; NMPs, blue outlined region. **e**, Single-cell quantification of HCR signals using the Nucleus pipeline. Outlined area indicates node region. **f**, Maximum-intensity projection (top view) of a 7S chicken embryo around the node. CHRD and LEFTY1 HCR

signal intensity is shown. Dotted line indicates the optical section shown in **g**. **g**, Axial optical section showing dorsal NotoPs expressing FOXA2 (green arrows mark dorsal expression) and CHRD. **h,i**, Spatial mapping based on HCR expression of CHRD, LEFTY1 and SPRY2. Box plot (**h**) shows the HCR signals for each cluster; the middle line is the median across cells, the lower and upper hinges correspond to the first and third quartiles, and the whiskers extend from the hinge to the largest and smallest value no further than 1.5× the interquartile range from the closest hinge; n represents the number of cells in each cluster. Scatter plot (**i**) with 3D centroids for each segmented cell and their cluster assignment according to **h**. **j**, Schematic of the dorsal and ventral view of an elongating 7S embryo. Colour highlights the HCR domains identified with cytoskeletal and adhesion components, and the expression of key BMP and NODAL inhibitors. Images in **a**,**b**,**f**,**g** are representative of at least two independent experiments with a minimum of three embryos each. Scale bars, 100 μm (**a**(top)), 10 μm (**a**(bottom),**b**,**g**) and 30 μm (**f**).

and SEMA3E, semaphorins implicated in the control of cell morphology and motility, respectively[32].

Near-identical cell populations were present in the mouse (embryonic day 8.0–8.5)[33] and macaque (Carnegie stage 9–11)[34], highlighting the conservation of trunk formation across vertebrates (Supplementary Fig. 3). Pairs of cells from NMP and notochord clusters exhibit some of the shortest mean Jaccard distances between their transcriptomes, highlighting their similarity in gene expression (Extended Data Fig. 2). Mouse NMP cells, as in chick, exhibited the highest proportion of double-positive SOX2⁺TXBT⁺ cells and expressed CYP26A1

and NEFM. Together, our analyses detected both established and new gene markers for each population (Fig. 1g and Supplementary Table 1).

## Mapping axial progenitor populations

To locate the position of progenitor populations, we mapped them back to chick embryos. NEFM, expressed in NMPs, marked the dorsal part of the node–streak border where the SOX2⁺TBXT⁺ NMPs reside, extending from the anterior primitive streak to the lateral edge of

the preneural domain around the caudal node (Fig. 2a). The equivalent domain in the mouse has a smaller inverted-U shape surrounding the primitive streak reflecting the distinct geometry of rodent embryos[26]. NEFM was also found further away from this region in the caudal portion of somites and at the folding edges of the neural tube (Supplementary Fig. 4a). CNTN2, which delineates preneural cells, marked a complementary rostral domain. Similar to NEFM, CNTN2 was mostly dorsal, but we noted expression in the cells near the node and the caudal onset of NOTO. Rostrally, NOTO and CNTN2 form two mutually exclusive dorsal–ventral domains before any morphological segregation between notochord and neural plate was apparent (Fig. 2b). CNTN2 expression was spatially distinct from the more rostral expression of the neural marker SOX1 (Supplementary Fig. 4b). CYP26A1 and ADAMTS18 were expressed more caudally in a narrow ellipsoid domain abutting CNTN2. This domain partially overlapped NEFM but extended ventrally from the primitive streak to just above the pit (Supplementary Fig. 4c).

Complex signalling networks regulate the behaviour of cells in the node–streak border[17,26]. Constructing a gene expression trajectory between SOX1[+] neural progenitors and MESP1[+]TBX6[+] PXM identified a cell population of CLDN1[+]NEFM[+] NMPs (Fig. 2c). Characteristic dynamics of WNT, FGF and retinoic acid signalling components were evident[17,35]. Along the trajectory to neural identity, WNT5A and FGF8 were downregulated, followed by WNT8A and SPRY2 downregulation, and then the upregulation of WNT4, RXRA and RARB retinoic acid receptors. An equivalent analysis of mouse data showed similar behaviour, despite differences in some specific gene orthologues (Supplementary Fig. 5a). Consistent with previous studies[36,37], our analyses showed that NMPs expressed high levels of BAMBI (a pseudoreceptor inhibiting BMP and NODAL signalling[38]; mirrored in mouse by FST), and nascent PXM and notochord expressed NOG and CHRD (Fig. 2d and Supplementary Fig. 5b).

We considered cells in the median pit to be NotoPs. On the basis of single-cell expression, these are CHRD[+]SPRY2[+] as well as FOXA2[+] CYP26A1[+]CLDN1[+] but lack LEFTY1 expression, a NODAL inhibitor expressed only in mature notochord (Fig. 2d and Extended Data Fig. 1c). Using a custom imaging analysis pipeline (Methods and Supplementary Fig. 6), we clustered cells on the basis of hybridization chain reaction (HCR) signals and mapped these to the embryo (Fig. 2e). HCR for CHRD, LEFTY1, SPRY2 and FOXA2 (Fig. 2f,g and Supplementary Fig. 7a,b) showed a CHRD[+]SPRY2[+] population of cells predominantly in the dorsal caudal region of the node (Fig. 2h,i). FOXA2 was expressed in the same region (Supplementary Fig. 7c). Additionally, the intersection of high levels of NOTO expression and high levels of CNTN2 expression identifies the same cell population around the node in our data and in long-term resident cells isolated at HH8 (ref. 39 and Supplementary Fig. 8).

Together, these results indicate that there is a structured architecture in and around the node. The metalloprotease ADAMTS18 and CYP26A1 mark ingressing cells at the caudal primitive streak extending ventrally past the node. Dorsally, NEFM together with CYP26A1 and BAMBI marks epithelial neuromesodermal axial progenitors. Rostrally, a small CHRD[+]SPRY2[+]FOXA2[+] domain demarcates the dorsal caudal median pit expressing BMP inhibitors, whereas in the preneural tube, dorsal CNTN2 abuts ventral NOTO before the first SOX1[+] cells mark the future neural tube (Fig. 2j).

## An in vitro model of axial fate decisions

To test the function of specific signalling pathways in the specification of trunk progenitors, we established an in vitro model using human embryonic stem (ES) cells. Several in vitro protocols have been proposed to generate axial progenitors[40–43]. Most of these rely on WNT and FGF activation, and we used these as a starting point to generate human trunk progenitors.

We first used a 3-day monolayer protocol consisting of exposure to FGF2 and CHIR99021 (CHIR), a WNT agonist acting through GSK3β inhibition[44]. Reflecting our observations in the chick, we added BMP and NODAL inhibition. This resulted in a high frequency (89%, $n = 6,997$) of SOX2[+]TBXT[+] cells (Supplementary Fig. 9a). We examined whether geometric confinement resulted in an ordered pattern of gene expression[10,45,46]. Applying the same treatment to both H9 and MasterShef4 human ES cells seeded on circular micropatterned laminin substrates resulted in a pronounced pattern at day 3: SOX2[hi] cells were in the centre and TBXT[hi] cells were at the edge of the colony (Fig. 3a, Extended Data Fig. 3a and Supplementary Video 1). We termed these posterior neuruloids.

Clumps of TBXT-expressing cells formed distinct structures at the periphery of neuruloids. The number of these aggregates scaled with the diameter of the micropattern: the smallest-diameter colonies (200 μm) typically contained a single TBXT aggregate, whereas the larger 500-μm-diameter colonies had 2–3 clusters (Fig. 3b and Supplementary Fig. 9b,c). Cells in the peripheral aggregates expressed the PXM marker TBX6. Consistent with previous observations[9,47,48], our analyses showed that a few TBXT[hi] cells co-stained with FOXA2, indicating the presence of a small number of node-like or notochord cells (Fig. 3c white arrowheads). The central SOX2[hi] cells expressed the neural marker SOX1, and assaying TJP1 (ZO-1) indicated that they had acquired an organized apical–basal polarity (Fig. 3c and Extended Data Fig. 3b).

Time-lapse imaging indicated that colonies were highly dynamic with a stereotypical inward movement (about 4 μm h[−1]) driven by cells located in an intermediate ring between the edge and the colony's centre (Supplementary Video 2). This caused the apical side to fold towards the centre (Fig. 3a–c), resulting in a doughnut-shaped structure at day 3.

Image segmentation indicated an average of 8,749 cells per colony (s.d. 1,521; 3 experiments, each with 3 colonies). The proportion of fates was consistent across different experiments with 40–50% of cells expressing SOX2 alone, 20% TBXT alone and 10% double-positive. Notochord TBXT[+]FOXA2[+] progenitors and TBX6[+]TBXT[−] cells were a minority (2%, Fig. 3d). Cells expressing high levels of TBXT and cells expressing high levels of SOX2 were located at the edge and centre of the colony, respectively, whereas cells expressing mid-levels of both factors were located on the top of the colony, in an intermediate ring position closer to the edge (Extended Data Fig. 3c).

Finally, to allow comparisons with the chick and other vertebrates, we performed single-cell RNA sequencing (scRNA-seq). This confirmed the caudal character of the colonies by expression of CDX2 and HOXB9 and presence of PXM and primitive streak-like populations expressing WNT3A, DKK1, MIXL1, TBX6 and FOXC2 (Fig. 3e and Extended Data Fig. 3d). A large fraction of cells expressed both *SOX2* and *TBXT* mRNA. As in the chick, this NMP cluster expressed NEFM (66% of cells in cluster, $n = 1,739/2,623$). CYP26A1 was also present but more restricted to primitive streak cells. The co-expression of NOTO, CHRD and FOXA2, in addition to SHH and TBXT, in a small group of cells confirmed the presence of notochord cells (Extended Data Fig. 3e). Together, the data indicate the presence of PXM, notochord and neural progenitors, cell populations present in the caudal embryo during neurulation stages, suggesting that posterior neuruloids offer a good model to investigate mechanisms of trunk tissue formation.

## YAP, pERK1 and pERK2 regulate WNT and TBXT

To dissect the relationship between signalling and the pattern of cell types in neuruloids, we examined the early stages of neuruloid development (Fig. 3f). Phosphorylated ERK1 and phosphorylated ERK2 (hereafter, pERK1/2) showed colony-wide activation as early as 3 h after addition of induction medium. By 12 h, pERK1/2 were restricted to a ring at the edge of the pattern that remained at 24 h. Only a few cells in the centre of the colony contained pERK1/2. This pattern was followed by TBXT expression. At 12 h, TBXT[+] cells were found in a scattered fashion

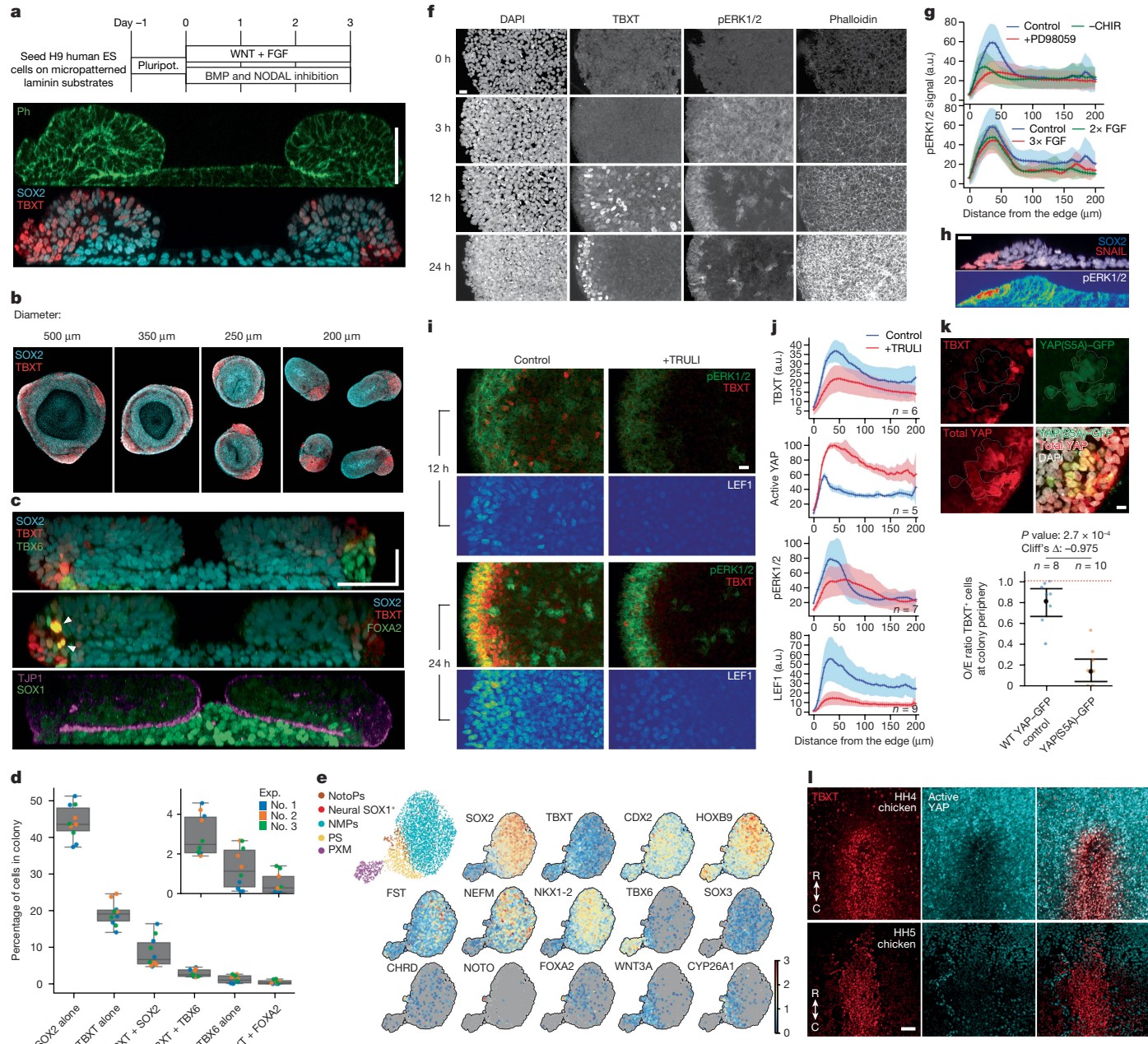

**Fig. 3 | In vitro model of human trunk formation showing that coordinated FGF, WNT and YAP signalling induce TBXT. a**, Top: posterior neuruloid protocol using human ES cell colonies grown on micropatterned laminin substrates. Pluripot., pluripotent. Bottom: optical section immunofluorescence stain of SOX2 (neural), TBXT (mesodermal) and phalloidin (Ph; F-actin). **b**, Size-dependant organization of TBXT+ tissue (red) for colony diameters between 200 and 500 μm. **c**, Medial optical sections of immunofluorescence showing the location of paraxial (TBX6+TBXT+) and axial (FOXA2+TBXT+) mesoderm, SOX2 and the SOX1 neural markers, and the tight-junction marker ZO-1 (TJP1). White arrowheads point to the rare TBXT+FOXA2+ cell population. **d**, Quantification of cells in each neuruloid colony showing cell type proportions obtained by immunofluorescence. $n = 8,749 \pm 1,521$ average cells per colony over 3 distinct colonies and for 3 independent biological experiments (Exp.). Data are represented as box plots as in Fig. 2h. Corresponding data points are overlaid as dots. **e**, scRNA-seq characterization of posterior neuruloids. **f**, Immunofluorescence time course during the first 24 h of the posterior neuruloid. **g**, Quantification of radial pERK1/2 levels at 12 h post-induction (two experiments, each with three samples) without the WNT agonist (CHIR) and with varying concentrations of FGF and its inhibitor (PD98059). a.u., arbitrary units. **h**, Optical section of pERK1/2 at 48 h post-induction co-localizes with SNAIL and TBX6 at the edge of the colony. **i,j**, Treatment with YAP activator (TRULI) results in reduction of TBXT+ and LEF1+ cells at 12 h post-induction. **k**, Expression of a dominant active YAP mutant, YAP(S5A)–GFP, inhibits TBXT expression. Observed over expected (O/E; all colony) ratio of TBXT+ cells in wild-type versus YAP(S5A)–GFP. Point plot represents mean and s.d. **l**, Staining of TBXT and active YAP (unphosphorylated) in a chicken embryo at full primitive streak (HH4) and head process (HH5) stages. Images in **a**–**c**,**i**,**k** are representative of at least three independent experiments and images in **f**,**h** are representative of two independent experiments. Images in **l** are representative of two independent experiments with three embryos each. Radial profile plots in **g**,**j** show the mean signal over indicated $n$ colonies across at least two independent experiments. Shaded areas represent the s.d. around the mean. Scale bars, 50 μm (**a**,**c**(horizontal),**l**), 30 μm (**c**(vertical)), 20 μm (**f**,**h**,**i**) and 10 μm (**k**).

throughout the colony, but by 24 h, TBXT had become localized to colony edges (Supplementary Fig. 10a).

The pERK1/2 TBXT-expressing ring was approximately 2–3 cell diameters wide. Application of PD98059 (an inhibitor of MEK1 and MEK2) abolished the ring, indicating that both pERK1/2 and TBXT expression were dependent on FGF signalling (Fig. 3g and Supplementary Fig. 10b). Increasing FGF concentrations did not change the width of the pERK1/2 ring, despite a progressively larger TBX6[+] domain at the periphery by day 3 (Extended Data Fig. 3f). In the absence of FGF, no TBX6[+] cells were present at day 3, indicating that endogenous FGF signalling was insufficient to drive PXM formation. The pERK1/2 ring was also diminished in the absence of the WNT agonist, indicating synergy between WNT and FGF pathways (Fig. 3g and Supplementary Fig. 10b). By 48 h, pERK1/2 was still detected at the edge where the first TBX6[+]SNAIL[+] cells were induced (Fig. 3h). These cells constitute a SNAIL[+] population found underneath the initially flat epithelial layer of the micropattern to initiate the three-dimensional (3D), doughnut-like shape of the neuruloid, reminiscent of the behaviour of prospective mesodermal cells ingressing at the primitive streak.

We examined YAP signalling, as the pathway has been associated with modifying WNT signalling[49,50], and with edge- and density-sensing cell–cell mechanics[51]. The role of YAP signalling during trunk formation remains unclear despite YAP-null mice exhibiting a truncated axis[52] and YAP being implicated in controlling the segmentation clock[53]. In agreement with previous observations[54], our analyses showed that cells at the periphery retain nuclear YAP, whereas internal cells progressively exclude YAP from the nucleus (Extended Data Fig. 4a). We perturbed YAP signalling using TRULI, which promotes nuclear YAP accumulation, probably through inhibition of LATS1 and LATS2 (ref. 55 and Extended Data Fig. 4b). TRULI treatment caused a shift of YAP to the nucleus and a substantial reduction in TBXT[+] cells at 12 and 24 h (Fig. 3i), leading to the absence of axial and PXM derivatives at day 3 (Supplementary Fig. 11). By contrast, the ring of pERK1/2 at 24 h was relatively unaffected albeit lower levels were apparent (Fig. 3i,j and Extended Data Fig. 4c). Thus, the ring of pERK1/2 activity is not dependent on YAP, although YAP signalling can modulate pERK1/2 levels. By contrast, TBXT expression seems to be delayed by YAP activity. Moreover, the expression of YAP(S5A)–GFP (ref. 56), a version of YAP that cannot be phosphorylated by LATS1 and LATS2 and translocates to the nucleus, also resulted in the inhibition of TBXT expression compared to that of the wild-type control (Fig. 3k and Extended Data Fig. 4d–g). Consistent with these data, in control neuruloids YAP[hi] nuclei had reduced TBXT levels at 12 h, 24 h and 48 h (Supplementary Fig. 12a). Conversely, TBXT[hi] cells tended to have low nuclear YAP levels at 12 h.

TBXT is a target of WNT[57]. YAP activation has been linked to both promoting[58] and suppressing[59] WNT signalling. We assessed WNT signalling using LEF1 expression as a readout[60]. As early as 12 h, control neuruloids had mounted a WNT response that was higher at the edge of the colony with levels of LEF1 decreasing towards the centre (Fig. 3i,j). In TRULI-treated samples, expression of LEF1 was diminished or undetectable at 12 h, suggesting that nuclear YAP inhibits WNT signalling activity. At 24 h, LEF1 was observed in TRULI-treated colonies, but compared to the control, there were fewer positive cells and lower levels, suggesting that YAP activity delayed or weakened the WNT response. This suggests that YAP activation blocks rapid TBXT induction by inhibiting WNT signalling. To investigate whether the absence of YAP activity also correlated with TBXT expression in vivo, we assayed HH4 and HH5 chicken embryos for active YAP. Indeed, TBXT expression in cells of the primitive streak and early node is accompanied by downregulation of active, unphosphorylated YAP (Fig. 3l and Supplementary Fig. 12b). Together, these data suggest that persistent WNT and FGF signalling facilitated by loss of active YAP stabilizes TBXT expression, resulting in the upregulation of TBX6 and SNAI2.

## BMP and NODAL timing guides notochord fate

Next we turned our attention to the small number of notochord-like cells in neuruloids. Cross-species transcriptomic comparisons identified a population of TBXT[+]FOXA2[+]CDX2[+] cells that expressed NOTO and CHRD together with FGF and WNT ligands (Extended Data Fig. 5 and Supplementary Fig. 3). The expression of several BMP and NODAL inhibitors in cell populations in and around the node (Fig. 2) prompted us to test the effect of altering the timing and duration of transforming growth factor-β (TGFβ) signalling (Fig. 4a).

Almost no SOX2[+] or TBXT[+] cells were produced in the absence of BMP and NODAL inhibition or by delaying inhibition until 48 -h after addition of WNT and FGF (Fig. 4b). Instead, a marked increase in endoderm cells, co-expressing SOX17 and FOXA2, was evident (Supplementary Fig. 13a). Moreover, scRNA-seq revealed the presence of cells expressing low levels of TBXT, CDX2, BMP4, KDR, HAND1, HAND2, GATA3, GATA4 and GATA6, which is consistent with blood progenitors of the lateral plate mesoderm (Fig. 4c and Supplementary Fig. 13b). Notably, the different populations were organized in a stereotypical fashion (Fig. 4b,e and Supplementary Video 3).

A 24-h delay to the addition of TGFβ inhibitors also resulted in a marked reduction of SOX2[+]TBXT[+] cells. However, in this case many cells adopted a TBXT[+]FOXA2[+] fate (from <2% in constant inhibition, see Fig. 3d, to 44 ± 8%; average 3D segmented nuclei per colony were 4,991 ± 907), suggesting a notochord identity (Extended Data Fig. 6a). These cells were located on colonies' surface and periphery, whereas cells in the centre were SOX2[+] (Fig. 4b and Supplementary Video 4). scRNA-seq data confirmed the cell types (Fig. 4c). The TBXT[+]FOXA2[+] cells expressed NOTO, SHH, CHRD, SOX9, FOXJ1 and SEMA3E, similar to notochord in chick (Fig. 4d and Extended Data Fig. 6b). In addition, SOX2-expressing cells were apparent along with PXM, which expressed TBX6 and MEOX1. A classifier trained on chick and mouse cell clusters correctly placed the different neuruloid populations in our assigned cell fates (Supplementary Fig. 14a,b). Thus, transient TGFβ signalling results in the generation of notochord-like cells along with neural and PXM, whereas prolonged signalling generates endodermal and lateral plate mesoderm tissue.

The effects of delaying TGFβ signalling on colony formation prompted us to examine the dynamics of morphogen gene expression (Fig. 4f). The WNT plus FGF signalling regime rapidly induced NODAL during the first 9 h, before induction of WNT3A ligand itself (12–18 h). CHRD expression preceded SOX17 (24–30 h) by about 6 h and, although NOTO is upregulated at 42 h, it rapidly decreases as BMP4 and HAND1 are upregulated. These results not only argue for a limited window in which NODAL and BMP signalling specifies notochord fate but also highlight the need for ongoing TGFβ inhibition to maintain NMPs.

## A 3D trunk organoid model with notochord

We sought to adapt the signalling conditions defined on micropatterns to 3D culture. We exposed 3D aggregates of human ES cells to the 24-h-delayed TGFβ signalling inhibition regime (as in Fig. 4a orange) and then cultured these for 4 additional days in the presence of a retinoic acid precursor[12] (Fig. 5a). This resulted in the elongation of the aggregates; 70–75% (MShef4, $n = 53$; H9, $n = 36$) had a prominent stripe of TBXT[+] cells in their interior suggestive of a notochord (Fig. 5b and Extended Data Fig. 7). The outer cell layer was SOX2[+]TBXT[−] and had a morphology similar to a neuroepithelium (Fig. 5c). Consistent with the presence of notochord-like structures, our analyses showed that TBXT[+] cells expressed FOXA2 and were SOX2[−]. The presence of SOX2[+]FOXA2[+] foci also indicated that ventral patterning of neural tissue could be occurring.

To assess the cell types present in these 3D structures, which we named notoroids, we performed scRNA-seq (Fig. 5d). These data revealed ventral neural and mesodermal cell populations (Extended

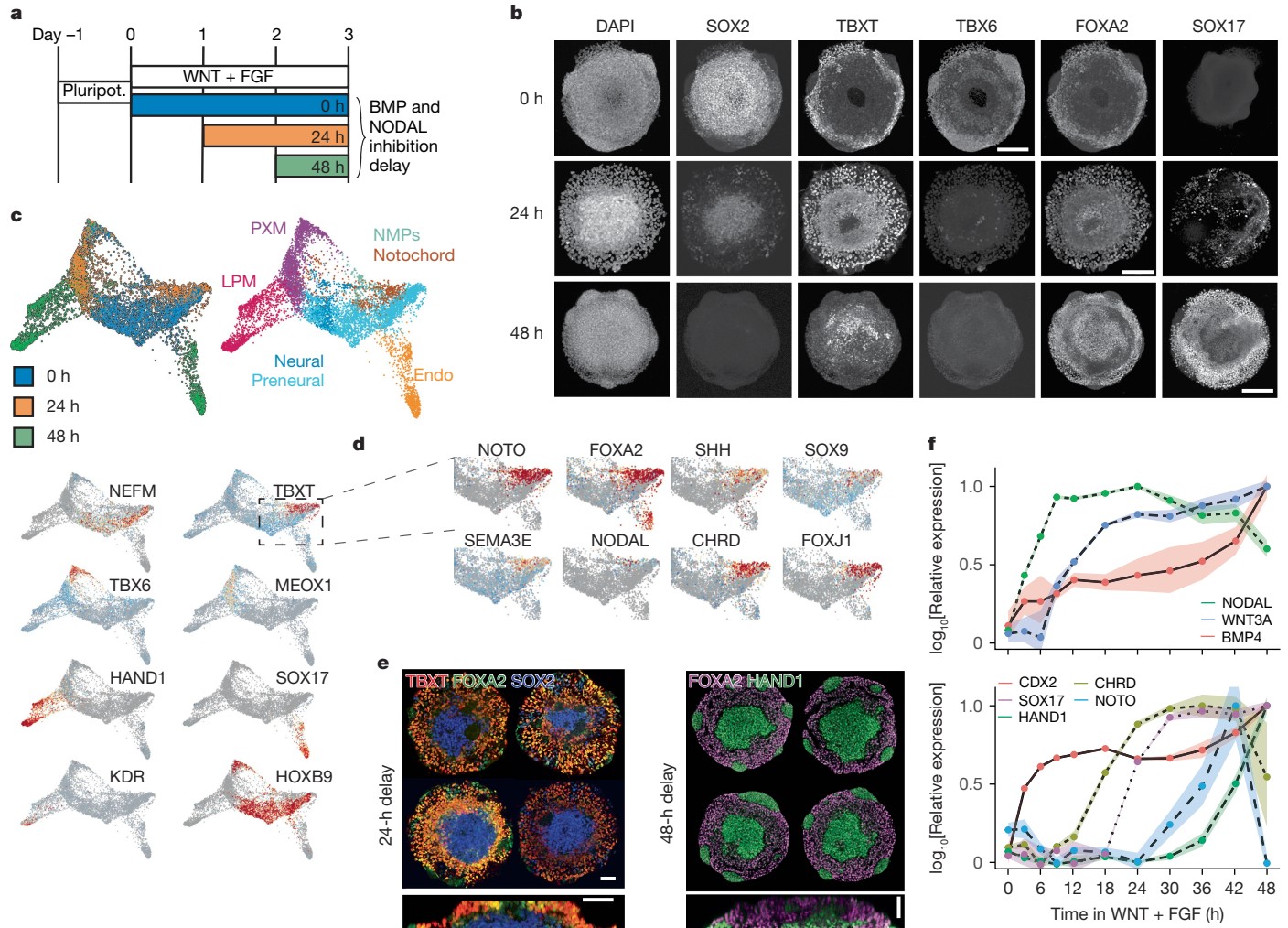

**Fig. 4 | Timing of BMP and NODAL inhibition and the generation of notochord-like cells. a**, Posterior neuruloid differentiation with modifications to the timing and duration of BMP and NODAL inhibition. The control condition with constant inhibition is depicted in blue (0 h); 24-h-delayed inhibition is depicted in orange; and inhibition starting 48 h after induction is depicted in green. **b**, Immunofluorescence for cell type markers (SOX2, TBXT, TBX6, FOXA2 and SOX17) for the different BMP and NODAL delay treatments at day 3. **c**, scRNA-seq characterization of the different inhibition treatments. Combined embedding of constant (blue), 24-h-delayed (orange) and 48-h-delayed (green) inhibition shows largely non-overlapping cell populations generated in the different treatments. Colouring reflects the timing of BMP and NODAL inhibition (that is, the different treatments; top left plot), the cell lineages generated (top right plot) and the expression of a few key genes of each population (bottom plots). **d**, Embedding detail showing the expression of notochord markers in TBXT^hi cells, mostly present in the 24-h-delay treatment (Extended Data Fig. 6). **e**, Reproducibility and spatial segregation of the different 24-h- and 48-h-inhibition-delay micropatterns. **f**, Time course of relative expression of morphogens and cell fate markers from quantitative PCR data collected every 3–6 h during the first 48 h of treatment with WNT agonist (CHIR) and FGF (no BMP and NODAL inhibitors present). Shaded areas represent the s.d. around the mean of three replicates. Images in **b,e** are representative of at least three independent experiments. Scale bars, 100 μm (**b,e**(left bottom)) and 50 μm (**e**(left top and right)).

Data Fig. 8), and classification using mouse and chick embryo data unbiasedly assigned cell identities (Supplementary Fig. 14). We confirmed the existence of a TBXT⁺ cluster of notochord cells co-expressing NOTO, CHRD, FOXA2 and SHH, as well as high levels of FOXJ1, SOX9, PIFO and COL1A2 (Supplementary Fig. 15). A few cells co-expressing GSC and OTX2 were found at day 3, suggesting the presence of an earlier organizer or node fate. At day 7, increasing TGFβ signalling inhibition delay resulted in fewer motor neuron progenitors and an enrichment of notochord and PXM (Fig. 5e).

We validated these findings by image analyses of trunk organoids with inhibition delays ranging from 12 to 36 h (Fig. 5f and Supplementary Fig. 16a,b). The elongation was highest for the 18 h delay and sharply decreased for a 30-h delay resulting in spherical aggregates of high mesodermal content (Supplementary Fig. 16c–e). The amount of mesoderm increased with the amount of time in CHIR + FGF unopposed by

TGFβ signalling inhibitors. The highest ratio of axial to PXM tissue at day 7 was obtained with a 24-h delay (Fig. 5f and Supplementary Fig. 16d).

In the single-cell data and in the TBXT⁺ streaks, we detected SHH expression, indicating that patterning of neural and mesodermal tissue could be occurring (Fig. 5g and Supplementary Video 5). Single-cell data revealed a TBX6⁺FOXC2⁺ mesodermal population at day 3 that progressively downregulated TBX6 and upregulated NKX3-1 (ref. 61), NKX3-2 (ref. 61) and PAX1 (ref. 62) at days 5 and 7, consistent with a sclerotome fate (Extended Data Fig. 8b,c). The neural population showed patterning characteristic of the ventral neural tube. We found p1, p2, motor neuron, p3 and floor plate progenitors[63] (Fig. 5h, Extended Data Fig. 8b and Supplementary Fig. 17a). These were sensitive to an inhibitor of the hedgehog signalling pathway (Extended Data Fig. 9a). Gene expression of the more ventral cell types was associated with the presence of nearby notochord-like cells: notochord and floor plate cells

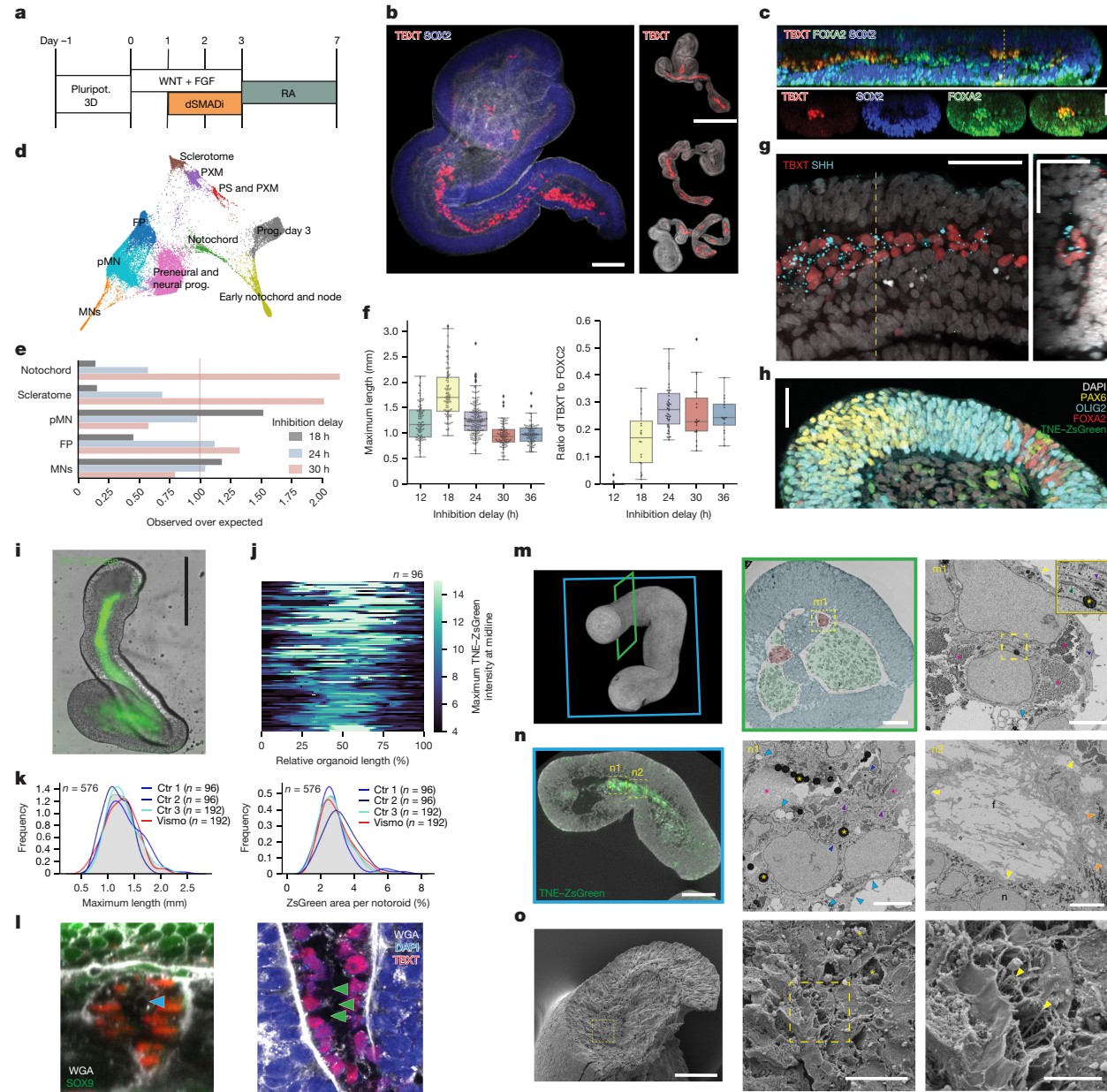

**Fig. 5 | See next page for caption.**

were significantly closer than notochord cells were to PAX6[+] dorsal neural cells (n = 663 notochord cells, 66 µm versus 164 µm; Extended Data Fig. 9b–f).

To validate the notochord cell identity in notoroids, we generated a reporter cell line with a *cis*-regulatory region containing a TBXT notochord enhancer (TNE)[64]. This showed robust expression inside the trunk organoid along the midline (Fig. 5i and Supplementary Fig. 18a,b), overlapping with expression of TBXT, NOTO and CHRD (Extended Data Fig. 10a and Supplementary Video 6). Most notoroids had a strong, uninterrupted signal for an average of 37.1 ± 19.9% of their length (n = 192; 37 ± 5% in volumetric 3D data n = 4; Supplementary Video 7), mostly occurring in the middle of their long axis (Fig. 5j and Extended Data Fig. 10b,c). We estimated the average maximum length to be 1.26 mm at day 7 (n = 576; 2.33 ± 0.133 mm in 3D data, n = 4), with the longest structure being 2.5 mm. Conservative segmentation of the brightest TNE reporter signal showed a positive area for all structures, occupying an average of 3% per trunk organoid that was anti-correlated with total trunk organoid area (R = −0.32, P value = 6.4 × 10[−15]; Fig. 5k and

Supplementary Fig. 18c–e). Vismodegib treatment did not significantly affect elongation or notochord induction, consistent with the continued presence of notochord in *Shh*-null mice[65].

Finally, we assessed the ultrastructure of the notochord in notoroids. TBXT[+]SOX9[+] notochord cells exhibited a noticeable deposition of extracellular matrix (ECM) surrounding them, suggestive of a notochordal sheath (Fig. 5l and Supplementary Fig. 19). Some structures exhibited a lumen, which could indicate the presence of a canal as previously seen in Carnegie stage 8 human embryos[66] (Fig. 5l, blue arrowhead). Furthermore, these cells were morphologically distinct and much larger (Fig. 5l, green arrows) than the tightly packed adjacent neuroepithelium; they were also associated with LAMP1[+] vesicles (Supplementary Videos 8 and 9). Serial block-face scanning electron microscopy (SBF SEM) revealed the expected neuroepithelium surrounding mesenchymal cells and a cluster of cells compatible with known notochord morphology (Fig. 5m and Extended Data Fig. 11). Consistent with a notochord fate, these cells had abundant cytoplasmic glycogen (high and low density), lipid droplets and different-sized vacuoles[67–69]. They

**Fig. 5 | Generation of 3D trunk organoid model with notochord and ongoing ventral patterning. a**, Notoroid protocol. RA, all-trans retinal. **b**, Elongated structures with TBXT⁺ cells surrounded by SOX2⁺ epithelial cells. **c**, Optical sections (at position of dashed line) highlighting TBXT and FOXA2 co-expression. **d**, Combined 2D embedding of notoroid single-cell data for different days (3, 5 and 7) and initial inhibition delays (18, 24 and 30 h post-induction). The plot highlights cell types present. prog., progenitors. MN, motor neurons. **e**, Proportion of cell types present in each inhibition delay condition normalized to total proportion across all conditions. **f**, Left: estimation (GastrUnet) of maximum notoroid length at day 7 per inhibition delay using bright-field images. Right: Ratio of notochord (TBXT) and somitic mesoderm (FOXC2) volume fraction for each trunk organoid generated with 12-, 18-, 24-, 30- and 36-h inhibition delay (*n* = 15, 20, 46, 19 and 16 samples, respectively, over 2 independent experiments) through quantification of immunofluorescence images. Data are shown as a box plot identical to Fig. 3d overlaid with a plot containing individual observations. **g**, Immunofluorescence staining showing SHH in TBXT⁺ notochordal cells. Dashed line indicates position of optical section in adjacent image. **h**, Notoroid cryosection showing ventral neural patterning associated with notochord: floor plate (FOXA2⁺) and motor neuron (OLIG2⁺) progenitors. **i**, Bright-field image of notoroid with TNE–ZsGreen reporter. **j**, Heat map showing the maximum TNE-ZsGreen signal along the standardized length of each notoroid (*n* = 96; see Extended Data Fig. 10c). **k**, Distributions of maximum length and area proportions of ZsGreen signal in control (Ctrl) and vismodegib (Vismo)-treated notoroids. **l**, Optical section of TBXT⁺SOX9⁺ notochordal cells showing extensive ECM deposition. Blue arrowhead marks putative lumen, and green arrowheads highlight enlarged cytoplasm. **m**, Left: 3D view of micro-computed tomography (micro-CT) of notoroid, showing representative transverse (green, **m**) and longitudinal (blue, **n**) orientations for EM imaging. Middle: SBF SEM image showing a transverse section of notoroid with neuroepithelium (blue), mesenchymal cells (green) and putative notochord (red). Right: representative SBF SEM image of notochordal cells. **n**, Left: micro-CT and TNE–ZsGreen overlay of longitudinal notoroid section. Middle: representative SBF SEM image of TNE–ZsGreen⁺ notochord cells. Right: representative SBF SEM image of ECM fibres (f) surrounding the notochord and mitochondria (orange arrowheads) in adjacent neural cells (n). Notochordal cells (in **m** and **n**) were closely packed, had abundant cytoplasmic glycogen (magenta asterisks), desmosomes (purple arrowhead in inset in **m**, and in **n** middle), lipid droplets (yellow asterisks), endolysosomal structures (blue arrowheads), vacuoles (cyan arrowheads) and covered by basal lamina (yellow arrowheads), rough ER (dark green arrowhead in inset in **m**) was also observed. **o**, SEM images of a freeze-fractured notoroid showing lipid droplets (yellow stars) and a sheath of ECM fibres (yellow arrowheads). Images in **b**,**c**,**h** are representative of three independent experiments across two human ES cell lines; images in **g**,**i**,**j**,**l** are representative of at least two independent experiments. Electron microscopy and micro-CT images in **m**–**o** are representative of at least three trunk organoids analysed. Scale bars, 0.5 mm (**b**(right)), 0.4 mm (**i**), 100 µm (**b**(left),**m**(left),**n**(left)), 50 µm (**c**,**g**(left),**h**,**m**(middle),**o**(left)), 30 µm (**g**(right)), 10 µm (**l**,**o**(middle)), 5 µm (**n**(middle),**n**(right)) and 3 µm (**m**(right),**o**(right)).

were covered by a basal lamina and surrounded by a sheath of ECM fibres (Supplementary Videos 10 and 11). The cells were connected by the presence of multiple desmosomes (Supplementary Video 12), as previously seen in the notochord of human embryos[67]. Dilated ER and endolysosomal structures were frequent. In some cells, intermediate filaments, cilia (Supplementary Video 13) and rough endoplasmic reticulum were observed. Correlative light and electron microscopy indicated that TNE–ZsGreen⁺ marked cells had the expected morphology (Fig. 5n, Extended Data Figs. 12–14 and Supplementary Figs. 20 and 21). Freeze-fracture SEM further highlighted the sheath of ECM fibres around the notochord structure (Fig. 5o and Supplementary Fig. 22).

## Discussion

In this study, we identify and locate the main progenitor populations orchestrating vertebrate trunk formation. Guided by this in vivo map, we developed new in vitro platforms to study axial progenitors and trunk tissue formation. These complement existing models of gastrulation using BMP4 (ref. 45) or 'embryo models' of the whole embryo[70], and expand primitive streak models that use WNT3A ligands[71-73].

We showed that despite generalized WNT activation, LEF1 and TBXT induction was restricted to the edge of micropatterns. This required persistent MAP kinase pathway activity and was facilitated by YAP inactivation, which seems to enhance WNT signalling. Consistent with this, TBXT expression is increased in *YAP1*-knockout human gastruloids induced by BMP4 stimulation[74], and mouse ES cells with *MST1* and *MST2* knocked out show impaired mesoderm formation and overall resistance to differentiation[75,76].

In line with the node protecting prospective somite and neural territory from surrounding BMP signalling activity[36,77], our analyses showed that BMP and NODAL inhibition was critical to generate and maintain trunk progenitors and posterior identity. We demonstrate that unchecked WNT and FGF signalling promotes endoderm and lateral mesoderm differentiation through sequential production of endogenous NODAL and BMP. In part, this explains why 3D gastruloid protocols that use CHIR and FGF contain substantial amounts of endoderm[3,4]. We exploit the signalling cascade to expand current 3D models of early human development with a preparation that results in the robust production of notochord cells surrounded by somitic and neural tissue. Previously, notochord cells were lacking from in vitro-derived structures, and notochord has proved difficult to generate by directed differentiation[78,79]. As a defining feature of the chordates, the notochord is responsible for patterning trunk tissues, and consistent with this, our analyses showed that the neural tissue adjacent to in vitro-generated notochordal cells had the molecular characteristics of floor plate and ventral neural progenitors. Together, the data offer new insight into the mechanisms organizing the vertebrate body plan and provide a foundation for future synthetic tissue design.

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

## Methods

### Single-cell chick transcriptomics

Experiments with fertilized hens' eggs followed relevant guidelines and regulations and do not fall under the requirements of the Animals (Scientific Procedures) Act 1986 UK. The eggs, obtained from Henry Stewart & Co., were incubated for 29 to 45 h at 38 °C with about 40% humidity to yield a minimum of 4 embryos with specific somite numbers: 4, 7, 10 and 13 somites. All embryos were dissected to retain tissue caudal to the third somite (inclusive). A single-cell suspension was obtained by incubating the dissected embryos at 37 °C in a dissociation solution consisting of Accutase (Stemcell Technologies) with 3 U mg$^{-1}$ papain (Sigma-Aldrich, 10108014001) and 1 mg ml$^{-1}$ of collagenase 4 (Gibco, 17104019) for 20 min. Half-way through incubation and at the end, the embryos were mechanically dissociated with a P1000 pipette. After dissociation, 200 µl of resuspension buffer (DMEM/F12 with 1% BSA) was added. The cell suspension was then spun for 4 min at 0.6$g$, resuspended in 250 µl of resuspension buffer and filtered through a 40 µm Flowmi cell strainer (136800040) and twice through a pre-wet 20-µm pluriSelect strainer (43-10020-60). The yield and cell viability for the 4S, 7S, 10S and 13S samples were 640, 500, 1,300 and 1,600 cells µl$^{-1}$ with a viability of 88%, 93%, 92% and 95%, respectively. A single-cell suspension was loaded independently for each sample onto the channels of a Chromium Chip G for use in the 10x Chromium Controller (PN-1000120) with the goal of obtaining 10,000 cells. The cells were partitioned into nanolitre-scale gel beads in emulsions and lysed using the 10x Genomics Single Cell 3′ Chip V3.1 GEM, Library and Gel Bead Kit (PN-1000121). cDNA synthesis and library construction were performed as per the manufacturer's protocol for the Chromium Single-Cell 3′ mRNA V3.1 protocol. cDNA amplification involved 12 PCR cycles. Libraries for the samples were multiplexed so that the number of reads matched one lane per sample and were sequenced on an Illumina HiSeq4000 using 100-base-pair paired-end runs.

### Chick scRNA-seq analysis

Reads were aligned to the *Gallus gallus* GRCg6a.101 reference genome using CellRanger (v4.0.0, 10X Genomics) and a custom-made reference 10X package including a gtf file with the protein coding, pseudogene and lncRNA gene biotypes. Read counts were computed using DropEst[80] (v0.8.6) with the parameters -f -V -w -L eiEIBA. The remaining analyses were performed using Scanpy[81] (v1.7.0) unless otherwise indicated. Data for the different chick stages were independently filtered for high percentages of mitochondrial unique molecular identifiers (6–7.5%) and low total counts (2,500). Potential doublet cells were filtered out using Scrublet[82] with thresholds between 0.2 and 0.3. Counts were normalized to a target sum of 10,000 excluding 1% of highly expressed genes. 'Highly variable genes' were called using Scanpy's function with default parameters. Data of different stages were integrated using harmonypy, a python port of the harmony[83] R package. Clustering of cells was performed unbiasedly using the Leiden algorithm with a high-resolution parameter of 3.5, as this distinguished neural crest cells as a separate cluster. Cell clusters with similar marker genes were merged for simplicity; for example, the notochord cluster is composed of subclusters 38, 30 and 35 (Supplementary Fig. 1a), but subclusters 30 and 35 were merged as mature notochord as they both express SHH and LEFTY1. Principal component analysis and uniform manifold approximation and projection (UMAP) embedding were run with default parameters, with a neighbourhood graph computed for 10 neighbours and 40 principal components. Pseudotime inference was performed in R using Slingshot[84] for the clusters between PXM and neural.

### Chick and notoroid RNA fluorescence in situ hybridization

Third-generation in situ HCR DNA probe sets for the chicken mRNA *NOTO*, *FOXA2*, *SPRY2*, *CHRD*, *LEFTY1*, *TBX6*, *CNTN2*, *ADAMTS18*, *NEFM* and *CYP26A1* together with HCR amplifiers, HCR probe hybridization buffer and HCR probe wash buffer were ordered from Molecular Instruments[85]. For human ES cell-based notoroids, probes to CHRD and NOTO were ordered. Chicken embryos with seven somites were dissected to preserve caudal tissue, and multiplex in situ hybridizations were performed according to the manufacturer's protocol (Molecular Instruments HCR v3.0rev7 protocol for whole-mount chicken embryos). The same protocol was followed for notoroids. Embryos were mounted in ProLong SlowFade Mountant (Invitrogen, S36917) and imaged on a Leica SP8 confocal microscope.

### Nucleus segmentation pipeline

To segment 3D nuclei in whole-mount HCR chick embryo confocal stacks and posterior neuroloid micropatterned colonies, we developed a bespoke pipeline using Detectron2 (ref. 86), a pytorch-based computer vision library. To train the model, we manually segmented 13 cropped images of micropatterned neuroloids with a total of 928 nuclei instances. With these images, we used transfer learning from an ImageNet pre-trained cascade R-CNN architecture[87] to obtain an average precision at 0.5:0.95 of 55% and an average precision at 0.5 of 86% on the validation set. The resulting 2D segmentation forms the basis for 3D consolidation. To merge nuclei detected in each $z$-plane of the confocal stack, we use a supervised strategy. First, we construct a graph linking nuclear masks across $z$-planes if they share a minimum area overlap of 30%. This results in a collection of connected components (subgraphs in which nodes are nuclei linked across $z$-planes) that are further refined using two manually defined thresholds, one with the typical number of $z$-planes for a single nucleus and another with a hard, upper limit of this value. Given a particular subgraph with a number of nodes greater than the defined thresholds, we sequentially prune edges in the following order: first for edges with Jaccard distance greater than 0.7, then the edge with highest Jaccard distance for nodes sharing the same $z$, then for the next edge with the highest distance, and finally, if the subgraph is still larger than the hard limit and no distance is above 0.2, a random edge is taken. Each individual subgraph constitutes a single-nucleus model. All nuclei are taken to generate a 3D mask that is used to calculate nuclear features such as average channel intensity. Cell-based features were analysed independently of their 3D coordinates for clusters and specific cell populations. These were subsequently visualized by mapping the clusters' identities back to the embryo coordinate space. For the HCR analyses in Fig. 2, the levels of each marker gene were classified in three categories: not expressed, low and high expression level.

### Human ES cell culture

H9 (WiCell) and MasterShef4 (UKSCB) human ES cells were routinely cultured in StemFlex medium (Thermo Fisher Scientific, A3349401) on 0.5 mg cm$^{-2}$ laminin-coated plates (Thermo Fisher Scientific A29249). Cells were passaged using ReLeSR according to the manufacturer's instructions (StemCell Technologies, 05872). Cells were tested for *Mycoplasma* spp. at 3-month intervals.

All human ES cell experiments were performed at the Francis Crick Institute and followed the Guidelines for Stem Cell Research and Clinical Translation published by the International Society for Stem Cell Research and the UK Code of Practice for the Use of Human Stem Cell Lines. Our culture system models a specific stage of development and complies with section 2.2.1A of the International Society for Stem Cell Research Guidelines. The WiCell line H9 was used under agreement 17-W0054 'Developmental dynamics of tissue formation'. The work was approved by the Steering Committee for the UK Stem Cell Bank and for the Use of Stem Cell Lines (ref. SCSC20-13).

### Generation of micropatterned posterior neuroloids

Coverslips with micropatterned laminin were generated using an adapted protocol described previously[88]. In brief, isopronanol-cleaned 18-mm coverslips were UVO-cleaned for 10 min before incubation with

PLL-g-PEG(5) for 1 h in the dark. Coverslips were rinsed three times in deionized water before placing them on a custom-made chrome mask previously activated by UVO-cleaning (2 min). Close contact between the mask and coverslip was ensured by pressing firmly with a pipette tip. The reverse, silver side of the mask was exposed to ultraviolet for 8 min, after which coverslips were placed in 70% ethanol for 15 min. The patterned coverslips were allowed to dry and used within 4 weeks.

To seed cells, coverslips were first incubated for 3 h at 37 °C with rh-Laminin-521 (Thermo Fisher Scientific, A29248) diluted 1:10 in PBS+/+ (PBS with calcium and magnesium; Thermo Fisher Scientific, 14040-091) and thoroughly washed with PBS+/+ as detailed previously[46]. Cells were then dissociated by washing with PBS (PBS without calcium and magensium) once, followed by incubation with Accutase for 5 min at 37 °C. Mechanically dissociated cells in Accutase were diluted with a 4× volume of StemFlex with 10 µM Y-27632 (ROCK inhibitor, Tocris, 1254) and manually counted with trypan blue. Concentration of the single-cell suspension was adjusted to 670,000 cells ml$^{-1}$ with StemFlex with 10 µM Y-27632. Cells were seeded by adding 3 ml of the single-cell suspension to the coated coverslip in a 6-well plate well. After 3 h, the wells with coverslips were washed once with PBS and the medium was replaced with fresh StemFlex for overnight incubation. The following day (18 h after), the colonies were induced by first washing with PBS and then adding 2.5 ml of 3 N induction medium[46,89] with CHIR (3 µM), FGF2 (5 ng ml$^{-1}$), SB431542 (10 µM) and LDN193189 (0.1 µM, Selleck Chemicals). The medium was replaced the next day. The coverslips were fixed at day 3 (80 h) in 4% fresh PFA for 30 min at room temperature, washed twice in PBS and stored at 4 °C until further analysis. The YAP(S5A)–GFP (ref. 56) was generated with Addgene plasmid 174170 and using the lentivirus protocol described below.

## Single-cell transcriptomics of posterior neuruloids

Coverslips with micropatterned neuruloids and the two protocol variations of 24-h and 48-h delay in the TGFβ family NODAL and BMP inhibitors (SB431542 and LDN193189) were dissociated by first washing with PBS and then incubating with Accutase for 10 min at 37 °C following mechanical dissociation. Accutase suspension was then diluted in a 4× volume with resuspension buffer, washed and strained as described above for chicken transcriptomics. The yield was 1,110, 950 and 1,170 cells µl$^{-1}$, respectively, with a viability >95%. The samples were separately loaded for capture with the Chromium System using the Single Cell 3′ v3.1 reagents (10X Genomics). Reads were aligned to the *Homo sapiens* GRCh38-3.0.0 reference genome using CellRanger (v4.0.0, 10X Genomics), and the analyses were performed using Scanpy[81] (v1.7.0). Cells were filtered for a minimum of 200 genes, mitochondrial unique molecular identifiers between 1 and 20%, and total counts between 10,000 and 50,000, and doublet cells were filtered using Scrublet[82] with a threshold of 0.2. Counts were normalized to a target sum of 10,000 excluding highly expressed genes (3%). 'Highly variable genes' were called using Scanpy's function with default parameters. Data of different conditions were integrated using harmonypy. Principal component analysis and UMAP were run with default parameters, with a neighbourhood graph computed for 15 neighbours and 30 principal components. A 2D embedding using force-directed graph drawing was computed by taking the UMAP coordinates as the initial position.

## Quantitative PCR analysis of micropatterns

Plates with the micropatterned coverslips were washed with PBS, and the cell colonies were lysed in RLT buffer (QIAGEN 1015762). RNA extraction was performed using the RNeasy mini kit (QIAGEN 74106) according to the manufacturer's instructions. cDNA synthesis was performed using Superscript III (ThermoFisher 18080051) from 1 µg RNA using random hexamers and the cDNA was amplified using PowerUp SYBR green (Applied Biosystems A25918). Quantitative PCR with reverse transcription was performed using the QuantStudio 12K Flex

Real-Time PCR system (ThermoFisher) and the SYBR Green PCR assay (ThermoFisher A25742). Expression values for each gene were normalized against *ATPF1*, using the $\Delta\Delta C_t$ method implemented in the 'pcr' R package. Relative expression across the time course was computed by normalizing $\log_{10}$[expression + pseudo-count] to the maximum value. Primer sequences are given in Supplementary Table 2.

## 3D cultures

To generate 3D cultures, we first started a pre-culture of human ES cells by seeding 200,000 cells onto a 6-cm Petri dish with StemFlex medium with 10 µM Y-27632 (ROCK inhibitor) for 2 days. Cells were then dissociated using 0.5 ml of Accutase for 4 min at 37 °C and added to 4 ml of StemFlex with Y-27632 for manual counting with trypan blue. A total of 1,000 cells were seeded on each 96-well plate well with 80 µl of StemFlex with Y-27632, and the plate was spun for 3 min at 140$g$. The cells were then allowed to aggregate for 5–6 h before slowly adding 150 µl of StemFlex medium per well. Cell aggregates were induced 15–16 h after (next day) by washing twice with PBS and replacing twice the medium present with 150 µl of 3 N medium used in the posterior neuruloid protocol, with 3 µM CHIR and 5 ng µl$^{-1}$ FGF2. After 24 h, the medium was replaced with 3 N medium with LDN193189 and SB431542 added, for 2 days. At day 3, the medium was replaced with 3 N medium supplemented with 40 nM all-*trans* retinal (Sigma, CAS 116-31-4) for 4 days. At the day 7 end-point, cultures were fixed in fresh 4% PFA for 2 h at room temperature and thoroughly washed with PBS before further analysis. Vismodegib (GDC-0449; APExBIO) was used at 5 µM.

## Analyses of 3D cultures

Shape analysis of 3D cultures was performed from phase-contrast microscope images segmented using a custom-made neural network pipeline using the deep learning library fast.ai (v2.7.10) - GastrUnet. In brief, 25 phase images were manually segmented, augmented using the albumentations package, and used to train a dynamic U-Net with resnet34 architecture that includes self-attention layers and Mish activation function. The accuracy of this semantic segmentation task was 0.96. GastrUnet was subsequently used to analyse fluorescence images including TBXT and FOXC2 stains as well as the TNE–ZsGreen reporter line. Light-sheet microscopy data were acquired with a Bruker MuVi SPIM microscope, and volumetric 3D measurements were performed using Imaris v9.5.1 and Fiji v2.14.

## Immunostaining

Micropatterned cells and 3D colonies were blocked and permeabilized for 1 h at room temperature in PBS with 1% Triton-X, 10% dimethylsulfoxide, 10% SDS and 4% normal donkey serum (D9663-10ml Sigma). They were then rinsed for 1 min in PBS before overnight incubation with primary antibodies at 4 °C. After incubation, samples were washed three times for 5 min in PBS and incubated overnight with secondary antibodies conjugated with Alexa Fluor 488, 555, 594 and 647 (1:1,000 dilution) and 10 ng ml$^{-1}$ of DAPI (ThermoFisher Scientific). Finally, samples were washed for 10 min in PBS before mounting with ProLong Glass Antifade Mountant (Invitrogen, P36980). Images were acquired using Zeiss LSM880 Zen or Leica SP8 software. Antibodies used are listed in Supplementary Table 3.

## Single-cell transcriptomics of notoroids and cross-species comparisons

A total of 16 (day 7) to 32 (day 3) trunk organoids were dissociated by first washing with PBS and then incubating with Accutase for 10 min at 37 °C following mechanical dissociation and another incubation at 37 °C (5 min of day 3, 10 min of day 7). Accutase suspension was then diluted in a 4× volume with resuspension buffer, washed and strained as described above for chicken transcriptomics. The single-cell suspensions obtained were diluted to 1,300 cells µl and had a viability >95%. Analyses were performed as described above for neuruloids.

To classify cell types across species, a random forest classifier with 5,000 trees was trained on chick and mouse cell clusters with 25% data held for testing. The classifier used 166 genes representing the intersection of highly variable genes across species. Orthologous genes were found using ensmbl database, these were further refined manually by matching external gene names. Non-unique translations across species were removed except for the entries for which gene names matched. In the classifier, all genes were translated to human.

## TNE reporter line

To establish a reporter of TNE *cis*-regulatory region activity, we cloned TNE[64] upstream of the *Shh* minimal promoter[90] and a ZsGreen fluorophore into a lentivirus backbone harbouring a puromycin resistance gene. We stably integrated this into human ES cells at a multiplicity of infection of <1 and isolated clones. To generate lentivirus, HEK293T were plated at 1.5 M per 6-cm plate. The next day, the medium was changed (3.5 ml). A mixture of third-generation packaging plasmids (0.24 µg of CMV-Rev, 0.46 µg pMDLg and 0.34 µg VSV-G) and 2.28 µg of the transfer plasmid was vortexed with 11.9 µl XtremeGene-HP (Roche) and 360 µl Optimem (GIBCO). This was added dropwise on top of the cells. The medium was changed 16 h later to StemFlex and collected 30 h later. The medium was filtered through a 0.45-µm filter, aliquoted and frozen at −80 °C. Aliquots were thawed and used at 200 µl ml$^{-1}$ StemFlex with Y-27632. Puromycin selection (5 ng ml$^{-1}$) was started 2 days later. A titration curve was used to identify the volume of virus that yielded less than 30% of surviving colonies with respect to an uninfected, unselected control.

## Micro-CT and SBF SEM

For SBF SEM, samples were fixed by adding 8% (v/v) formaldehyde (Taab Laboratory Equipment) in 0.2 M phosphate buffer (PB) pH 7.4 to the cell culture medium (1:1) for 60 min at room temperature for Fig. 5m and Extended Data Fig. 11 or overnight at 4 °C for Fig. 5n and Extended Data Figs. 12–14. For correlative light and electron microscopy work, TNE–ZsGreen samples were then washed and imaged in 0.1 M PB using a Leica SP8. Samples were then processed using a Pelco BioWave Pro+ microwave (Ted Pella) and following a protocol adapted from the National Centre for Microscopy and Imaging Research protocol[91]. Each step was performed in the Biowave, except for the PB and water wash steps, which consisted of two washes on the bench followed by two washes in the Biowave without vacuum at 250 W for 40 s. All of the chemical incubations were performed in the Biowave for 14 min under vacuum in 2 min steps alternating with/without 100 W power. The SteadyTemp plate was set to 21 °C unless otherwise stated. In brief, the samples were fixed again in 2.5% glutaraldehyde (TAAB)/4% formaldehyde in 0.1 M PB. They were then stained with 2% osmium tetroxide (TAAB)/1.5% potassium ferricyanide (Sigma), incubated in 1% thiocarbohydrazide (Sigma) with the SteadyTemp plate set to 40 °C, and further stained with 2% osmium tetroxide in double-distilled H$_2$O. The samples were then incubated in 1% aqueous uranyl acetate (Agar Scientific) with the SteadyTemp plate set to 40 °C, and washed in distilled H$_2$O with the SteadyTemp set to 40 °C. Samples were then stained with Walton's lead aspartate with the SteadyTemp set to 50 °C, and dehydrated in a graded ethanol series (20%, 50%, 70%, 90% and 100%, twice each), followed by three dry acetone washes at 250 W for 40 s without vacuum. Exchange into Durcupan ACM resin (Sigma) was performed in 25%, 50% and 75% resin in acetone, followed by 4 pure Durcupan steps, at 250 W for 3 min, with vacuum cycling (on/off at 30-s intervals), before embedding at 60 °C for 48 h.

The resin blocks were then trimmed and mounted for micro-CT on SBF SEM specimen holders using conductive epoxy resin (Circuitworks CW2400). Tomographic imaging was conducted in an Xradia Versa 510 (Carl Zeiss). Low- and high-resolution scans were captured at 60 kV and 5 W, with 4× and 20× objectives and pixel size of 2 µm and 0.5 µm, respectively. The tomograms were reconstructed using

the Reconstructor program (Carl Zeiss) and exported as tiff files. The regions of interest were identified in each block using the Crosshair plugin in Fiji[92].

Before the SBF SEM imaging runs, the samples were coated with a 2-nm layer of platinum to further enhance conductivity. SBF SEM data were collected using a 3View2XP (Gatan) attached to a Sigma VP SEM (Carl Zeiss) instrument. The SEM instrument was operated in high vacuum with focal charge compensation on and set to 50–70% with a 20 µm or 30 µm aperture for Fig. 5m and Fig. 5n, respectively, at an accelerating voltage of 1.8 kV. Inverted backscattered electron images were acquired with a 2 µs dwell time for each 50-nm slice and consisted of a low-resolution overview image (pixel size of 40–80 nm) and several high-resolution images of the different regions of interest (pixel size of 8.3–9 nm). A minimum of 50 consecutive slices were acquired for each ROI to get enough 3D information. All of the images were converted to tiff format in Digital Micrograph (Gatan), and the resulting tiff stacks were aligned and montaged using TrakEM2 plugin in Fiji[93].

For micro-CT–ZsGreen and low-resolution SBF SEM–ZsGreen 3D registrations, image stacks from confocal microscopy and micro-CT were manually aligned to each other using the BigWarp plugin in Fiji[94]. An affine transformation was applied to the confocal data, which was then merged with the micro-CT data to produce the overlay shown in Fig. 5n and Extended Data Figs. 13a,h and 14a. This micro-CT–ZsGreen composite was then registered to the low-resolution SBF SEM stack, using an affine transformation. The transformed ZsGreen channel was then merged to the SBF SEM stack to create the overlay shown in Extended Data Figs. 13b,i and 14b.

## Freeze-fracture SEM

Samples were fixed by adding 8% (v/v) formaldehyde in 0.2 M PB pH 7.4 to the cell culture medium (1:1) for 60 min at room temperature. The samples were then fixed in 4% formaldehyde/2.5% glutaraldehyde in 0.1 M PB pH 7.4 for 1 h at room temperature. The samples were then washed three times for 5 min in 0.1 M PB and cryoprotected in 25% sucrose, 10% glycerol in 0.05 M PB overnight. They were then fast-frozen on filter paper and fractured in liquid nitrogen using a combination of scalpel and pick. The samples were then placed back into the cryoprotectant at room temperature and allowed to thaw. After three washes in 0.1 M PB, the samples were stained in 1% OsO$_4$/1.5% potassium ferricyanide, washed in 0.1 M PB and H$_2$O, and dehydrated in a graded ethanol series (50%, 70%, 90%, 2 × 100% ethanol for 20 min), critical point dried with CO$_2$ using a Leica EM CPD300 and mounted on aluminium stubs using adhesive carbon tabs. The samples were mounted to present the fractured surfaces to the beam and coated with a thin, 2 nm, layer of platinum using a Quorum Q150 R S sputter coater. SEM images were recorded with an FEI Quanta 250 FEG scanning electron microscope with the ETD detector, at 3 kV, spot 2.5 with a 5 µs dwell time.

## Reporting summary

Further information on research design is available in the Nature Portfolio Reporting Summary linked to this article.

## Data availability

scRNA-seq data have been deposited in the Gene Expression Omnibus (GEO) under the accession numbers GSE223189, GSE224404 and GSE255338 for chick trunk, human micropatterns and 3D notoroids, respectively. SBF SEM data have been deposited in the Electron Microscopy Public Image Archive (EMPIAR) with accessions 12161, 12162, 12163 and 12164.

## Code availability

The scripts to analyse single-cell transcriptomic data are available via GitHub at https://github.com/tiagu/trunk_dev_scRNAseq. The nucleus

segmentation pipeline is available via GitHub at https://github.com/tiagu/Nucleus, and the respective PyTorch models have been deposited at Zenodo at https://doi.org/10.1101/2023.02.27.530267 (ref. 95). The pipeline to analyse 3D cultures is available via GitHub at https://github.com/tiagu/gastrunet and the respective model at Zenodo at https://zenodo.org/records/12684780 (ref. 96).

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

**Acknowledgements** We thank N. Tapon, T. Frith and R. Maizels for discussions; N. Moris, S. Santos and A. Warmflash for critically reading the manuscript; R. D'Antuono, A. Ciccarelli, M. Renshaw, D. Bell, L. Collinson, M. Howell, E. Nye and C. Dix for help; and the Advanced Sequencing Science Technology Platform; C. Soudy and B. Aerne for TRULI. This work was supported by the Francis Crick Institute, which receives its core funding from Cancer Research UK (CC001051), the UK Medical Research Council (CC001051) and the Wellcome Trust (CC001051); by the Engineering and Physical Sciences Research Council (UK) (EP/W023865/1), and by the Wellcome Trust (220379/D/20/Z). A.R.G.L. was supported by a long-term fellowship from the European Molecular Biology Organization (ALTF 149-2020). J.C.-S. was supported by a Boehringer Ingelheim Fonds and Francis Crick Institute PhD Fellowships. For the purpose of open access, the authors have applied a CC BY public copyright licence to any author accepted manuscript version arising from this submission.

**Author contributions** T.R. and J.B. conceived the project, interpreted the data and wrote the manuscript. T.R. and A.R.G.L. dissected chick embryos and performed scRNA-seq and HCR. T.R. performed human ES cell experiments. J.C.-S. and T.R. designed and built the TNE–ZsGreen reporter. T.R. devised and performed bioinformatic and image analysis. T.R. and M.D. characterized posterior neuruloids by quantitative PCR and performed trunk organoid experiments. M.-C.D. characterized 3D notoroids by electron microscopy.

**Funding** Open Access funding provided by The Francis Crick Institute.

**Competing interests** The authors declare no competing interests.

**Additional information**
**Correspondence and requests for materials** should be addressed to Tiago Rito or James Briscoe.

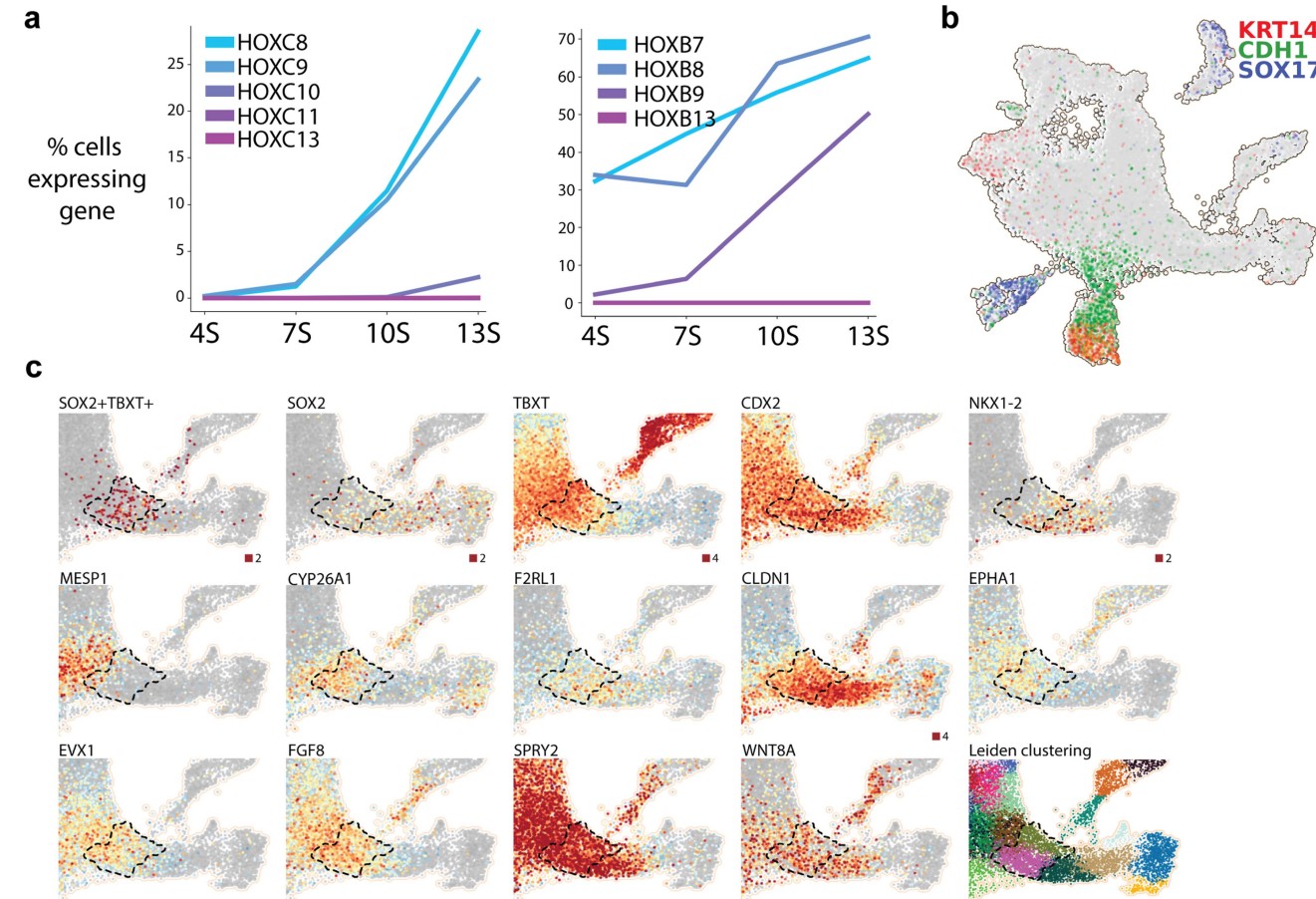

**Extended Data Fig. 1 | Single-cell RNA-seq of chick trunk development: HOX induction and markers of the neuromesodermal cell population. a**, Percentage of cells expressing different genes in the HOXB and HOXC clusters showing induction of HOXB/C9 between 7-10S. **b**, Embedding of chick trunk (4-13S) coloured by markers of endoderm and surface ectoderm. **c**, UMAP embedding detail plots highlighting the expression of several genes at and around the NMP cluster. Linear scale with maximum counts indicated by the squares.

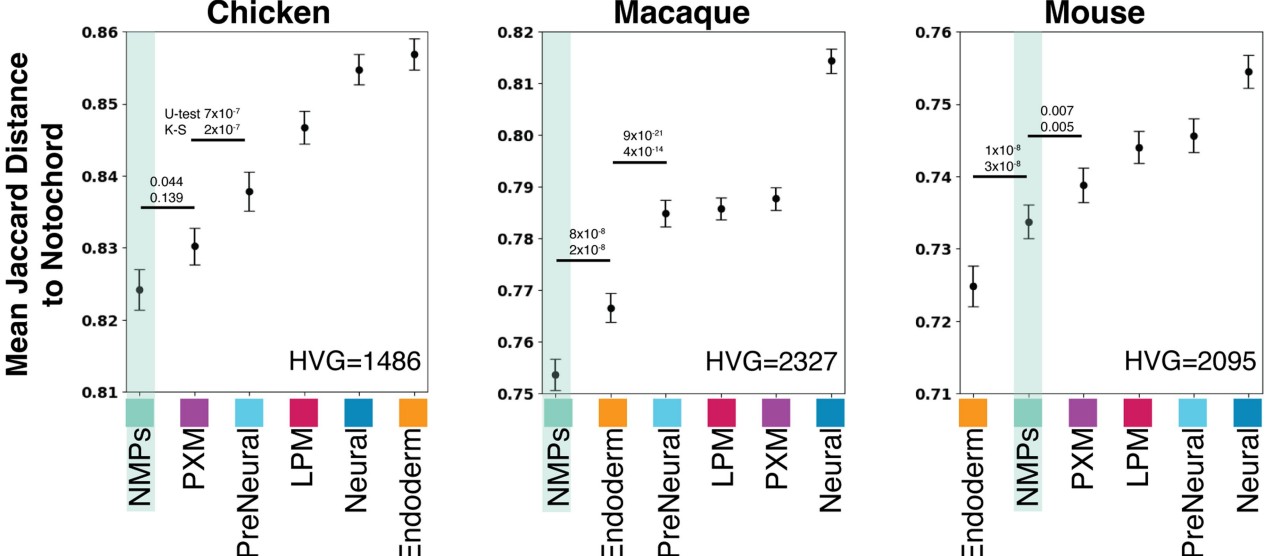

**Extended Data Fig. 2 | Transcriptome proximity between notochord cells and other trunk cell types.** Mean and 95% confidence interval of the Jaccard distances between the binarized transcriptomes of 2000 random pairs of cells from two given clusters. Highly variable genes calculated for each species were used to subset transcriptomes and comparisons were filtered to keep just those involving the Notochord cluster. The p-values indicated correspond to a Mann-Whitney U Test (top) and a Kolmogorov-Smirnov test (K-S, bottom values) performed using the pairwise jaccard distances between two pairs of cell clusters.

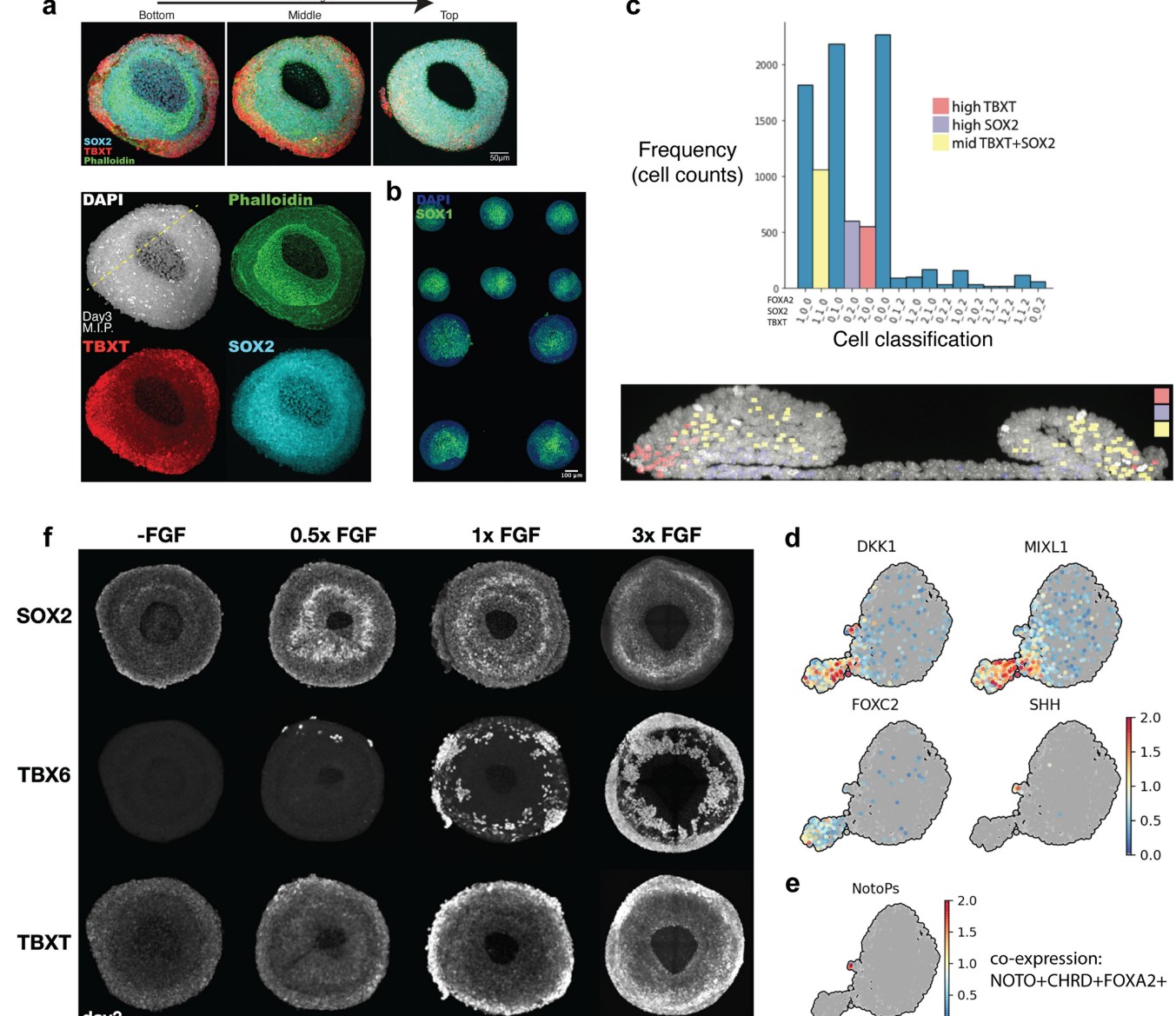

**Extended Data Fig. 3 | Additional posterior neuroloid characterization.**
**a**, Single-z plane images and maximum intensity projection (M.I.P.) of
immunofluorescence staining of SOX2, TBXT and phalloidin of day 3 posterior
neuroloid. **b**, SOX1 immunostaining overview illustrating the robustness of the
platform. **c**, SOX2, TBXT and FOXA2 nuclear levels in day 3 posterior neuroloids
were discretized into low, mid and high-levels. Optical section showing high
SOX2 cells located at the centre of the colony whereas high TBXT cells occupy a
position at the periphery. Mid TBXT + SOX2+ cells localize to an intermediate
ring in the colony. **d**, UMAP plot of neuroloid day 3 single-cell RNA-seq showing

additional gene markers for mesoderm population and SHH expression in
the notochord progenitors. **e**, UMAP plot highlighting 7 NotoP single cells
co-expressing NOTO + CHRD + FOXA2 + . The presence of these cells was both
rare and variable across replicates which formed the basis for our exploration
of conditions to further increase them. **f**, M.I.P. of mesodermal populations
(TBX6 and TBXT) in the day 3 neuroloid with various levels of FGF. Images in a-b
are representative of at least three independent experiments and images in f
representative of two independent experiments.

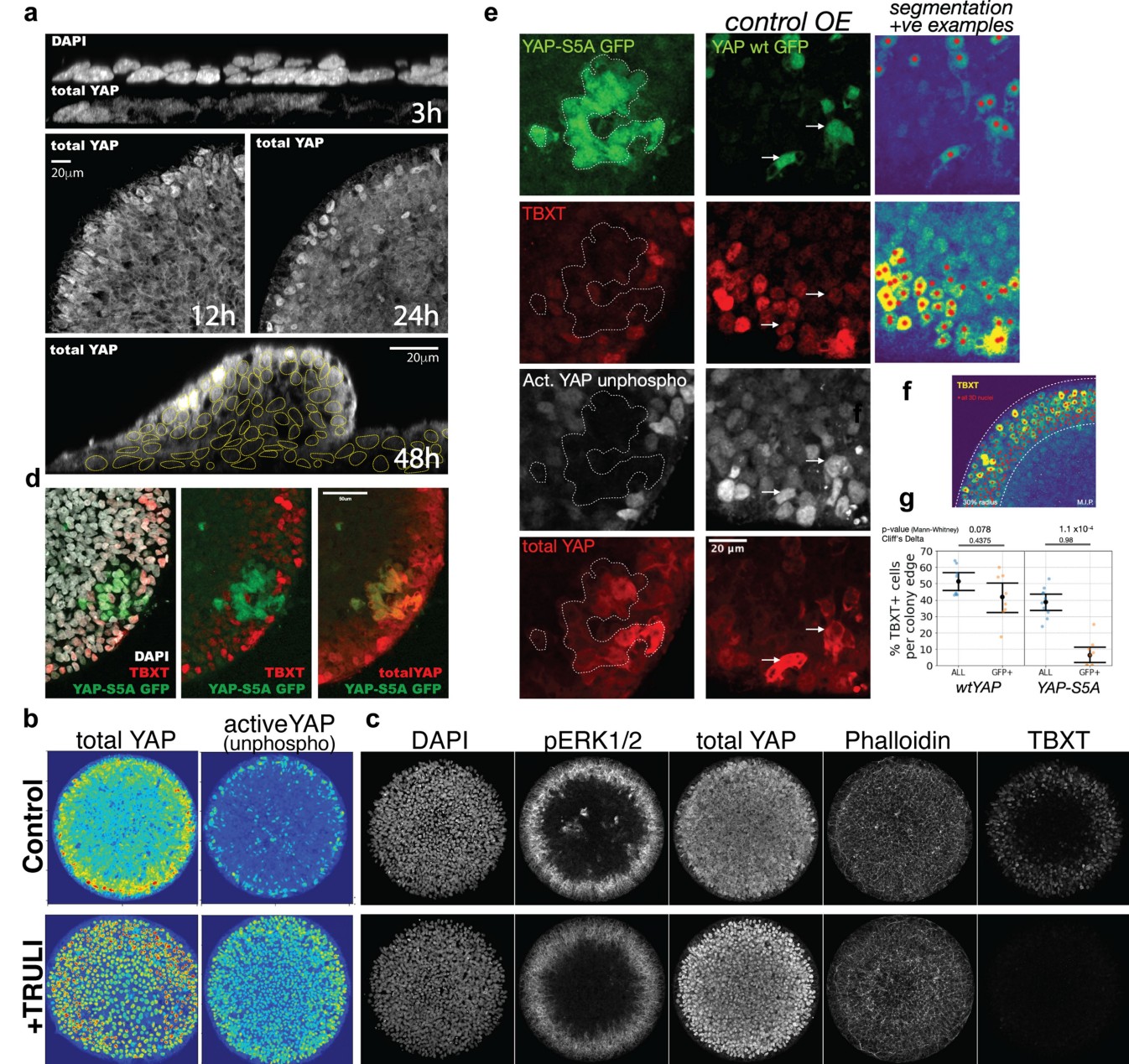

**Extended Data Fig. 4 | YAP signaling and YAP activation on posterior neuruloids. a**, Localization of total YAP during neuruloid formation. **b**, Treatment with TRULI promotes nuclear (total) YAP at 12 h post-induction and active YAP usually confined to the edge is now throughout the colony. **c**, Additional stains of posterior neuruloid at 12 h post-induction without and with TRULI showing a dampened pERK1/2 ring and no TBXT+ cells as well as the nuclear accumulation of YAP. Differences in phalloidin staining were also observed. M.I.P. is shown. **d**, Edge of neuruloid colony at 24 h with S5A-YAP-GFP expressing cells. **e**, Zoom of a neuruloid colony at 24 h showing ring of TBXT at the edge and YAP-S5A-GFP vs wtYAP-GFP expression. In contrast to wtYAP-GFP, YAP-S5A-GFP+ cells have low/no TBXT; they show high YAP levels both nuclear

and cytoplasm and as expected from the mutated serine residues, show no signal for active, unphosphorylated YAP. Quantification of the colonies' edge (30% radius, panel **f**) compares the proportion of TBXT+ cells in GFP+ cells versus all cells showing a significant reduction in YAP-S5A case (p-value = 0.00011, panel **g**). From the images analysed this is a conservative estimation, impaired by cell proximity or segmentation errors. Another way to quantify this effect is by using, colony-wise, the observed TBXT+ cells of all cells at the edge as expectation for the TBXT + GFP+ population. This shows ratios close to 1 for the wtYAP and closer to zero for the YAP-S5A (p-value = 0.00027; main Fig. 3k). Scale bars: 20 µm (a,e), 50 µm (d), 250 µm colony (b,c).

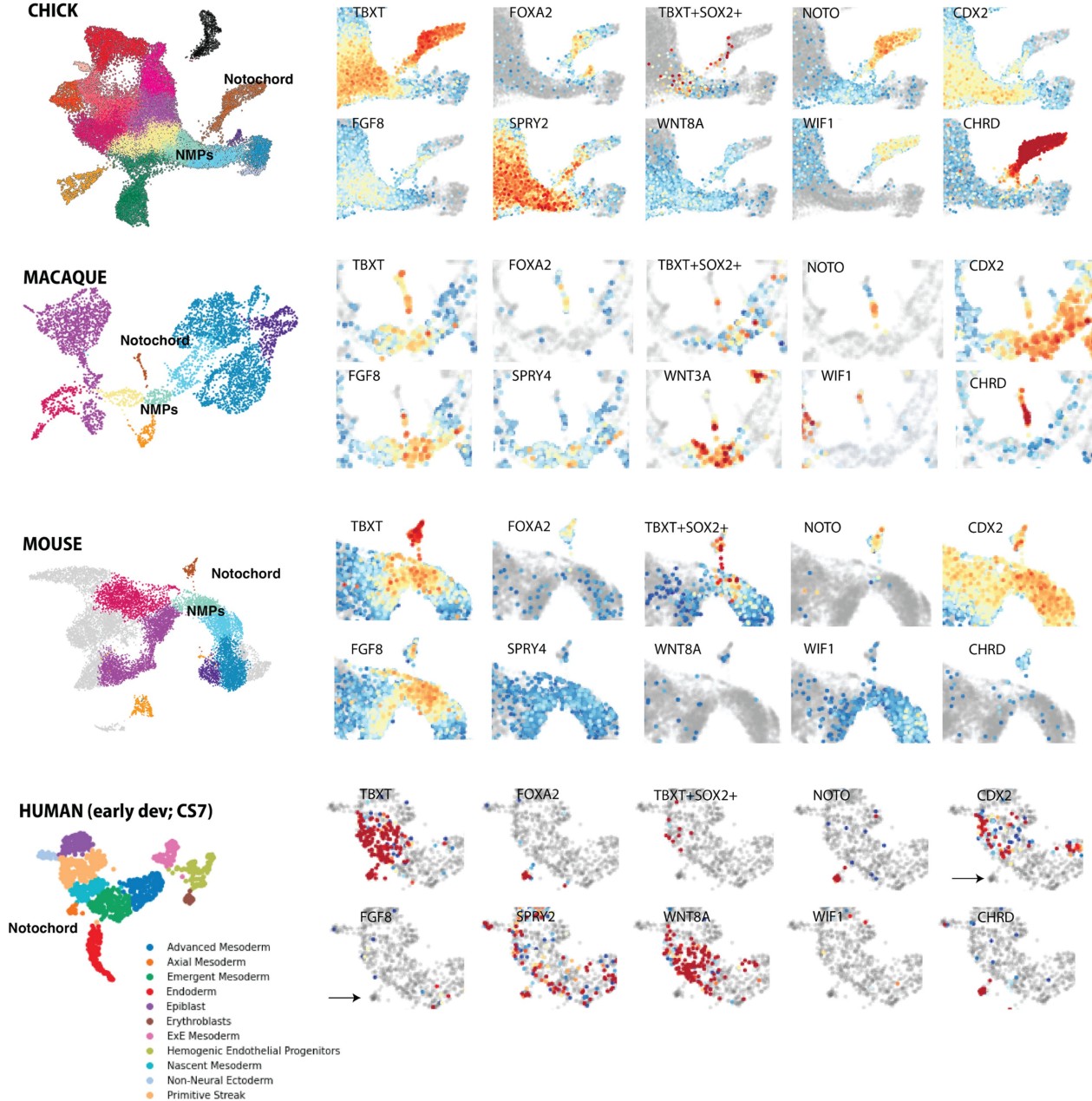

**Extended Data Fig. 5 | Cross-species single-cell RNA-seq comparison of notochord during trunk formation.** Cross-species gene expression comparison of the TBXT + CHRD + NOTO + CDX2+ notochord cell populations in chick, macaque, mouse and human CS7 single-cell transcriptomics data. The human data is from an earlier developmental stage; note the absence of CDX2, WNT8A and FGF8 expression.

**a**

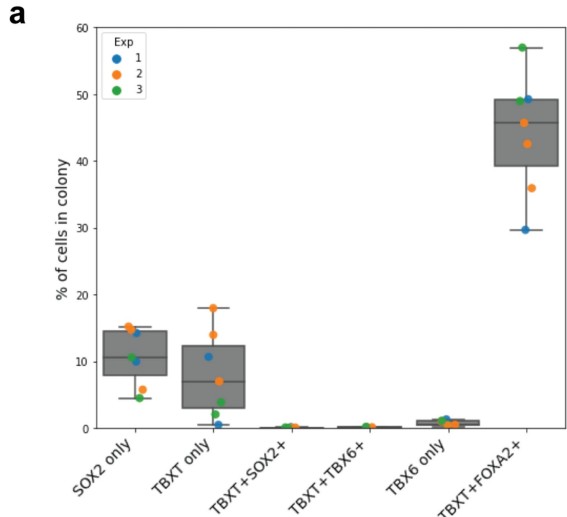

**b**

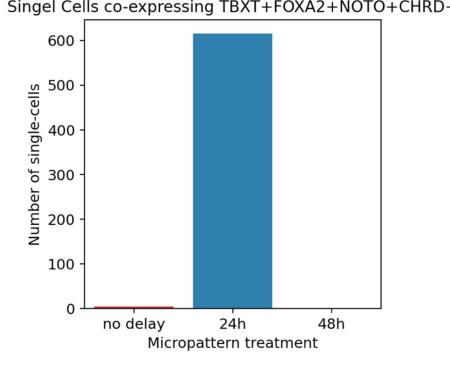

Singel Cells co-expressing TBXT+FOXA2+NOTO+CHRD+

|  | NotoPs | total_cells | Percentage NotoPs |
|---|---|---|---|
| **no delay** | 5 | 3422 | 0.15 |
| **24h** | 615 | 2193 | 28.04 |
| **48h** | 0 | 2177 | 0.00 |

**c**

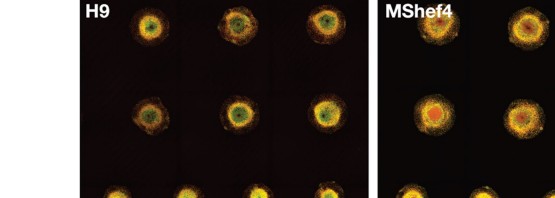

**Extended Data Fig. 6 | Delayed inhibition increases notochord progenitor cells.** a, Nucleus 3D quantification of high-resolution 24 h delay colonies (mean 4991 ± 907 cells per colony) showing the proportions of cell types obtained by immunofluorescence. 44 ± 8% of cells acquired a TBXT + FOXA2+ identity in this 24 h delay condition (compare without delay in Fig. 3d). b, Percentage of single cells co-expressing TBXT + FOXA2 + NOTO + CHRD+ in each of the different delayed inhibition conditions showing 0.15% of NotoP cells in the constant inhibition neuruloid are boosted to 28% (using this single-cell transcriptome signature) in the 24 h delay condition. c, micropatterned coverslip overview of 24 h delay condition across cell lines. Scale bar: 200 μm colony (c).

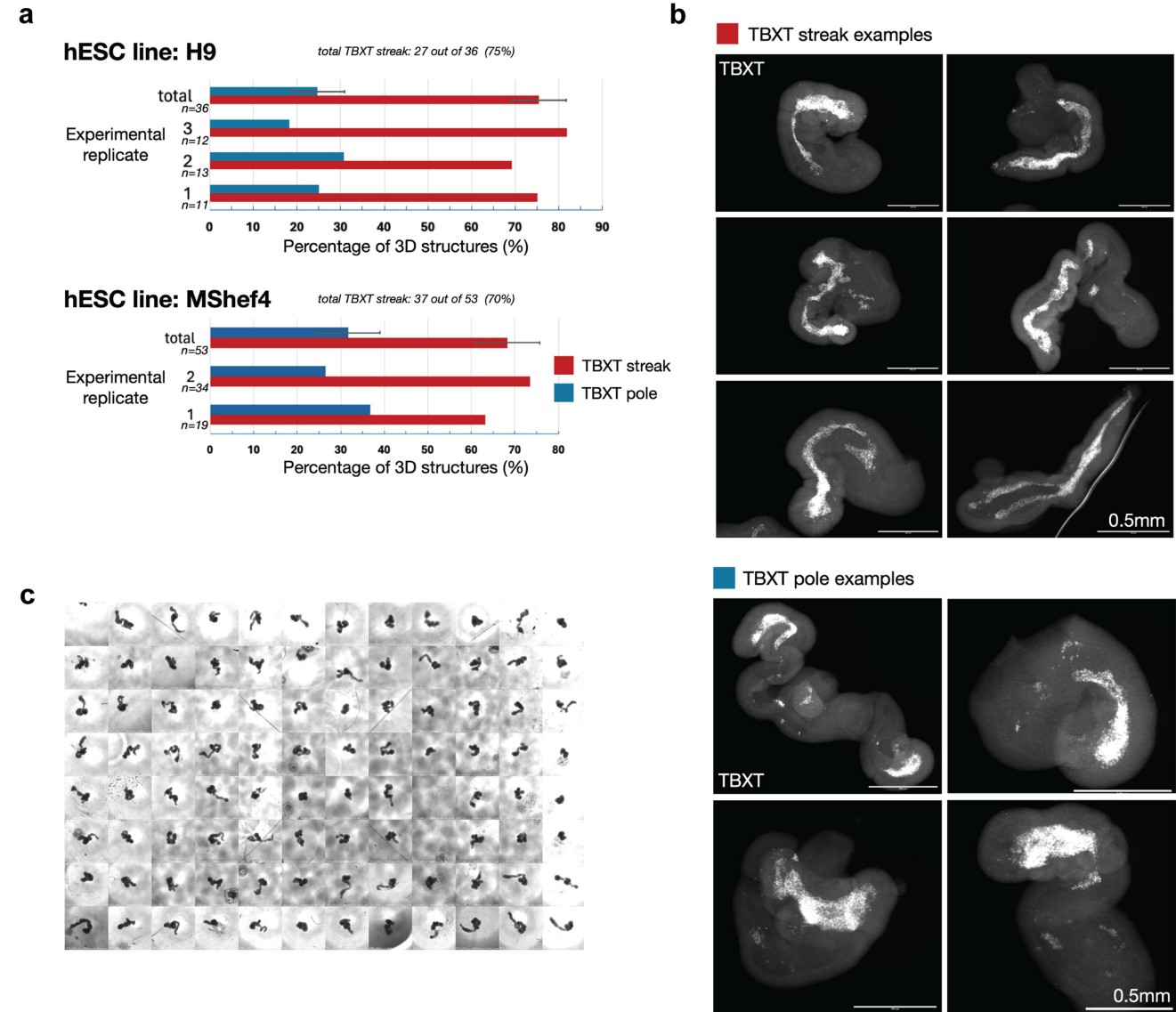

**Extended Data Fig. 7 | Reproducibility of trunk-organoids.** a-b, Quantification of TBXT immunofluorescence signal in trunk-organoids for H9 and MShef4 hESC lines. Details for experimental replicates are shown as well as examples of TBXT signal characterized as streak or pole. Overall 70-75% of structures display a streak-like organization (n = 89). c, Phase-contrast images of notoroids in a 96-well plate shows elongation present in most wells. Scale bars: 0.5 mm (b).

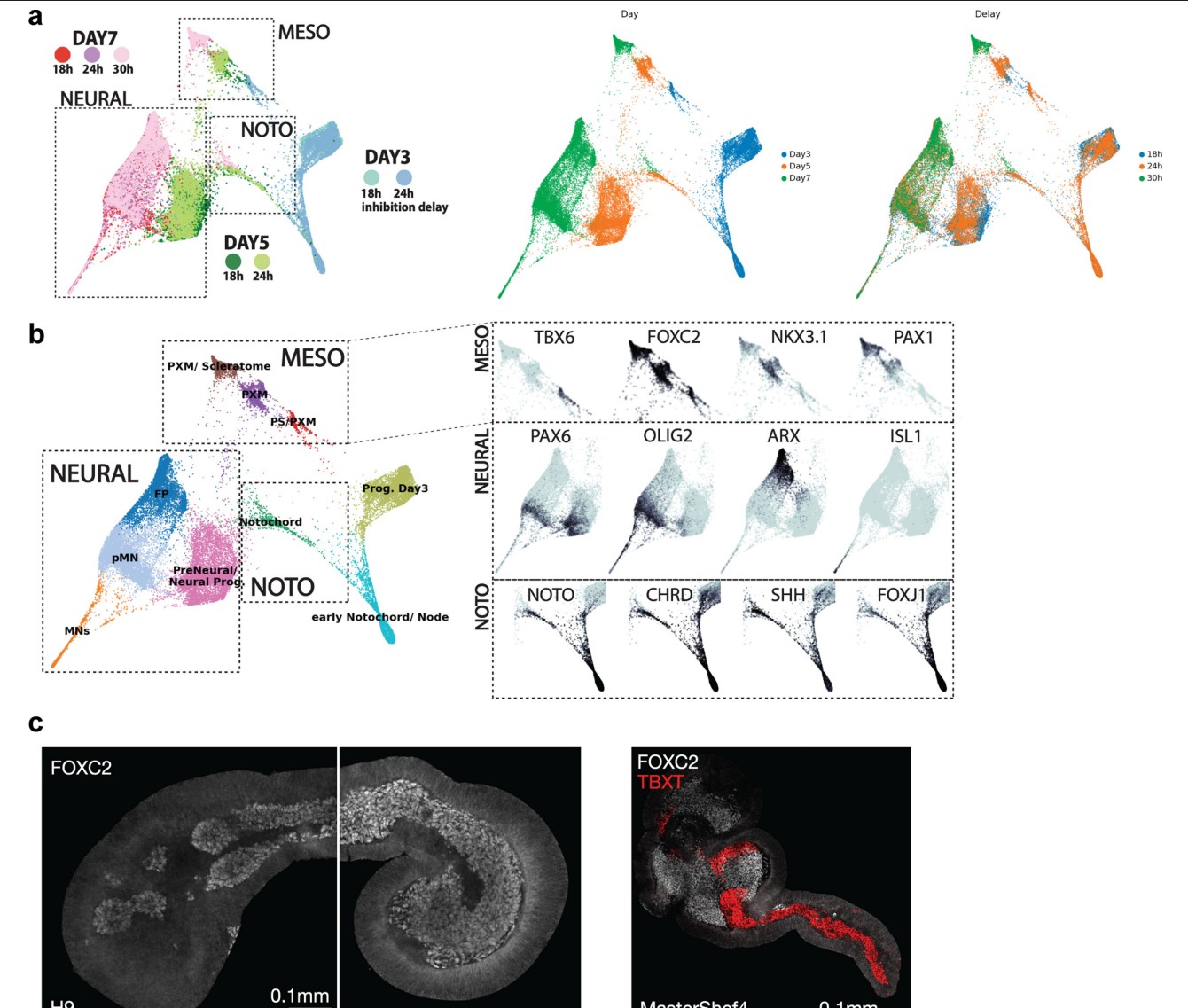

**Extended Data Fig. 8 | Single-cell RNA-seq of trunk-organoids.** a, Direct-force graph embedding layout with the integrated transcriptomic data of notoroids at day 3, 5 and 7 with delays of 18 h, 24 h (all days) and 30 h (only day 7). Populations of different end-point Days and the different Delay conditions are highlighted. b, Embedding detail showing expression of marker genes for different cell types. c. Single-z plane of the inside of a notoroid in H9 (*left*) or MShef4 (*right*) showing the presence of FOXC2+ somitic mesoderm in these 3D preparations. In scRNA-seq the majority of the FOXC2+ cell population expresses the Pax1 ventral marker. Scale bars: 0.1 mm (c).

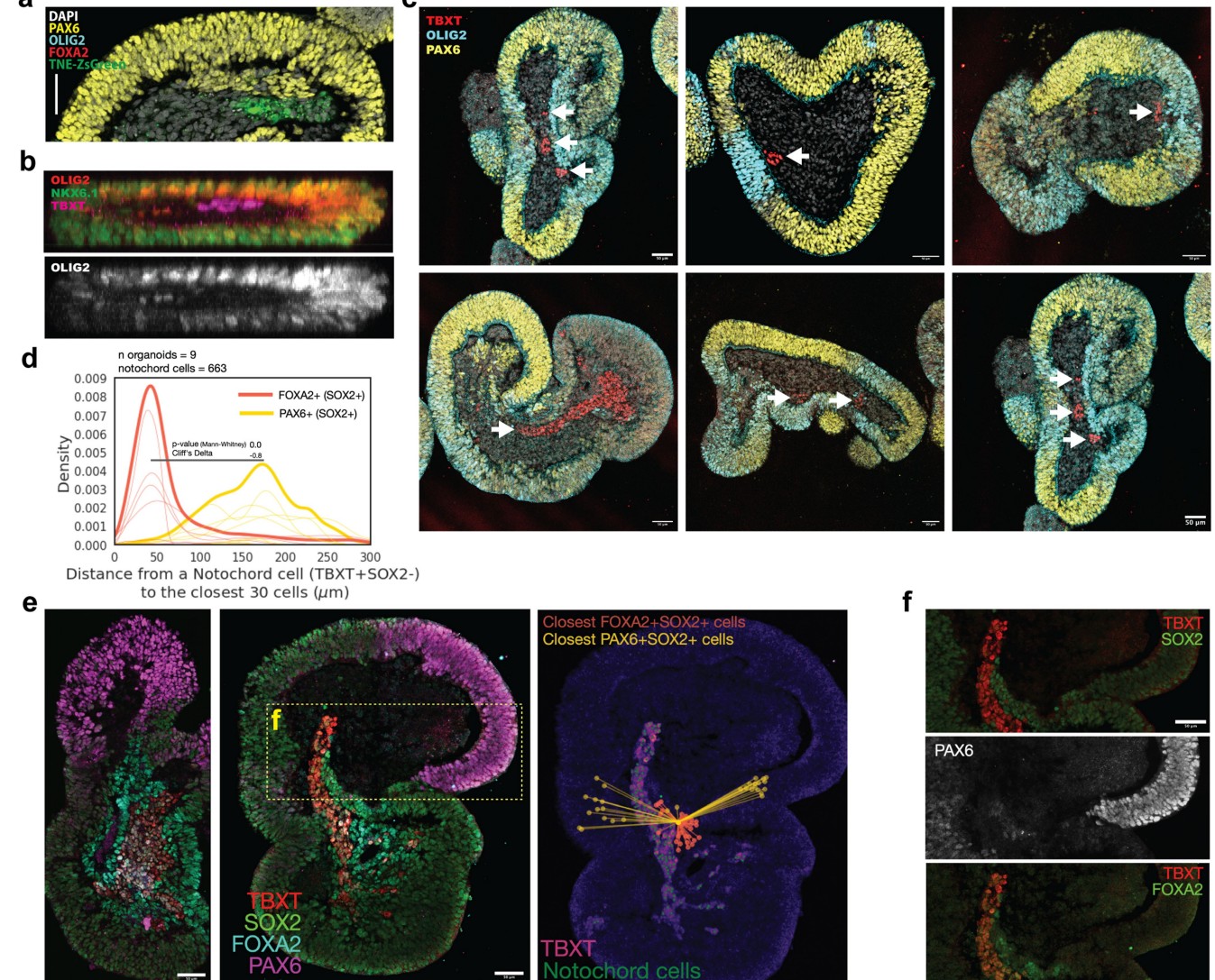

**Extended Data Fig. 9 | Ventral neural patterning in trunk-organoids/ notoroids. a,** Cryosection of H9 notoroid treated with vismodegib showing ventral neural patterning associated with notochord is abolished. **b,** Optical confocal section of H9 notoroid stained with the ventral neural markers NKX6.1 and OLIG2. OLIG2 expression appears polarized toward the TBXT+ notochord streak. **c,** Cryosections of MShef4 notoroids from three independent experiments (per row, n = 2) stained for TBXT, PAX6 and OLIG2, the ventral motor neuron progenitor maker, showing association of notochord and ventral patterning. **d-f,** Analyses of the proximity of notochord and neural cell types. **d,** Density plot showing distances of FOXA2 + SOX2+ are significantly closer to notochord cells (n = 663, n organoids=9) than PAX6 + SOX2+ cells. The thinner, overlayed density curves for each individual trunk-organoid are scaled by the number of pairwise distances contributing to the overall density plot. **e,** Cryo-sections of notoroids stained for neural markers SOX2, PAX6, the notochord marker TBXT and FOXA2 (both neural floor plate and notochord). Example of cryo-section segmentation and notochord cells selection (TBXT + SOX2-). Example of the 30 closest PAX6 + SOX2+ and FOXA2 + SOX2+ cells from an example notochord cell picked. **f,** zoomed inset of cryosection in panel e showing different marker combinations. Scale bars: 50 μm (a,c,e,f).

**a**

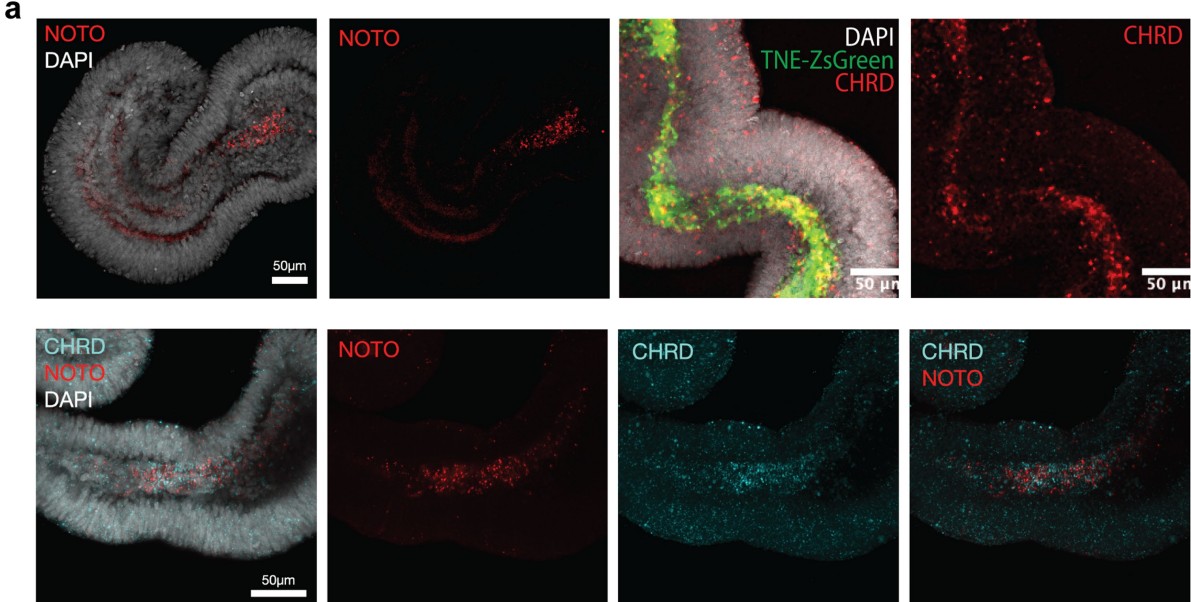

**b**

### Volumetric measurements (example Supplementary Video 7)

| Notoroid | total 3D length (um) | largest uninterrupted ZsGreen signal (% 3D length) |
|---|---|---|
| 1 | 1876 | 29.38 |
| 2 | 2525 | 44.04 |
| 3 | 2580 | 34.72 |
| 4 | 2327 | 39.69 |

**c**

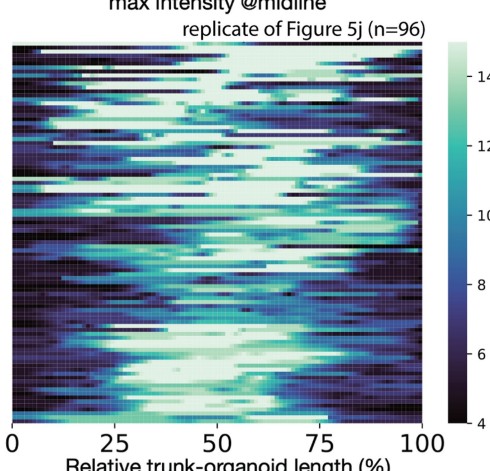

TBXT notochord-enhancer (TNE) ZsGreen max intensity @midline
replicate of Figure 5j (n=96)

**Extended Data Fig. 10 | TNE-ZsGreen H9 reporter line.** a, Single z-plane HCR stains of day7 notoroids for NOTO, CHRD and TNE-ZsGreen. b, Length and uninterrupted GFP signal measures in 3D (Imaris) based on lightsheet microscopy images (n = 4). c, independent replicate of Fig. 5j showing the distribution of ZsGreen signal along the midline of trunk-organoids. In the two replicates we detect an uninterrupted high signal for 30 ± 18% (Fig. 5j, n = 96) and 43 ± 19% (Extended Data Fig. 10c, n = 96) of the estimated length. Scale bars: 50 µm (a).

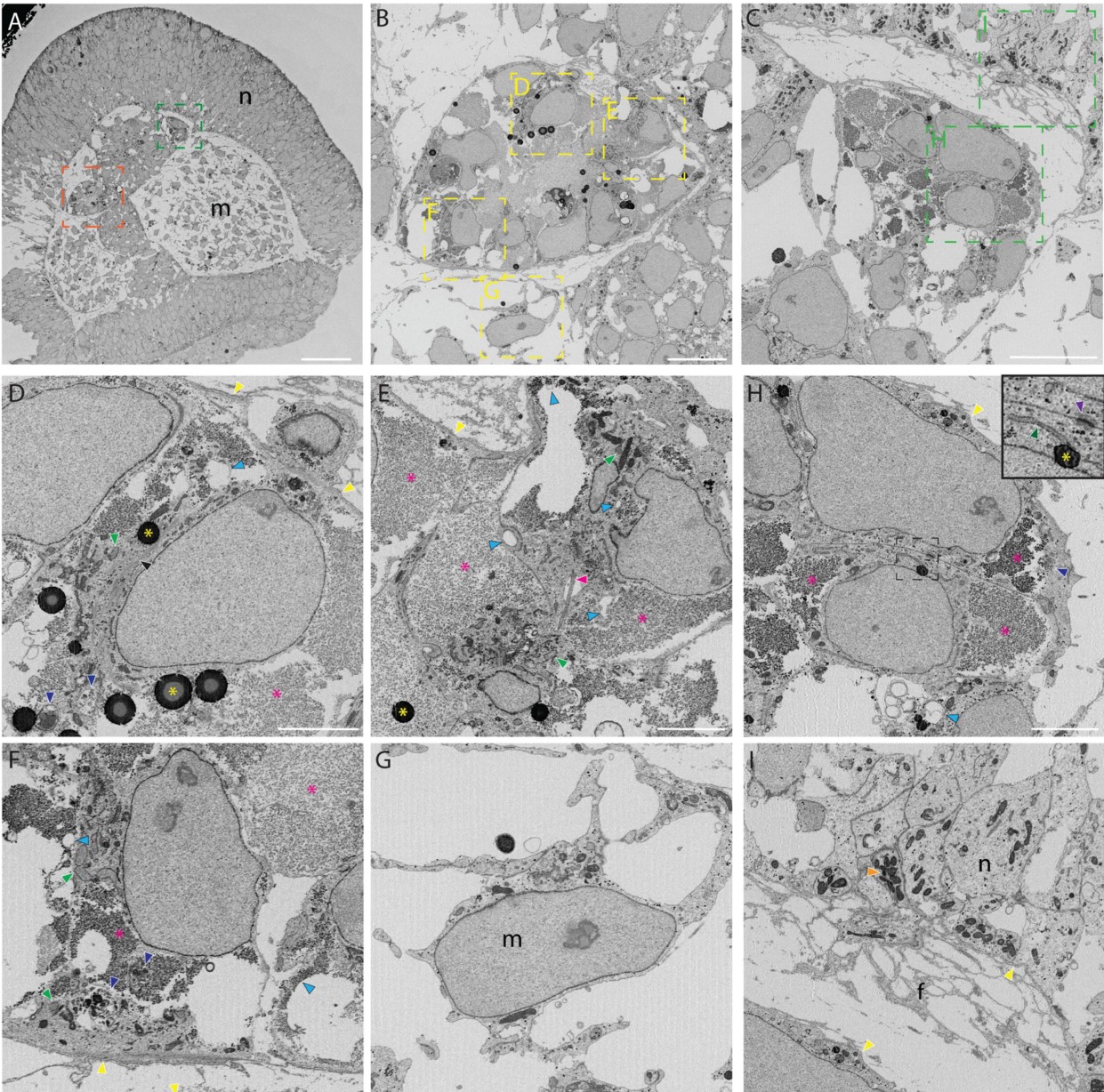

**Extended Data Fig. 11 | Volume EM of notochord in notoroids.** a, Low resolution SBF SEM image of a notoroid oriented to show transverse sections of the notochord. Neuroepithelium (**n**) and mesenchymal cells (**m**) are shown. Orange (b) and green (c) coloured boxes show regions of interest with putative notochord cells. b-c) High resolution SBF SEM images of notochord. Regions of interest shown in panels d-i are denoted with colour boxes. d-i) Cropped and enlarged SBF SEM images showing representative features of notochord in notoroids (d-f), mesenchymal cells (g) and neuroepithelial cells near the notochord (i). Notochord cells have abundant cytoplasmic glycogen (magenta stars), lipid droplets (yellow stars) and vacuoles (cyan arrows), are covered by a basal lamina (yellow arrows) and surrounded by a sheath of ECM fibres (**f**). The cells are closely packed, connected by desmosomes (purple arrow in h) and often ciliated (pink arrow in e). Dilated ER (green arrows) and endolysosomal structures (blue arrows) are frequent. In some cells, intermediate filaments (black arrow in d) and rough ER (dark green arrow in h) were observed. Mesenchymal cells are star-shaped with long protrusions in contact with neighbouring cells. Neuroepithelial cells in proximity to the notochord have some vacuoles and show an accumulation of mitochondria (orange arrow) towards the basal lamina. Scale bars: 50 μm (a), 10 μm (b-c), 3 μm (d-i).

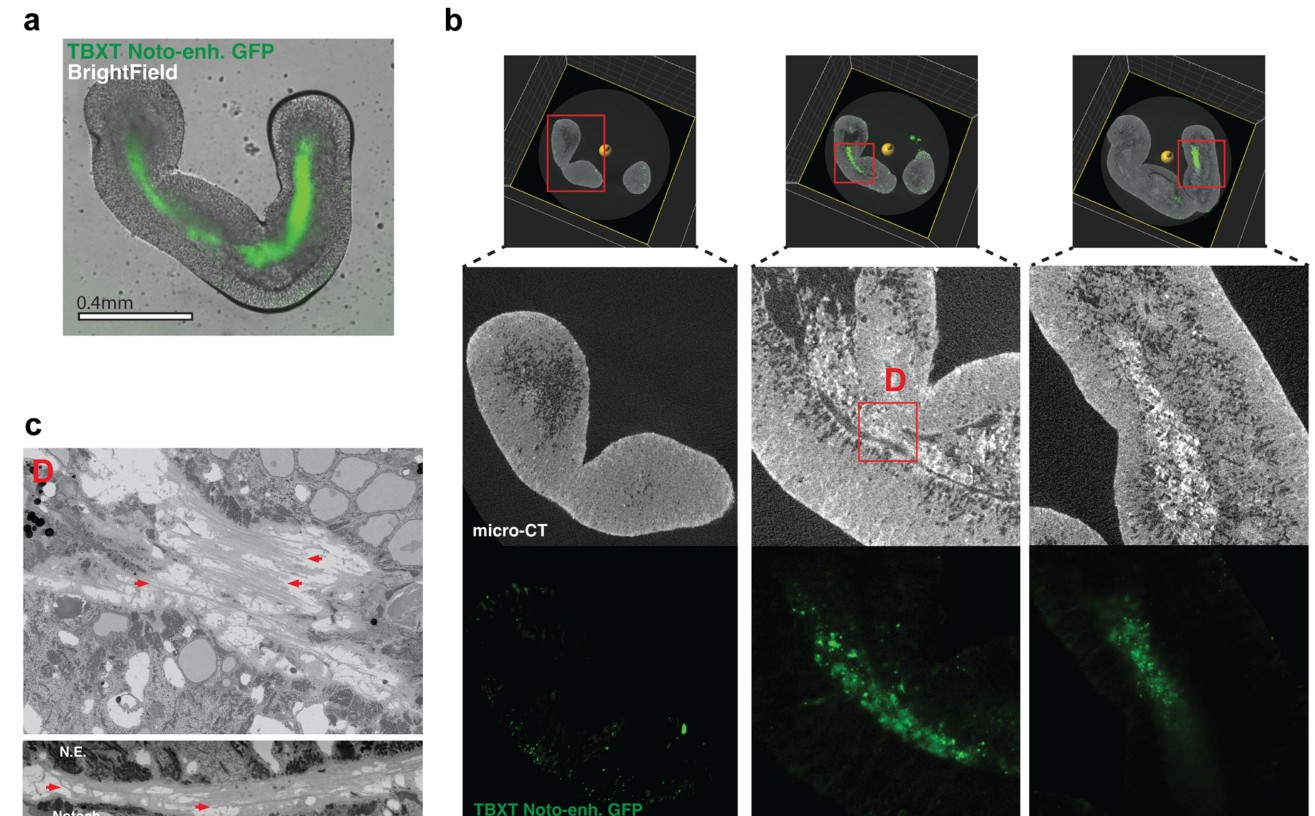

**Extended Data Fig. 12 | Correlative Light and Electron Microscopy (CLEM) of notoroids.** a, Brightfield image of notoroid used to perform Correlative Light and Electron Microscopy (CLEM). b, Micro-CT and TNE-ZsGreen overview of notoroid; single z-planes of notoroid used in CLEM showing fluorescence of the TNE-ZsGreen reporter at the expected locations. c, Zoom of red box D in b) marking the beginning of the notochord population and the fibers associated with it shown as a maximum intensity projection of the SBF SEM data and a orthogonal reconstruction view below.

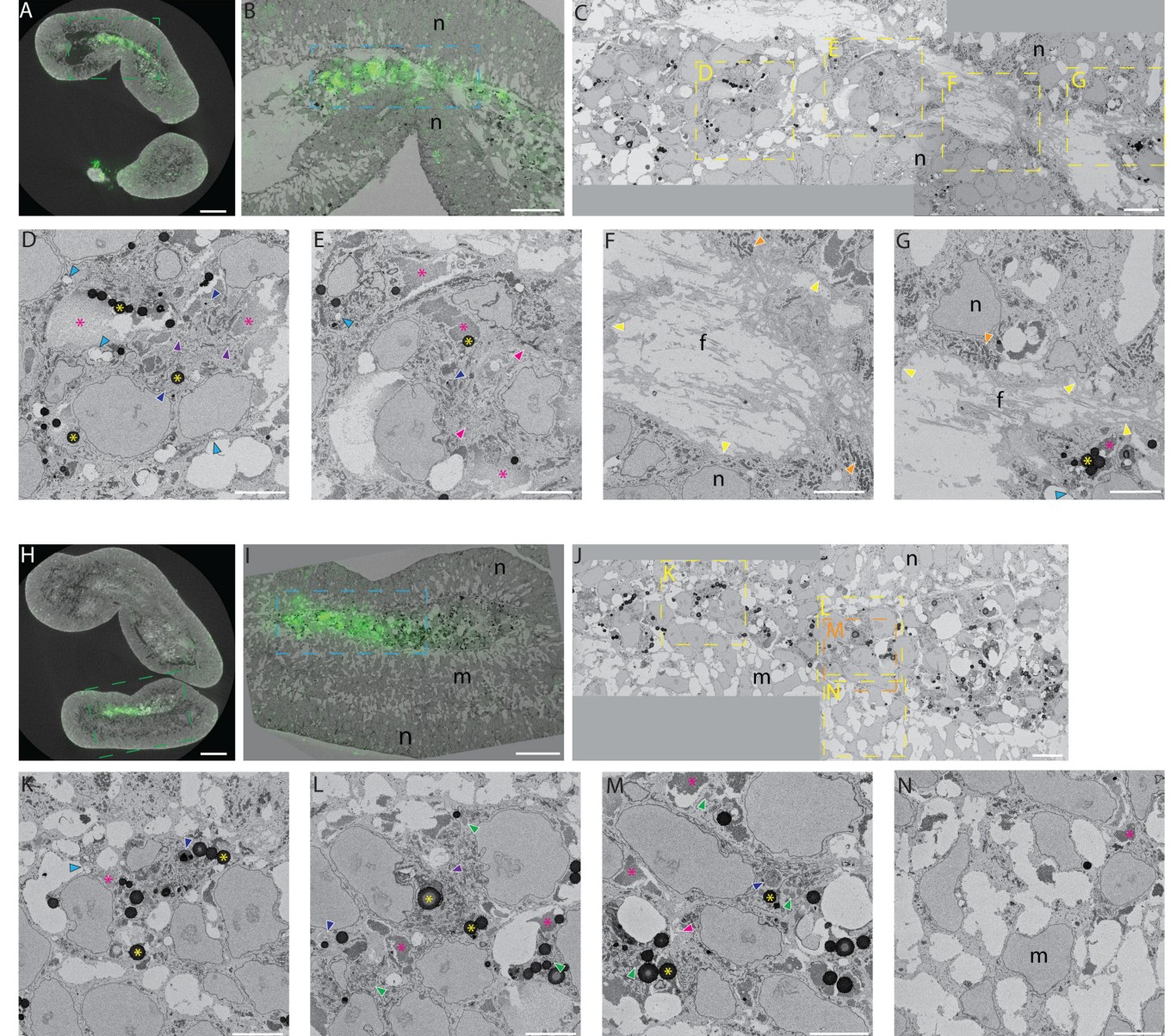

**Extended Data Fig. 13 | Volume CLEM of regions of interest (ROIs) of TNE-ZsGreen+ cells confirms the characteristics of notochord cells.** a, h) Micro-CT and ZsGreen overlay images of trunk-organoid oriented to show longitudinal section of notochord. Green box shows the localization of b and i respectively. b, i) Low resolution SBF SEM and ZsGreen overlay images of ROI of notochord. Cyan box shows the localization of c and j respectively. Neuroepithelium (n) and mesenchymal cells (m) are shown. c, j) High resolution SBF SEM images of TNE-ZsGreen+ notochord. Regions of interest shown in panels below are denoted with colour boxes. d-g, k-n) Cropped and enlarged SBF SEM images showing representative features of each region of ZsGreen notochord, mesenchymal cells (**m**) and neuroepithelial cells (**n**). Notochord cells have abundant cytoplasmic glycogen (magenta stars), lipid droplets (yellow stars) and vacuoles (cyan arrows), are covered by a basal lamina (yellow arrows) and surrounded by a sheath of ECM fibres (**f** in panels f and g). The cells are closely packed, connected by desmosomes (purple arrows) and often ciliated, with many microvilli (pink arrows). Dilated ER (green arrows) and endolysosomal structures (blue arrows) are frequent. Mesenchymal cells (**m** in panels i, j and n) are star-shaped with long protrusions in contact with neighbouring cells. Neuroepithelial cells (**n** in panels c, f, j and g) in proximity to the notochord are vacuolated and show an accumulation of mitochondria (orange arrow) towards the basal lamina. Panel m shows an image from the area shown in j, but in another slice (58 slices before j). Scale bars: 100 μm (a, h), 50 μm (b, i), 10 μm (c, j), 5 μm (other panels).

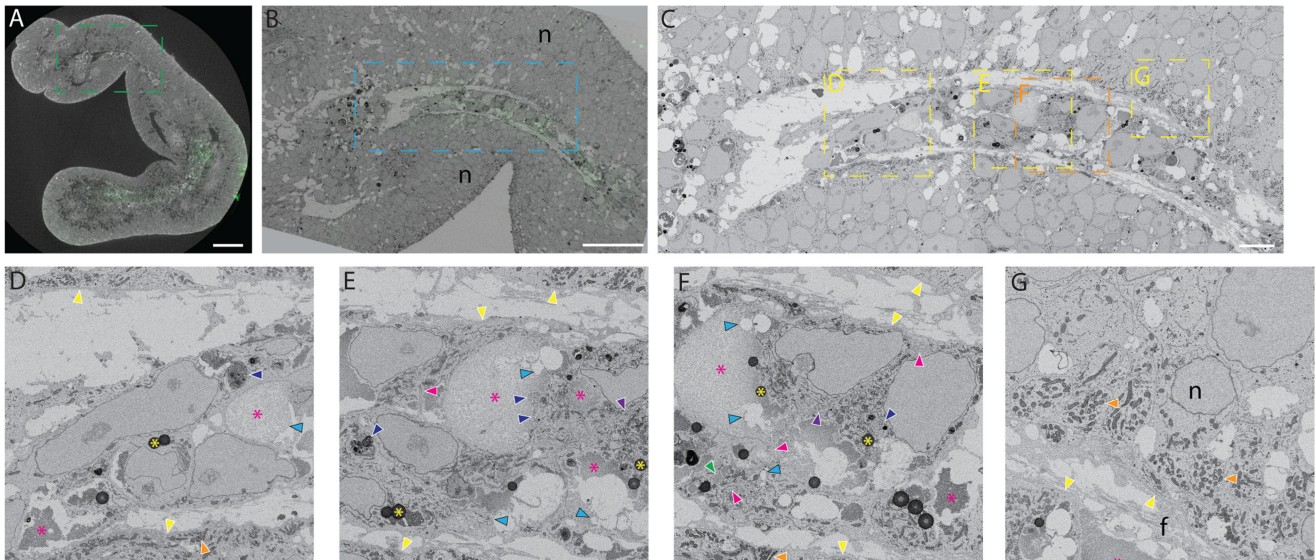

**Extended Data Fig. 14 | Volume CLEM of regions of interest (ROIs) of TNE-ZsGreen+ cells confirms the characteristics of notochord cells.** a) Micro-CT and ZsGreen overlay image of trunk-organoid oriented to show longitudinal section of notochord. Green box shows the localization of b. b) Low resolution SBF SEM and ZsGreen overlay image of ROI of notochord. Cyan box shows the localization of c. Neuroepithelium (n) shown. c) High resolution SBF SEM images of TNE-ZsGreen+ notochord. Regions of interest shown in panels below are denoted with colour boxes. d-g) Cropped and enlarged SBF SEM images showing representative features of each region of ZsGreen notochord, and neuroepithelial cells (n). Notochord cells annotation as in Extended Data Fig. 13: cytoplasmic glycogen (magenta stars), desmosomes (purple arrows), microvilli (pink arrows), dilated ER (green arrows), endolysosomal structures (blue arrows), lipid droplets (yellow stars), vacuoles (cyan arrows), basal lamina (yellow arrows) and sheath of ECM fibres (f in panel g) are frequent. Neuroepithelial cells (n in panel b and g) in proximity to the notochord are vacuolated and show an accumulation of mitochondria (orange arrows) towards the basal lamina. Panel f shows image of area shown in c but in other z-slices (31 slices after c). Scale bars: 100 μm (a), 50 μm (b), 10 μm (c), 5 μm (other panels).

# Reporting Summary

## Statistics

For all statistical analyses, confirm that the following items are present in the figure legend, table legend, main text, or Methods section.

| n/a | Confirmed | |
|---|---|---|
| ☐ | ☒ | The exact sample size (*n*) for each experimental group/condition, given as a discrete number and unit of measurement |
| ☐ | ☒ | A statement on whether measurements were taken from distinct samples or whether the same sample was measured repeatedly |
| ☐ | ☒ | The statistical test(s) used AND whether they are one- or two-sided *Only common tests should be described solely by name; describe more complex techniques in the Methods section.* |
| ☒ | ☐ | A description of all covariates tested |
| ☒ | ☐ | A description of any assumptions or corrections, such as tests of normality and adjustment for multiple comparisons |
| ☐ | ☒ | A full description of the statistical parameters including central tendency (e.g. means) or other basic estimates (e.g. regression coefficient) AND variation (e.g. standard deviation) or associated estimates of uncertainty (e.g. confidence intervals) |
| ☐ | ☒ | For null hypothesis testing, the test statistic (e.g. *F*, *t*, *r*) with confidence intervals, effect sizes, degrees of freedom and *P* value noted *Give P values as exact values whenever suitable.* |
| ☒ | ☐ | For Bayesian analysis, information on the choice of priors and Markov chain Monte Carlo settings |
| ☒ | ☐ | For hierarchical and complex designs, identification of the appropriate level for tests and full reporting of outcomes |
| ☐ | ☒ | Estimates of effect sizes (e.g. Cohen's *d*, Pearson's *r*), indicating how they were calculated |

*Our web collection on statistics for biologists contains articles on many of the points above.*

## Software and code

Policy information about availability of computer code

| Data collection | RT-PCR data was acquired using Applied Biosystems QuantStudio software. Immunofluorescence images were acquired using Zeiss Zen (v3.1) or Leica SP8 software. |
|---|---|
| Data analysis | Fiji v2.14 was used for image analyses, display and formatting. The custom-made pipelines to analyze single-cell data and for image quantifications were deposited on github. The scripts to analyse single-cell transcriptomic data are available https://github.com/tiagu/trunk_dev_scRNAseq. The Nucleus segmentation pipeline is available https://github.com/tiagu/Nucleus and the respective PyTorch models have been deposited in Zenodo (https://zenodo.org/records/11388472). The pipeline to analyse 3D cultures is available https://github.com/tiagu/gastrunet (DOI 10.5281/zenodo.12684779). Imaris v9.5.1 was used for volumetric measurements and lightsheet movie creation. The following packages were used for analyses: CellRanger (v4.0.0), Scanpy (v1.7.0), Scrublet (v0.2.2), harmonypy (v0.0.10), SAMtools (v1.9) dropEst (v0.8.6). |

For manuscripts utilizing custom algorithms or software that are central to the research but not yet described in published literature, software must be made available to editors and reviewers. We strongly encourage code deposition in a community repository (e.g. GitHub). See the Nature Portfolio guidelines for submitting code & software for further information.

# Data

Policy information about availability of data

All manuscripts must include a data availability statement. This statement should provide the following information, where applicable:
- Accession codes, unique identifiers, or web links for publicly available datasets
- A description of any restrictions on data availability
- For clinical datasets or third party data, please ensure that the statement adheres to our policy

Single-cell 3' mRNA sequencing data (10X Genomics) have been deposited under accession numbers GSE223189, GSE224404 and GSE255338 for chick trunk, human micropatterns and 3D notoroids respectively. Human data were mapped to the GRCh38-3.0.0 reference genome.
The following data series were re-analysed GSE122187 (mouse transcriptomics; mm10) and GSE193007 (cynomolgus monkey).
EM data is deposited in the Electron Microscopy Public Image Archive with the EMPIAR IDs: 12161,12162, 12163 and 12164.

# Human research participants

Policy information about studies involving human research participants and Sex and Gender in Research.

| | |
|---|---|
| Reporting on sex and gender | NA |
| Population characteristics | NA |
| Recruitment | NA |
| Ethics oversight | NA |

Note that full information on the approval of the study protocol must also be provided in the manuscript.

# Field-specific reporting

Please select the one below that is the best fit for your research. If you are not sure, read the appropriate sections before making your selection.

☒ Life sciences    ☐ Behavioural & social sciences    ☐ Ecological, evolutionary & environmental sciences

For a reference copy of the document with all sections, see [nature.com/documents/nr-reporting-summary-flat.pdf](http://nature.com/documents/nr-reporting-summary-flat.pdf)

# Life sciences study design

All studies must disclose on these points even when the disclosure is negative.

| | |
|---|---|
| Sample size | Sample size and depth of single-cell experiments were based on previous experience gained during the studies of Delile et al. 2019 and other similar studies. For qRT-PCR assays all samples collected over the course of exploratory and validatory stages of the study were aggregated for statistical analysis. For the remaining experiments no statistical test was performed to predetermine sample size. |
| Data exclusions | Melt-curve analysis and extreme cT values were used as a basis for exclusion outliers of qPCR technical replicates prior to statistical analysis of data. In micropatterned culture, pattens that were merged or not well isolated from their surroundings neighbors were excluded. |
| Replication | Reported results were repeated and confirmed in at least three independent experiments, except for single-cell RNA sequencing of embryos which was performed at closely-timed stages of development. For single-cell experiments of micropatterns and trunk-organoids, each experiment was reproduced and performed with multiple biological and/or technical replicates as stated in the Methods section. |
| Randomization | No particular randomization method was used in this work. For trunk organoid experiments, either the entire plate per experiment per condition was assayed or a given proportion of the plate was arbitrarily picked to be assayed (minimum n=8). |
| Blinding | Blinding was not performed in this study as the experimental setup and observation involves direct handling of the samples, e.g. application of treatments or cell counting. |

# Reporting for specific materials, systems and methods

We require information from authors about some types of materials, experimental systems and methods used in many studies. Here, indicate whether each material, system or method listed is relevant to your study. If you are not sure if a list item applies to your research, read the appropriate section before selecting a response.

## Materials & experimental systems

| n/a | Involved in the study |
|-----|----------------------|
| ☐ | ☒ Antibodies |
| ☐ | ☒ Eukaryotic cell lines |
| ☒ | ☐ Palaeontology and archaeology |
| ☐ | ☒ Animals and other organisms |
| ☒ | ☐ Clinical data |
| ☐ | ☒ Dual use research of concern |

## Methods

| n/a | Involved in the study |
|-----|----------------------|
| ☒ | ☐ ChIP-seq |
| ☒ | ☐ Flow cytometry |
| ☒ | ☐ MRI-based neuroimaging |

# Antibodies

| Antibodies used | The primary antibodies used in this study are detailed below:
mouse SOX2-AF488 BDBiosciences #561593 1:100
goat SOX2 R&D AF2018 1:500
rabbit TBXT D2Z3J #81694 1:500
goat TBXT R&D AF2085 1:1000
goat TBX6 R&D AF4744 1:500
TJP1 (ZO-1) Thermofisher #61-7300 1:200
goat SOX1 R&D AF3369 1:1000
rabbit FOXA2 Seven Hills WRAB-1200 1:2000
rabbit Phospho-p44/42 MAPK (Erk1/2) Cell Signalling 9101S 1:200
mouse total YAP Santa Cruz  63.7 (sc-101199) 1:100
rabbit active YAP Abcam EPR19812 (ab205270) 1:100
goat SNAIL R&D AF3639 1:500
rabbit LEF1 Cell Signalling #2230 1:200
goat SOX17 R&D AF1924 10ug/ml
goat HAND1 R&D AF3168-SP 1:200
mouse SHH DSHB 5E1 1:500
goat OLIG2 R&D AF2418 1:1000
sheep FOXC2 R&D AF5044 1:500
mouse NKX6.1 DSHB F55A10 1:100

Secondary antibodies (1:1000):
AF488 Donkey anti-mouse Invitrogen Cat#A21202
AF488 Donkey anti-goat Invitrogen Cat#A11055
AF488 Donkey anti-rabbit Invitrogen Cat#A21206
NL557 Donkey anti-Goat RnDsystems Cat#NL001
NL557 Donkey anti-Rabbit RnDsystems Cat#NL004
NL557 Donkey anti-Mouse RnDsystems Cat#NL007
AF594 Donkey anti-rabbit Invitrogen Cat#A32754
AF647 Donkey anti-mouse  Invitrogen Cat#A31571
AF647 Donkey anti-goat  Invitrogen Cat#A21447
AF647 Donkey anti-rabbit  Invitrogen Cat#A31573
AF647 Donkey anti-sheep  Invitrogen Cat#A21448
WGA iFluor™ 488 STEMCELL Technologies Cat#100-0816
DAPI Cell Signaling Technology Cat#4083S |
|---|---|
| Validation | All antibodies are commercial and have been validated by the companies from which they were purchased from in addition to being highly referenced in the field across several laboratories. Please see a list below of the main antibodies employed in this study. Furthermore, antibodies were tested for possible unspecific signals at unrelated embryonic stages or in vitro control conditions. Only antibodies with a reproducible signal were used.

goat SOX2 R&D AF2018; IF validation in ADLF1 and FAB2 Stem Cell Lines; 352 references.
rabbit TBXT D2Z3J #81694; IF validation in MUG-CHOR-1 and MCF 10A cell lines; 32 references.
goat TBXT R&D AF2085; IF validation in BG01V and differentiated human stem cells as well as mouse notochord; 240 references.
goat TBX6 R&D AF4744; IF validation in Mouse Mesoderm and JOY6 iPS cells ; 19 references.
goat SOX1 R&D AF3369; IF validation in Human Developing Brain, differentiated BG01V and NTera-2 cell lines; 127 references.
rabbit Phospho-p44/42 MAPK (Erk1/2) Cell Signalling 9101S; IF validation in  ; 32 references.
rabbit active YAP Abcam EPR19812 (ab205270); IF validation in HUVEC and 293A ; 85 references.
goat OLIG2 R&D AF2418; IF validation in human brain and oligodendrocytes in Rat Cortical Stem Cells ; 191 references. |

# Eukaryotic cell lines

Policy information about cell lines and Sex and Gender in Research

| | |
|---|---|
| Cell line source(s) | Human ESCs from the H9 line were obtained directly from WiCell and MasterShef4 from UKSCB. HEK293T cell line was obtained and authenticated by HPA cultures. |
| Authentication | WiCell database. Karyotyping was performed to verify genome integrity. |
| Mycoplasma contamination | All cell lines used in this study were routinely (every 3months) screened for Mycoplasma spp. and tested negative. |
| Commonly misidentified lines (See ICLAC register) | No lines from this register were used in the study. |

# Animals and other research organisms

Policy information about studies involving animals; ARRIVE guidelines recommended for reporting animal research, and Sex and Gender in Research

| | |
|---|---|
| Laboratory animals | Fertilized hens' eggs were obtained from Henry Stewart & Co. Ltd. |
| Wild animals | NA |
| Reporting on sex | Sex information in the transcriptomic data of chicken embryos was addressed by excluding genes in the W chromosome. |
| Field-collected samples | NA |
| Ethics oversight | Chicken embryos were incubated to less than two thirds of their gestation therefore do not fall under the UK's Animals (Scientific Procedures) Act 1986. |

Note that full information on the approval of the study protocol must also be provided in the manuscript.

# Dual use research of concern

Policy information about dual use research of concern

## Hazards

Could the accidental, deliberate or reckless misuse of agents or technologies generated in the work, or the application of information presented in the manuscript, pose a threat to:

No | Yes
- ☒ ☐ Public health
- ☒ ☐ National security
- ☒ ☐ Crops and/or livestock
- ☒ ☐ Ecosystems
- ☒ ☐ Any other significant area

## Experiments of concern

Does the work involve any of these experiments of concern:

No | Yes
- ☒ ☐ Demonstrate how to render a vaccine ineffective
- ☒ ☐ Confer resistance to therapeutically useful antibiotics or antiviral agents
- ☒ ☐ Enhance the virulence of a pathogen or render a nonpathogen virulent
- ☒ ☐ Increase transmissibility of a pathogen
- ☒ ☐ Alter the host range of a pathogen
- ☒ ☐ Enable evasion of diagnostic/detection modalities
- ☒ ☐ Enable the weaponization of a biological agent or toxin
- ☒ ☐ Any other potentially harmful combination of experiments and agents

