## [Peer Review File · Nature]

Manuscript Title: Timely TGF- β signalling inhibition induces notochord during trunk formation

Reviewer Comments & Author Rebuttals

Reviewer Reports on the Initial Version:

Referees' comments:

Referee #1 (Remarks to the Author):

Rito et al. investigate trunk tissue formation using a combination of single-cell RNA-sequencing, spatial mapping of transcripts, in vitro models of mammalian trunk development, and chemical modulations. The work is timely and interesting, and addresses important questions regarding the in vivo development, and in vitro reconstitution of the notochord, a defining feature of chordates. It is therefore unfortunate that, at least in its current form, the manuscript comprises a patchwork of multiple experiments/datasets, forced into a story. Whereas the individual parts are interesting, they lack rigour; many findings feel rather preliminary, and need extensive further experimentation to substantiate the, sometimes rather bold and unjustified, claims. Moreover, probably as a consequence of the patchwork structure of the paper, it remains unclear what the aim of the work is: e.g. if the authors aimed to generate more complete embryoids, they did not succeed given the apparent ventralisation of the 3D structures; if on the other hand they aimed to delineate the progenitor populations of the embryonic trunk, the experiments aiming to provide mechanistic insights lack rigour. I will specify my concerns below.

Major points:

1. The Abstract makes multiple claims that are not backed by the data. I will detail this below, but to summarise: i) the current data do not unequivocally support the claim that LATS1/2 repression of YAP activity induces TBXT and facilitates WNT; ii) forming a stripe of node/notochordal cells does not mean a notochord has been created; iii) "spatially patterned neural tissue formation" - there is no evidence to support this, the data appear to show that the notochord-like stripe induces full ventralisation of the posterior neural tissue.
2. The claim in the Abstract that "LATS1/2 repression of YAP activity, in conjunction with FGF-mediated MAPK activation, induced the transcription factor Bra/TBXT and facilitated WNT signalling" is a huge overstatement given the experimental evidence presented. In the Kastan et al. 2021 paper that identified TRULI, it is stated that their data demonstrate that "although TRULI inhibits Lats kinases, it might also interfere with the activity of other enzymes." Although the authors indeed show that TRULI increases active (nuclear) YAP in their 2D neuruloids, the current experiments, mainly IFC for selected targets in the presence and absence of TRULI, do not prove a (direct) link between LATS1/2 mediated repression of YAP activity on one side, and TBXT and WNT induction on the other side. This would

require more sophisticated experiments that provide orthogonal evidence (e.g. genetic perturbations, such as (conditional) YAP KO, YAP phospho mutants (S112 in mice, S127 in human) etc). Moreover, even the current data should be much better quantified, for example by correlating the YAP nuclear/cytoplasmic ratio to WNT activity, TBXT levels, etc.

3. The dissection of trunk progenitor populations and their progeny by comparative single cell RNA-Seq and spatial mapping (HCR) provides interesting data, but functional experiments are much needed. The authors e.g. claim that they demonstrate a “highly structured architecture in and around the caudal node”. Interesting, but what is the functional relevance of this? Big claims, that need functional evidence. Does perturbation of cytoskeletal components disrupt the architecture? What would be the functional consequence of this?

4. The authors should include readily available published human scRNA-seq data in their comparative analysis (Tyser et al. Nature 2021, Xu et al. NCB 2023). This is especially relevant since their ultimate aim is to induce node/notochord like cells in 2D and 3D neuruloids.

5. The authors develop a human model guided by the chick data “taking advantage of the evolutionary conservation of trunk cell populations”. However, they do not show that there is indeed conservation with human genetic programs.

6. The findings that the size of the neuruloids impacts patterning is interesting, in particular in light of the findings presented here: <https://doi.org/10.1101/2022.12.20.521167>. What is unclear to me is why the authors opted to continue with an intermediate (?) size, whereas the 200µm neuruloids display elongation and form a single pole of TBXT, as is observed in the embryo.

7. The authors claim that their 2D in vitro model gives rise to a subpopulation of node-like cells or notochord progenitors. The data do not provide very solid evidence for this claim. Although indeed the single cell RNA-Seq data show a very small cluster (maybe 10-20 cells out of thousands) with some enrichment of Noto, Chrd, Foxa2, Shh), as far as I can judge by presented data only 1, maybe 2, cells co-express these markers at high levels, as one would expect in node/notochordal-like cells. Moreover, Chrd is expressed in the other clusters as well. The quantifications based on IFC indicate that at least in one of the experiments (#1) there are no TBXT+/FOXA2+ cells detected. It should further be noted that the combined expression of TBXT and FOXA2 does not unequivocally demonstrate a node/notochordal identity, but could also indicate gut progenitors. More substantial evidence regarding the reproducible induction of this population, their localisation in the neuruloids, their functional relevance for the formation of the structures, and unequivocal demonstration of a node/notochord identity, is needed to substantiate the authors' claims.

8. Given the conceptual importance of the population of TBXT+FOXA2+CDX2+ cells identified by scRNA-seq in macaque and chick for the presented work, the authors should confirm this finding by spatial mapping of this population (and show co-expression of Noto, Chrd, Fgfs, Wnts etc) by HCR (at least in chick).

9. The authors claim that Figure 5b shows “a dramatic increase in endoderm cells, expressing both SOX17 and FOXA2.” However, colocalization is not shown, only separate stainings.

10. The induction of a notochord like stripe of cells in the 3D neuruloids is interesting, but needs a much better characterisation and quantification. The authors claim a 75% efficiency, but which criteria is this based on? How reproducible is it between experiments? How variable is this stripe in length, width etc. A detailed morphometric analysis of the structures and the notochord like stripe is absolutely needed.

11. The authors should provide a much better molecular characterisation of the “notochord like stripe” in their 3D structures. Given the amount of scRNA-seq experiments performed in the manuscript, it is surprising that no comparative analysis of the 3D structures with the embryo is provided. In addition, HCR should be performed for notochord markers such as Noto, Chrd, etc to ensure a notochord identity. The current combination of markers does not exclude the possibility that these represent gut-like midline progenitors secreting Shh, which would also “pattern” adjacent structures. The formation of notochord in 3D embryoids is a very strong claim, and requires equally strong evidence.

12. To provide further functional evidence that the cells have a true notochord identity, grafting experiments can be performed to show that the 2D/3D neuruloid cells can induce an ectopic floor plate, as shown in Placzek et al. Science 1990.

13. The notochord is not only an essential signalling centre, but also provides structural support to the embryo. In this regard, notochord cells have specific structural features (reviewed in e.g. Jurand, J Embryol Exp Morphol 1974). The authors should use EM to assess if the in vitro generated node/notochord-like cells also display these typical features (e.g. notochord vacuoles).

14. I am surprised about the strong claim that the notochord stripe generated in vitro induces patterning of the neural tube. The data presented by no means back such a bold claim. Based on the data, the 3D neural tissue in the structures appears to be ventralized in the presence of the notochord like stripe. This is a long stretch from obtaining a patterned neural tube, with both dorsal and ventral identities. The current structures appear to switch from one “in vitro issue”, i.e. dorsalization in the absence of a notochord, to another issue (full ventralisation). The authors should provide a detailed, quantitative analysis how the presence of a notochord-like stripe changes the “patterning” of the neural tissue. Orthogonal sections are needed.

15. In Figure S11e the authors show the presence of somitic tissue (FOXC2+). In what percentage of the structures do the authors obtain both posterior neural and somitic tissue? Do the structures with somitic tissue also develop notochord-like stripes? If so, does the somitic tissue also display “patterning” (or ventralization, see previous comment). Also, given the ability to form both somitic and neural tissue, do the 3D structures harbour NMPs? Do the authors indeed generate more complete structures by inducing a notochord like structure?

16. In case their final model indeed forms NMPs, posterior neural and somitic tissue, the authors should check if the “highly structured architecture” they claim to observe in vivo is recapitulated in the 3D in vitro system. This would increase the value of the system and, hence, that of the manuscript.

17. The authors state that all scripts are deposited at GitHub. However, for two of the provided links (scRNAseq and gastrunet) the code has not yet been deposited. The authors should do this immediately to ensure pipelines can be reviewed.

18. Many of the imaging data presented lack a scale bar (e.g. the entire Figure 4 & almost every supplemental figure). Please add.

19. The paper suffers from poor statistical description of the imaging data. What is e.g. the penetrance of observed phenotypes upon the described perturbations?

Minor points:

1. The authors repeatedly refer to the in vitro model as a “3D model of human gastrulation”. Gastruloids model the outcome of gastrulation, but not the gastrulation process per se. This should be clarified. Also, I would not refer to these models as “integrated”, since per ISSCR definition integrated models refer to models comprising both embryonic as well as extra-embryonic tissues.

2. When discussing single-cell RNA-seq data, the authors repeatedly make claims about cells expressing markers together, without showing data that the markers are actually expressed in the same cell (as opposed to being expressed in different cells of the same cluster).

3. The authors might want to include a second often used mouse atlas (Pijuan-Sala et al., Nature 2019)

4. “To construct a fate decision axis reflecting both time and space” - what do the authors mean?

5. It is unclear to me how the spatial mapping of the clustered HCR data was conducted. The Methods section does not clarify this.

6. The authors should discuss the differences and commonalities of their 2D micropatterned gastruloids with the 2D gastruloids developed in the work of Brivanlou, Warmflash, Heemskerk, Zandstra etc.

7. The authors state that the inward movement of cells causes the folding of the apical side towards the centre in the 2D neuruloids. I don't see how the provided data (IFC for some markers) shows this?

8. Figure S10a does not show expression of Chrd, as the authors mention.

Referee #2 (Remarks to the Author):

It might be wise for the authors to add a sentence or two to their manuscript noting the ethics and laboratory reviews that they undertook (if any) and perhaps noting recommendation 2.2.1A(d) in the ISSCR Guidelines, if (as I appeared to me) that recommendation explains why specialized ethics review was not sought/was not needed. I suggest there merely to preemptively address questions any reader might have about the ethics of this study, given that the study may be of interest to a variety of audiences.

Referee #3 (Remarks to the Author):

Previous attempts to generate in vitro models for the formation of mini-embryoids from human or mouse ESCs or iPSCs have generally led to embryoids containing many of the early tissues (resembling somites, lateral plate mesoderm, definitive endoderm, neural tissue, epidermis, blood etc.) but all existing models failed to generate notochord. This paper addresses this. It starts by analysing the transcriptome of single cells isolated from the caudal half or so (apparently including extraembryonic regions and more) of chick embryos at 4 closely spaced stages. Clustering methods are used to identify different cell types and their putative precursors based on gene expression, and the results then used to develop a protocol to generate axial mesoderm like tissue (notochord) from embryoids made from human ESCs by modifying a previously established method. Apart from critical timed manipulation of Wnt activity via YAP along with modulation of FGF, the protocol includes treatment with retinoic acid precursors, which is shown to lead to elongation of the organoids and importantly, dorsoventral patterning of the somite-like and neural-plate-like structures they contain. This is an important technical advance on the construction of these organoids and may also reveal important aspects about the specification of tissues that were hitherto absent from the embryo models.

In my opinion the main strength of this paper, as compared to many other -oid papers that have been appearing over the last few years, is that it is much more closely informed by direct, primary observations of real transcriptomes from real embryos, including taking into account real developmental time, and also verifying the expression of some of the markers by comparison to the patterns seen in embryos using HCR and immunostaining. Another major strength is that the paper follows a clear logical progression in experimental design. The elongated organoids shown in Fig. 5 g-i are quite impressive relative to what has been published before by people using this model. In principle it should be appropriate for Nature.

I have a few issues that in my opinion need attention especially to constrain some of the generalisations and conclusions to the actual results presented, as in many instances throughout the paper there are many assumptions and speculations that are then used as "facts". I think the paper would be much stronger scientifically if it distinguished predictions and generalisations from actual findings. Some of these issues plague a lot of the current literature but now that the field is quite mature, and given that

this group does have a good grounding in developmental biology, it is particularly important to present these things more transparently so that open questions are easier to identify for future study. In particular one problem is the assumption that it is possible to identify "precursors" of anything based on a static gene expression signature of clusters of single cells, based entirely on scRNAseq data and with no attempt to validate the conclusions with lineage tracing or other methods. This should really be avoided - at least specify (in every case) when something is a prediction. This includes NMPs - there is actually no clear evidence for single cells that either do give rise, or can give rise, to neural and mesodermal descendants at late stages - only for a region that contributes to these tissues, and that cells co-express some markers like TBXT and SOX2, but it is perfectly possible that this expression undergoes oscillations before cells settle for one or the other expression and we don't know whether all cells that coexpress these markers do/can give rise to both tissues. The same applies to all other assumptions about "precursors" (and in some cases tissue identity) throughout the paper - simple toning down of the claims will do, but the paper does need to keep open the possibility that these interpretations may turn out to be incorrect in due course.

Other more specific comments (some very minor, others less), presented more or less in the order in which they appear in the manuscript follow below. The comments are intended to help improve the manuscript and make it more rigorous (and reduce the "spin"):

In the introduction, "a structure known as the organiser or node" is misleading. The node (and/or its precursor cells) is only an organiser (ie can induce and pattern neural tissue and pattern adjacent mesoderm) only for a very short time before axial elongation even begins. The node as a structure persists for much longer and it is responsible for axial elongation, but loses its neural inducing properties very early. In anamniotes the equivalent structure is the embryonic shield (teleost fish) and the blastopore lip (amphibians).

p.3: the tissue isolated for scRNAseq from chick embryos is a huge piece of embryo. The diagrams in Fig 1 show essentially half the embryo. So only a tiny proportion of these cells are in any way relevant to this study. I don't understand why they did not dissect more finely to include only regions including and near the node and its descendant tissues which should be well within the ability of this group to do. This reduces the number of relevant cells sampled and could complicate the interpretation further because many of the "markers" are also expressed in unrelated tissues.

There are many statements like "TBX6 was expressed in the paraxial and presomitic mesodermal cells" and "Notochord cells expressed high TBXT, NOTO, SHH and CHR1" and more seriously (see above) "Crucially, the NMP cells, as in chick, exhibited the highest proportion of double-positive Sox2+T+ cells" and "transcriptome data indicated that these cells corresponded to the early notochord progenitors", etc. which sound as if these cells are identified by other means than the expression described whereas it is actually the other way around. These identities are assigned, or interpreted, based on the expression of the listed genes. This problem occurs throughout the manuscript.

p7: "Combining the single cell transcriptome data with in vivo mapping provides new insight into the

organisation of fate transitions driving axis elongation and the spatial dynamics of the signalling pathways controlling them." Here, the reference to in vivo mapping is misleading because it suggests that the precursor nature of the cells has been directly validated, when the reference is merely to in situ staining patterns of the genes.

p.9: the title refers to "posterior cell fate decisions" but the contents of this section are mainly about axial tissues. It is incorrect to consider the node and streak as "posterior fates". Even though they are located increasingly posteriorly during axial elongation, their descendant cells span almost the entire length of the body. So just "axial" should do for this title.

p. 10: "posterior neuruloids". Again these are not necessarily just posterior, but "neuruloids" is quite a good term. In fact the use of "gastruloids" and "gastrulation" in various parts of the paper is not strictly correct - what the study tries to generate is an in vitro model of neurulation, including the formation of axial (notochord) and a dorsoventrally patterned early neural plate which are the hallmarks of neurulation. "Neurulation" as a period begins when gastrulation ends, marked by the appearance of notochordal and neural tissues in front of the node (and the word gastrulation should not be a synonym for cell ingression or mesoderm formation because in many organisms the processes are uncoupled). Also, "mechanisms of trunk formation" seems a bit strong in the absence (at this point in the manuscript) of a description of cell morphologies/arrangements in notochord-like cells, or a clear description of elongation which is crucial for "trunk formation". This comes later in the manuscript and requires the retinoid precursor experiment. I could not find the movies in the submission but the description at this point only refers to the formation of a ring, not elongation.

p. 12: " To investigate how YAP activation impairs TBXT induction we used LEF1 expression as readout of WNT signalling". This does not make logical sense. Irrespective of whether it has been previously suggested by others that YAP and Wnt are linked (in fact the references indicate both agonistic and antagonistic effects), the experimental design as stated does not follow the question.

p 15: "To validate the assignment of cell identity, a classifier trained on chick and mouse cell clusters correctly placed the different neuruloid populations in our assigned cell fates" - this seems a circular argument. Using a AI or other predictor to validate another prediction is quite strange.

p 15: "... prolonged signalling generates endodermal ..." this is very strange because definitive endoderm is believed to be generated only quite early, ending at about the time when notochord starts to form. So what does this prolonged period of signalling correspond to in the embryo?

Two previous papers have addressed the compartmentalization of the node in terms of fates at single cell level: Selleck & Stern 1991 (<https://doi.org/10.1242/dev.112.2.615>) and Solovieva et al. 2022 (<https://doi.org/10.1073/pnas.2108935119>). Some of the statements here are supported by these two papers with direct cell lineage information so it would be useful to cite them. The latter paper also contains scRNAseq data of cells in different subregions of the chick node and at different stages (including CNTN2 expression at later stages) and it would be valuable to compare the chick (and human)

transcriptomes generated here with those described in individually isolated cells from different regions of the node.

There are also a few grammatical/typographical errors throughout the manuscript. For example "posterior" is an adjective not a noun in this context (as a noun it means "bum"!) so should always be followed by a noun like "posterior part", etc. (see abstract for example). "Evolutionarily conserved" (not evolutionary conserved - in the Introduction). Media is plural so it should be "inducing medium". p. 6 uses both "median pit" and "medial pit". In the embryological literature this is named "primitive pit" (rightly or wrongly) but just "pit" will do. There is no lateral pit. "caudal node" makes no sense. Just node (as there is no cranial node). "posterior axial fate decisions" is embryologically incorrect - just "axial" - these tissues span most of the length of the embryo. On p 12, the first sentence under the heading makes no sense: "is" seems out of place, and "their" is ambiguous.

In my opinion the Discussion is superfluous and could be deleted. Currently it is mainly another summary of the paper and does not add much, as well as reinforcing some of the extrapolations as if they were facts. A single concluding paragraph should be enough.

Referee #4 (Remarks to the Author):

In this paper, Rito et al investigate trunk tissue formation by using a combination of in vivo and in vitro models. First, they chart trunk formation in chick embryos by combining single-cell transcriptomic with HCR. This allows them to identify the signaling pathways that might lead to the formation of the trunk cell types. Based on these results, they developed a new and improved in vitro model of human trunk formation and characterize it to find the conditions under which specific cell types like notochord cells arise.

I find this study very elegant and interesting. The single-cell RNA-seq (scRNA-seq) experiments and the data analyses are overall well executed, but I have a few concerns and suggestions, as detailed below.

1. The authors use the scRNA-seq data to estimate the cell type composition and how it changes, for instance, across developmental stages in chick embryos. However, they should add a caveat in the text warning that scRNA-seq can provide biased estimations of cell type compositions (see, e.g., <https://genomebiology.biomedcentral.com/articles/10.1186/s13059-020-02048-6>), and perform a statistical test to check whether the differences in cell type composition are statistically significant or not (by using methods like <https://www.nature.com/articles/s41467-021-27150-6>).
2. After scRNA-seq data analysis, they use HCR to map the spatial patterns of a few selected genes. It is not explained, though, why those genes were selected and not others. Was the choice informed by scRNA-seq data analysis? If so, it would be useful to explain the criteria that were used to choose the genes.
3. It seems like some cell sub-populations (e.g., the two sub-populations of NOTO+CHRD+ notochord

cells described on page 4), were just identified "by eye" from 2D embeddings of the data. This is quite unreliable, as 2D embeddings can be affected by many artifacts. Instead, clustering algorithms should be used here.

4. Similarly, on page 15, the authors claim that there's a population of cells (TBXT+FOXA2+CDX2+) that "occupied a region of gene expression space between NMPs and mature notochord" and then refer to a 2D embedding of the data in Figure S10a. This claim should be verified by a more rigorous analysis in the high dimensional space of gene expression.

5. Some details that are crucial to ensure the reproducibility of the analysis are missing from the Methods section:

a. As a validation of the assignments of cell identity, they ran a random forest classifier trained on mouse and chick data (page 15), as they specify in the legend of Figure S10c-d. However, the values of some parameters are missing (e.g., number of trees?); moreover, did they use only 1:1 homologous genes?

b. On page 24: "counts were normalized to a target sum of 10000 excluding highly expressed genes". How many of the highly expressed genes were excluded?

c. About the clustering shown in Figure 2d, the authors say that they used hierarchical clustering with "K=5". Do they mean K-means clustering? what distance did they use?

6. It is very good that the authors created GitHub repositories to publish the scripts used for the analysis (these might also provide the answers to the above questions regarding the Methods section), but why are they still empty?

****Minor points****

- What do the authors mean by "fine-grained" Louvain clustering? it should either be clarified or removed

- The sentence on page 4 "SEMA3C/E , a semaphorin implicated in the control of cell morphology and motility" lacks a citation.

- Figure S10C has very low resolution and the x-axis labels are cut

Author Rebuttals to Initial Comments:

Referee #1 (Remarks to the Author):

Rito et al. investigate trunk tissue formation using a combination of single-cell RNA-sequencing, spatial mapping of transcripts, *in vitro* models of mammalian trunk development, and chemical modulations. The work is timely and interesting, and addresses important questions regarding the *in vivo* development, and *in vitro* reconstitution of the notochord, a defining feature of chordates. It is therefore unfortunate that, at least in its current form, the manuscript comprises a patchwork of multiple experiments/datasets, forced into a story. Whereas the individual parts are interesting, they lack rigour; many findings feel rather preliminary, and need extensive further experimentation to substantiate the, sometimes rather bold and unjustified, claims. Moreover, probably as a consequence of the patchwork structure of the paper, it remains unclear what the aim of the work is: e.g. if the authors aimed to generate more complete embryoids, they did not succeed given the apparent ventralisation of the 3D structures; if on the other hand they aimed to delineate the progenitor populations of the embryonic trunk, the experiments aiming to provide mechanistic insights lack rigour. I will specify my concerns below.

We thank the reviewer for carefully reading the paper and finding our work on *in vitro* generation of notochord timely and interesting. Below we address the reviewer's concerns point-by-point.

Major points:

1. The Abstract makes multiple claims that are not backed by the data. I will detail this below, but to summarise: i) the current data do not unequivocally support the claim that LATS1/2 repression of YAP activity induces TBXT and facilitates WNT; ii) forming a stripe of node/notochordal cells does not mean a notochord has been created; iii) "spatially patterned neural tissue formation" - there is no evidence to support this, the data appear to show that the notochord-like stripe induces full ventralisation of the posterior neural tissue.

RESPONSE:

We have clarified and strengthened the evidence supporting each of the conclusions in the abstract:

- (i) Nuclear YAP induced by TRULI (which functions via LATS1/2 repression) leads to a statistically significant reduction in LEF1, a common WNT readout (assessed over 3 biological experiments, i.e. cells of distinct vials and passage number), and has a marked, statistically significant reduction at 24h in the induction of TBXT. We have now added additional analyses and experimental evidence to support this conclusion.
- (ii) Current 3D models of the trunk have no notochordal cells and display at best a modest elongation, indeed for the human case the current models show very little morphological structure apart from Olmsted et al and axioids/ somitoids where a single population, somitic mesoderm, is obtained. The 3D model we describe not only generates notochordal cells that are consistently present, but we often find them in a rod-like arrangement, akin to a notochord. We have now added additional molecular and imaging data to support our conclusions.
- (iii) Consistent with the presence of neural and mesodermal tissue surrounding a functional notochord expressing SHH we see on-going ventralization of the tissues. By "Spatially patterned neural tissue" we meant the presence of multiple distinct neural progenitor domains associated with a progressive ventralisation of the ventral half of the neural tube. We have made this more explicit in the revision to avoid misinterpretation and added additional images to illustrate this point.

As detailed below, in the revised manuscript we strengthen each one of these claims and include further evidence supporting the conclusions on each one of these points.

2. The claim in the Abstract that “LATS1/2 repression of YAP activity, in conjunction with FGF-mediated MAPK activation, induced the transcription factor Bra/TBXT and facilitated WNT signalling” is a huge overstatement given the experimental evidence presented. In the Kastan et al. 2021 paper that identified TRULI, it is stated that their data demonstrate that “although TRULI inhibits Lats kinases, it might also interfere with the activity of other enzymes.” Although the authors indeed show that TRULI increases active (nuclear) YAP in their 2D neuruloids, the current experiments, mainly IFC for selected targets in the presence and absence of TRULI, do not prove a (direct) link between LATS1/2 mediated repression of YAP activity on one side, and TBXT and WNT induction on the other side. This would require more sophisticated experiments that provide orthogonal evidence (e.g. genetic perturbations, such as (conditional) YAP KO, YAP phospho mutants (S112 in mice, S127 in human) etc). Moreover, even the current data should be much better quantified, for example by correlating the YAP nuclear/cytoplasmic ratio to WNT activity, TBXT levels, etc.

RESPONSE:

As the reviewer highlights, the biological novelty uncovered using our platform is that inducing nuclear YAP has little bearing on FGF signalling whilst WNT signalling, and subsequently TBXT expression, is down-regulated. As such we agree that the claims should be more closely aligned to the data presented and not rely on the Kastan et al. 2021 findings linking TRULI with LATS1/2. The genetic dissection of YAP pathway kinase components and their multiple inputs, many of which are redundant, disrupt pluripotency (e.g. Aylon et al. 2014) and vary with cell density (e.g. Zhao et al., Genes Dev. 2007), means that genetic experiments in combination with micropattern assays will not be straight forward and will require a full exploration that is beyond the scope of the current study and the conclusions we reach.

As suggested by the referee, we further test the link between TBXT and YAP activity in the control (no drug) assay by performing additional analyses that measure 3D segmented, single-cells YAP and TBXT nuclear levels across multiple time-points, namely when TBXT cells first appear at 12h and 48h. These show, under a null hypothesis that YAP and TBXT are unrelated, that YAP high - TBXT low nuclei are disproportionately present in larger numbers the wild-type neuruloid micropattern at all time-points (vs YAP high + TBXT high). At the earliest time-point (12h) we also found a significant proportion of TBXT high + YAP low (vs YAP high + TBXT high). The analysis of these snapshots of a highly dynamic process points to an avoidance of cell states where nuclear YAP co-exists with TBXT.

To test the relationship between YAP activity and TBXT expression using an alternative approach we have expressed YAP-S5A, a version of YAP that cannot be phosphorylated by LATS1/2. In neuruloid cells that express YAP-S5A, TBXT expression is repressed. This supports that conclusion that YAP activity inhibits the induction of TBXT. We have included these data in new Figure 4h and S13a.

We also would like to emphasise that we observe a negative correlation between YAP and TBXT *in vivo*, in the chicken embryo at early primitive streak stages. We have extended these observations by included additional images including from another stage (HH5, Figure 4j). To our knowledge this is the first such report of the relationship between TBXT expression and YAP in the caudal epiblast *in vivo*.

3. The dissection of trunk progenitor populations and their progeny by comparative single cell RNA-Seq

and spatial mapping (HCR) provides interesting data, but functional experiments are much needed. The authors e.g. claim that they demonstrate a “highly structured architecture in and around the caudal node”. Interesting, but what is the functional relevance of this? Big claims, that need functional evidence. Does perturbation of cytoskeletal components disrupt the architecture? What would be the functional consequence of this?

RESPONSE:

In this section of the paper we describe the architecture around the medial pit/ end node at the 7-10 somite stages during chick development. For this description we use spatial localization of frequently used transcript markers together with novel genes that we discovered based on their predicted expression in the single-cell data. Some of these genes, e.g. NEFM in NMPs, are found across multiple species and we believe this is important to mention. Overall, these gene expression profiles and spatial architecture allowed us to describe a real embryo, and, as pointed out by referee #3, inspired us to formulate signalling hypothesis of the chemical treatments for the in vitro work that comes after in the paper (i.e. timing of TGFbeta inhibitors). Whilst we agree with the referee that finding the function of some of these structural components such as cytoskeletal or adhesion proteins would be interesting, this is not relevant to the work presented in this paper.

We agree with the referee that the architecture of the node and notochord progenitors was inferred indirectly by relying on CNTN2 and NOTO expression levels to locate this population which could be more easily targeted (this is also mentioned in the referee #1 pt. 8). To address this criticism, we have performed further HCRs in the chick embryo to capture early notochord progenitors. These progenitors are SPRY2+ CHR2+ FOXA2+ and WNTs but negative for the nodal inhibitor LEFTY1 and NEFM. Using our 3D quantifications, we pinpoint the spatial location of the SPRY2+ CHR2+ LEFTY1- cells and confirm these are also FOXA2+ and occupy a dorsal position in neur ectoderm. The cross-section of the 3D cell centroids of these quantifications also gives a more detailed view of this region of the embryo. These panels and text have now been added in Figure 2f-i.

4. The authors should include readily available published human scRNA-seq data in their comparative analysis (Tyser et al. Nature 2021, Xu et al. NCB 2023). This is especially relevant since their ultimate aim is to induce node/notochord like cells in 2D and 3D neuruloids.

5. The authors develop a human model guided by the chick data “taking advantage of the evolutionary conservation of trunk cell populations”. However, they do not show that there is indeed conservation with human genetic programs.

RESPONSE:

We agree with the referee that these comparisons would be most useful. However, as we show below, the existing datasets are still very limited compared to the ones of model organisms and over interpreting them could be misleading.

The dataset generated in Xu et al. NCB 2023 spans from CS12-CS16 and is the most promising as caudal populations should be present at the stages assayed. Using the unfiltered/ unprocessed raw data available we obtain only 4 NOTO+ cells out of 103,264. Additionally, there are 54 cells with raw counts >0 for TBXT and 693 for CHR2. The overlap however is very low with only a single-cell in the entire dataset being positive for both TBXT and CHR2. This is peculiar given the other datasets we considered. In addition, when looking at the 693 cells positive for CHR2, these fail to cluster and are spread over multiple populations.

Other combinations such as NOTO+CHRD or T+FOXA2 retrieve no cells. For these reasons, we believe using these data in our analyses would not only be challenging but potentially misleading.

Tyser et al. Nature 2021 describe sequencing of ~2000 cells from a single embryo staged at CS7 (2-3 pcw). The authors identified a robust cluster expressing CHRD and NOTO typical of the Hensen's node (anterior) and the notochord at these stages. As expected from an embryo this early in development (see Fig1a-b), no NMPs are present making it difficult to include in our cross-species comparisons. This is particularly problematic since our main point is the proximity between NMPs and early notochord progenitors (which has been refined as per referee #4 suggestion). Nonetheless, we believe there is value in keeping these earlier, more anterior clusters of cells (no NMP or Neural clusters available) in our comparisons to benchmark how they would perform in a mouse and chick trained classifier alongside macaque (which we previously used as validation) and our in vitro differentiations. These new comparisons are included in Figures S14 and S16.

6. The findings that the size of the neuruloids impacts patterning is interesting, in particular in light of the findings presented here: <https://doi.org/10.1101/2022.12.20.521167>. What is unclear to me is why the authors opted to continue with an intermediate (?) size, whereas the 200µm neuruloids display elongation and form a single pole of TBXT, as is observed in the embryo.

RESPONSE:

We thank the reviewer for the interesting question. It is indeed true that the 200µm patterns display striking fate segregation with a "TBXT+ clump" of cells on one side and a SOX2+ one on the other. However, in our view, it would be inappropriate to consider the TBXT "aggregate" in these structures as equivalent to the TBXT domain in an embryo. In particular, our single-cell analyses showed that at Day 3 these cells have already acquired a caudal identity (CDX2+HOXB9+). In our opinion it would be misleading to select a pattern based on this criterion and directly liken it to the organisation of an embryo. In the current study, we focussed most of our attention on the larger micropatterns, which form 2-3 TBXT+ aggregates, because these sizes proved the most tractable for imaging and analysis purposes. We note that for later experiments, such as the assays of the effect of delayed treatment with TGFb inhibitors, we chose to focus on a single size in order to control for variables such as geometry and mechanics. In the longer term we are interested in understanding the interplay between geometry and size, and aggregate formation and spacing, but this is beyond the scope of the current study. We have now included z-stack videos that provide details of the tissue architecture of these new patterns.

7. The authors claim that their 2D in vitro model gives rise to a subpopulation of node-like cells or notochord progenitors. The data do not provide very solid evidence for this claim. Although indeed the single cell RNA-Seq data show a very small cluster (maybe 10-20 cells out of thousands) with some enrichment of Noto, Chrd, Foxa2, Shh), as far as I can judge by presented data only 1, maybe 2, cells co-express these markers at high levels, as one would expect in node/notochordal-like cells. Moreover, Chrd is expressed in the other clusters as well. The quantifications based on IFC indicate that at least in one of the experiments (#1) there are no TBXT+/FOXA2+ cells detected. It should further be noted that the combined expression of TBXT and FOXA2 does not unequivocally demonstrate a node/notochordal identity, but could also indicate gut progenitors. More substantial evidence regarding the reproducible induction of this population, their localisation in the neuruloids, their functional relevance for the formation of the structures, and unequivocal demonstration of a node/notochord identity, is needed to substantiate the authors' claims.

RESPONSE:

The posterior neuruloid assay, presented in Figure 3 (no inhibition delay), has indeed very few notochord progenitors/ node cells as stated on multiple occasions (“a few TBXT high cells co-stained with FOXA2 indicating the presence of node-like cells or notochord progenitors” or “Notochord TBXT+FOXA2+ progenitors (..) formed a minority (2%...)”).

These cells are indeed a small population, variable from experiment to experiment, as we show and the reviewer points out. However, their presence in the neuruloid assay is consistent with classic embryology studies mentioned in the Introduction. It was these observations that led us to investigate the timing of BMP and NODAL inhibition in Figure 5 and discover that a 24h delay greatly increased this population (“we turned our attention to the small number of notochord-like cells in posterior neuruloids and sought to identify signals responsible for inducing these cells”).

We agree with the reviewer that the identity of these cells could be further strengthened by showing co-expression of markers in this small cluster of cells in the no delay condition. This is now included in Figure S10e (7 single-cells in the cluster mentioned before co-express simultaneously NOTO, CHRDL and FOXA2 which drops to 5 cells if TBXT is also included – remarkable given the typical rates of dropouts in single-cell data). Our main point however, that the delayed inhibition causes a dramatic increase of these cells, is now made clearer in Figure S15c.

8. Given the conceptual importance of the population of TBXT+FOXA2+CDX2+ cells identified by scRNA-seq in macaque and chick for the presented work, the authors should confirm this finding by spatial mapping of this population (and show co-expression of Noto, Chrd, Fgfs, Wnts etc) by HCR (at least in chick).

RESPONSE:

We agree with the reviewer and have now restructured Figure 2 of the manuscript to highlight the relevant populations more directly in the embryo. Please see point 3 above and the new HCR data with CHRDL, SPRY2, FOXA2, LEFTY1.

9. The authors claim that Figure 5b shows “a dramatic increase in endoderm cells, expressing both SOX17 and FOXA2.” However, colocalization is not shown, only separate stainings.

RESPONSE:

We agree with the reviewer that showing co-expression is important. Moreover, there was an inadvertent colour switch in Figure 5e making it potentially confusing. The co-expression IF has now been added in Figure S15a, agreeing with single-cell data, and the colour switch rectified.

10. The induction of a notochord like stripe of cells in the 3D neuruloids is interesting, but needs a much better characterisation and quantification. The authors claim a 75% efficiency, but which criteria is this based on? How reproducible is it between experiments? How variable is this stripe in length, width etc. A detailed morphometric analysis of the structures and the notochord like stripe is absolutely needed.

11. The authors should provide a much better molecular characterisation of the “notochord like stripe” in their 3D structures. Given the amount of scRNA-seq experiments performed in the manuscript, it is surprising that no comparative analysis of the 3D structures with the embryo is provided. In addition, HCR should be performed for notochord markers such as Noto, Chrd, etc to ensure a notochord identity. The

current combination of markers does not exclude the possibility that these represent gut-like midline progenitors secreting Shh, which would also "pattern" adjacent structures. The formation of notochord in 3D embryoids is a very strong claim, and requires equally strong evidence.

RESPONSE:

We agree with the reviewer that the 3D part of the paper was under-explored and warranted further development. As such, we now include a new main Figure 6 dedicated to the translation of the findings on micropatterns to 3D. This figure includes:

- New single-cell RNAseq characterization of the 3D structures, notoroids, at days 3,5 and 7. These data confirmed the presence of notochord and absence of gut progenitors.
- HCR validation of notochord stripe showing expression of CHR1, NOTO as the reviewer suggests.
- We generated a novel GFP line expressing the so-called "T-bound notochord enhancer" from Schifferl et al. (2021) which allowed for better quantifications of the strip-like structure.
- single-cell data on neural ventralization from notochord: cross-section IFs and single-cell RNA-seq.
- Extensive electron microscopy characterization using transverse and longitudinal section as well as correlative microscopy with TBXT-notochord enhancer reporter line so that notochord cells aren't identified only by morphological features.

12. To provide further functional evidence that the cells have a true notochord identity, grafting experiments can be performed to show that the 2D/3D neuruloid cells can induce an ectopic floor plate, as shown in Placzek et al. Science 1990.

13. The notochord is not only an essential signalling centre, but also provides structural support to the embryo. In this regard, notochord cells have specific structural features (reviewed in e.g. Jurand, J Embryol Exp Morphol 1974). The authors should use EM to assess if the in vitro generated node/notochord-like cells also display these typical features (e.g. notochord vacuoles).

RESPONSE:

While we agree that grafting experiments would provide another layer of functional evidence, we do not have the required legal permissions to perform these experiments with embryonic human stem cells. (The MTA for the use of the human ESC line forbids grafting into embryos.) Additionally, we believe this experiment would mainly demonstrate that SHH molecules from notoroids could pattern chicken neural cells, which would speak to our ability to optimize this signal transfer across tissues rather than to evidence of a "true notochord identity".

To assess the notochord identity in these organoids we undertook a detailed ultrastructural study using state-of-the-art electron microscopy. SBF-SEM revealed the expected neuroepithelium surrounding a mesodermal mesenchyme and a, morphologically distinct cluster of cells denoting the notochord. This morphology was confirmed by using CLEM with the reporter fluorescent line. These cells have abundant desmosomes, glycogen deposition and ECM fibres as expected from published literature on human embryos. We present evidence for cilia, vacuoles, and rough ER in these cells in agreement with the cells representing a notochord cell fate.

14. I am surprised about the strong claim that the notochord stripe generated in vitro induces patterning of the neural tube. The data presented by no means back such a bold claim. Based on the data, the 3D neural tissue in the structures appears to be ventralized in the presence of the notochord like stripe. This is a long stretch from obtaining a patterned neural tube, with both dorsal and ventral identities. The current

structures appear to switch from one “in vitro issue”, i.e. dorsalization in the absence of a notochord, to another issue (full ventralisation). The authors should provide a detailed, quantitative analysis how the presence of a notochord-like stripe changes the “patterning” of the neural tissue. Orthogonal sections are needed.

RESPONSE:

We agree with the referee, tissue patterning does not equal a fully dorso-ventral patterned neural tube and we did not mean to suggest that. We make this clearer now and present further evidence for on-going ventral patterning in our 3D structures, including transverse sections suggested by the reviewer. These show, together with new single-cell data, a range of dorso-ventral progenitors from the ventral half of the neural tube generated in the neuroepithelium as a consequence of SHH signalling. Consistent with this, the generation of these cell types is sensitive to vismodegib (a small molecule SHH signalling inhibitor). We show the presence of p2 (PAX6+OLIG2-), motor neuron (OLIG2+) and floor plate (FOXA2+SOX2+) in Figure 6c and h and S20 shows co-expression in single-cell data. A few PAX6+PAX3+ dorsal progenitors are present at day5 but these are mostly absent by day7, probably as a consequence of prolonged exposure to SHH from the notochord population (S20a).

15. In Figure S11e the authors show the presence of somitic tissue (FOXC2+). In what percentage of the structures do the authors obtain both posterior neural and somitic tissue? Do the structures with somitic tissue also develop notochord-like stripes? If so, does the somitic tissue also display “patterning” (or ventralization, see previous comment). Also, given the ability to form both somitic and neural tissue, do the 3D structures harbour NMPs? Do the authors indeed generate more complete structures by inducing a notochord like structure?

RESPONSE:

We thank the referee for suggesting to look more careful at the somitic tissue generated. Single-cell data shows a TBX6+FOXC2+ population at day3 that then down-regulates TBX6 and up-regulates PAX1 and NKX3-1/2 – makers of ventral somites (sclerotome). As we discussed in point 16 below, NMPs are absent at day7. How the timed inhibition of BMP/TGFb signalling affects the proportions and completeness of structures generated is an interesting question, as well as the reliability of these populations. Single-cell provides a clue to the populations present but we additionally wished to estimate the volumetric proportion of notochord (TBXT+ at day 7) and FOXC2+ somitic mesoderm in organoids. To this end we used GastrUnet to unbiasedly segment 14 equally-spaced optical sections of 15-20 organoids and estimate their overall 3D proportions per individual organoid. This was conducted for TGFb inhibitor delays between 12 to 36h (Fig. S19b). With 12-18h TGFb inhibitor delays, the FOXC2 proportions in the 3D structures are estimated to be between 0-5%. Only after 24h delay do they increase substantially to about 12%. This increase continues linearly until the final delay tested of 36h (to ~30%). The notochord population also increases dramatically at 24h to about 3%. Although notochord population also increases with delay, this increase is more modest (up to 6-7%) and does not scale with FOXC2+ mesoderm (Figure 6f). We also used gastrUnet on brightfield images to estimate the maximum elongation length of 3D structures as function of the inhibition delay. The first 3 time-points 12-24h displayed the greatest elongation (Fig. 6f), particularly 18h but at the expense of lower numbers of notochord cells. We therefore focused on the 24h inhibitor delay.

16. In case their final model indeed forms NMPs, posterior neural and somitic tissue, the authors should check if the “highly structured architecture” they claim to observe in vivo is recapitulated in the 3D in vitro system. This would increase the value of the system and, hence, that of the manuscript.

RESPONSE:

We agree with the reviewer that exploring the elongation and progenitor populations in the 3D system over time is of interest. However, by day 7 NMPs were no longer present in the 3D structures and we speculate these are rapidly depleted after the addition of the retinoid acid precursor. Indeed, the early progenitors found at day3 are depleted by day5 in our single-cell data (Figure S17-S18). In this paper we focused on notochord generation and characterization. Understanding the tissue architecture of progenitors over time and their mechanics will be the subject of future studies.

17. The authors state that all scripts are deposited at GitHub. However, for two of the provided links (scRNAseq and gastrunet) the code has not yet been deposited. The authors should do this immediately to ensure pipelines can be reviewed.

RESPONSE:

The code used in the manuscript is available on github:

<https://github.com/tiagu/gastrunet>

<https://github.com/tiagu/Nucleus>

https://github.com/tiagu/trunk_dev_scRNAseq

18. Many of the imaging data presented lack a scale bar (e.g. the entire Figure 4 & almost every supplemental figure). Please add.

RESPONSE:

Thank you - these have now been added to Figures 2, 4 and supplements.

19. The paper suffers from poor statistical description of the imaging data. What is e.g. the penetrance of observed phenotypes upon the described perturbations?

RESPONSE:

Our initial approach of showing the raw, unfiltered images of a typical 96well plate was meant to illustrate that elongation in our cultures was observed in most samples. We have now supplemented this with our reporter cell line showing ZsGreen fluorescence of the T-notochord enhancer reporter for 2 additional 96well plates (Figure S21c). We also use gastrUnet to unbiasedly measure length and a conservative GFP+ area proportion per organoid for over 570 structures. Other morphometrics of these organoid shapes are presented in supplementary (e.g. Figures S19e, S22c) as we feel they do not add much further information.

Minor points:

1. The authors repeatedly refer to the in vitro model as a “3D model of human gastrulation”. Gastruloids model the outcome of gastrulation, but not the gastrulation process per se. This should be clarified. Also, I would not refer to these models as “integrated”, since per ISSCR definition integrated models refer to models comprising both embryonic as well as extra-embryonic tissues.

We agree with the referee. “Integrated” in the abstract was removed to avoid misunderstanding. “human gastrulation” was referenced 3x in the article (1x abstract, 1x main and 1x in discussion). We agree that our model represents aspects of human trunk formation since most cell populations it contains are caudal. We have changed the text accordingly.

2. When discussing single-cell RNA-seq data, the authors repeatedly make claims about cells expressing markers together, without showing data that the markers are actually expressed in the same cell (as opposed to being expressed in different cells of the same cluster).

Due to the noisy nature of single-cell data, a cluster of cells may abundantly express two genes while the overlap in individual cells may be relatively low. Hence, speaking of expression in the cluster portrays the data more accurately. That said, we now include co-expression of several markers for the more relevant cell populations, namely notochord in notoroids (3D) and neuruloids (2D), and for ventral neural progenitor types. See Figures S10e, S15c, S18b and S20a.

3. The authors might want to include a second often used mouse atlas (Pijuan-Sala et al., Nature 2019)

We have considered the dataset suggested by the referee, but we decided not to further complicate the analysis by integrating data of different depths. We considered only the dataset with higher average number of detected genes per cell. This is particularly relevant when looking at lowly expressed genes. See below (not to be included in the paper).

4. “To construct a fate decision axis reflecting both time and space” - what do the authors mean?

We simply meant that pseudo-time sorting in this cell fate decision of neuro/meso captures a temporal component but also a spatial one since this occurs throughout the elongation stages analysed (albeit with a few expression differences, e.g. HOX genes). We have removed this wording to avoid confusing readers.

5. It is unclear to me how the spatial mapping of the clustered HCR data was conducted. The Methods section does not clarify this.

HCR data comes directly from our 3D segmentation tool with marker levels and 3D centroids for each cell. These levels were analysed and aggregated into clusters. Cluster identities were spatially highlighted by colouring 3D centroids of cells analysed back in 3D space. This is made clear in the methods section.

6. The authors should discuss the differences and commonalities of their 2D micropatterned gastruloids

with the 2D gastruloids developed in the work of Brivanlou, Warmflash, Heemskerk, Zandstra etc. These are referenced in the paper.

7. The authors state that the inward movement of cells causes the folding of the apical side towards the centre in the 2D neuruloids. I don't see how the provided data (IFC for some markers) shows this? The statement is backed up by 3 findings: 1) micropatterned colonies start as a flat monolayer of cells tightly packed which by day 2 has an elevation at the edge of the colony (Figures 4a and 4c); 2) By day3 the colony has grown and the mass of cells at the edge is continuous to the mass of cells that folded to the centre (Figure 3a and 3c) with the apical marker ZO-1 appearing in between these cells and the SOX1+ epithelium below; finally, 3) brightfield movies (see Movie 2) shows a clear movement of the outer edge of the doughnut neuruloid shape moving inwards as time progresses. Precise cell movement and tissue morphology changes will be the subject of future studies, but we feel there are strong indications that this is occurring from the images we have of multiple stages of this process.

8. Figure S10a does not show expression of Chrd, as the authors mention. Thank you. This is now corrected.

Referee #2 (Remarks to the Author):

It might be wise for the authors to add a sentence or two to their manuscript noting the ethics and laboratory reviews that they undertook (if any) and perhaps noting recommendation 2.2.1A(d) in the ISSCR Guidelines, if (as I appeared to me) that recommendation explains why specialized ethics review was not sought/was not needed. I suggest there merely to preemptively address questions any reader might have about the ethics of this study, given that the study may be of interest to a variety of audiences.

We agree with the referee and a sentence has now been added highlighting that our work falls within ISSCR 2.2.1A category.

Referee #3 (Remarks to the Author):

Previous attempts to generate in vitro models for the formation of mini-embryoids from human or mouse ESCs or iPSCs have generally led to embryoids containing many of the early tissues (resembling somites, lateral plate mesoderm, definitive endoderm, neural tissue, epidermis, blood etc.) but all existing models failed to generate notochord. This paper addresses this. It starts by analysing the transcriptome of single cells isolated from the caudal half or so (apparently including extraembryonic regions and more) of chick embryos at 4 closely spaced stages. Clustering methods are used to identify different cell types and their putative precursors based on gene expression, and the results then used to develop a protocol to generate axial mesoderm like tissue (notochord) from embryoids made from human ESCs by modifying a previously established method. Apart from critical timed manipulation of Wnt activity via YAP along with modulation of FGF, the protocol includes treatment with retinoic acid precursors, which is shown to lead to elongation of the organoids and importantly, dorsoventral patterning of the somite-like and neural-plate-like structures they contain. This is an important technical advance on the construction of these organoids and may also reveal important aspects about the specification of tissues that were hitherto absent from the embryo models.

In my opinion the main strength of this paper, as compared to many other -oid papers that have been appearing over the last few years, is that it is much more closely informed by direct, primary observations of real transcriptomes from real embryos, including taking into account real developmental time, and also verifying the expression of some of the markers by comparison to the patterns seen in embryos using HCR and immunostaining. Another major strength is that the paper follows a clear logical progression in experimental design. The elongated organoids shown in Fig. 5 g-i are quite impressive relative to what has been published before by people using this model. In principle it should be appropriate for Nature.

I have a few issues that in my opinion need attention especially to constrain some of the generalisations and conclusions to the actual results presented, as in many instances throughout the paper there are many assumptions and speculations that are then used as "facts". I think the paper would be much stronger scientifically if it distinguished predictions and generalisations from actual findings. Some of these issues plague a lot of the current literature but now that the field is quite mature, and given that this group does have a good grounding in developmental biology, it is particularly important to present these things more transparently so that open questions are easier to identify for future study. In particular one problem is the assumption that it is possible to identify "precursors" of anything based on a static gene expression signature of clusters of single cells, based entirely on scRNAseq data and with no attempt to validate the conclusions with lineage tracing or other methods. This should really be avoided - at least specify (in every case) when something is a prediction. This includes NMPs - there is actually no clear evidence for single cells that either do give rise, or can give rise, to neural and mesodermal descendants at late stages - only for a region that contributes to these tissues, and that cells co-express some markers like TBXT and SOX2, but it is perfectly possible that this expression undergoes oscillations before cells settle for one or the other expression and we don't know whether all cells that coexpress these markers do/can give rise to both tissues. The same applies to all other assumptions about "precursors" (and in some cases tissue identity) throughout the paper - simple toning down of the claims will do, but the paper does need to keep open the possibility that these interpretations may turn out to be incorrect in due course.

RESPONSE:

We thank the referee for carefully reading the manuscript and finding it appropriate for publication in the journal. We agree with the referee that we identify snapshots of the cell populations that exist in the trunk and as such we are agnostic as to their future (or past) lineage which can only be addressed by lineage tracing methodologies. For the specific case on NMPs, we rely on the large existing literature which performed several lineage tracing experiments of the node-streak border populations e.g. Cambray and Wilson (2007) or Guillot et al. (2021). These show that cells located at and around the node-streak border give rise to notochord, neural and mesodermal cells. As the reviewer points out, the strength of our paper is that we localize these progenitors in the embryo and hypothesize signalling regimes that we go on to test in our *in vitro* cultures.

Other more specific comments (some very minor, others less), presented more or less in the order in which they appear in the manuscript follow below. The comments are intended to help improve the manuscript and make it more rigorous (and reduce the "spin"):

In the introduction, "a structure known as the organiser or node" is misleading. The node (and/or its precursor cells) is only an organiser (ie can induce and pattern neural tissue and pattern adjacent mesoderm) only for a very short time before axial elongation even begins. The node as a structure persists for much longer and it is responsible for axial elongation, but loses its neural inducing properties very early. In anamniotes the equivalent structure is the embryonic shield (teleost fish) and the blastopore lip (amphibians).

RESPONSE:

We agree with the referee, in fact, this is often a point of discussion. The node/ organizer is a specific spatial and temporal cluster of cells marked by the expression of GSC and OTX2, none of which exists in the posterior structures at the end-point of our analyses. We have changed the text to refer to this as simply the node.

p.3: the tissue isolated for scRNAseq from chick embryos is a huge piece of embryo. The diagrams in Fig 1 show essentially half the embryo. So only a tiny proportion of these cells are in any way relevant to this study. I don't understand why they did not dissect more finely to include only regions including and near the node and its descendant tissues which should be well within the ability of this group to do. This reduces the number of relevant cells sampled and could complicate the interpretation further because many of the "markers" are also expressed in unrelated tissues.

There are many statements like "TBX6 was expressed in the paraxial and presomitic mesodermal cells" and "Notochord cells expressed high TBXT, NOTO, SHH and CHRD" and more seriously (see above) "Crucially, the NMP cells, as in chick, exhibited the highest proportion of double-positive Sox2+T+ cells" and "transcriptome data indicated that these cells corresponded to the early notochord progenitors", etc. which sound as if these cells are identified by other means than the expression described whereas it is actually the other way around. These identities are assigned, or interpreted, based on the expression of the listed genes. This problem occurs throughout the manuscript.

RESPONSE:

We thank the reviewer for allowing us to elaborate further on this part of the manuscript. The chosen dissection area was partly an historical one in the context of this project. However, it solved two, in our view, important issues: 1) The presence of primitive streak, paraxial and lateral mesoderm as well as

preneural and mature neural helped standard single-cell pipelines “align” and place the multiple progenitor cells in the broad neural-mesoderm cell fate space, e.g. pseudo-time sorting without a clear neural pole would make interpretation of cell fate harder. And 2) marker genes for the multiple progenitor populations of the trunk are still not fully known or subject to debate. The relatively large embryo region captured allowed us to be precise as to how broad the expression of a given gene is and to choose the more specific ones.

Our phrasing of cell identity assignment was meant to shorten the initial description of cell populations, but we agree it was too telegraphic. We have now made this clearer in the results section, e.g. “MESP1, SNAI2 and MSGN1 mark primitive streak cells... TBX6 was expressed in the paraxial and presomitic mesodermal cells” now reads “Expression of MESP1, SNAI2 and MSGN1 was used to demarcate primitive streak cells. TBX6 expression outlined the paraxial and presomitic mesodermal cells, the latter with additional MEOX1 co-expression”.

p7: "Combining the single cell transcriptome data with in vivo mapping provides new insight into the organisation of fate transitions driving axis elongation and the spatial dynamics of the signalling pathways controlling them." Here, the reference to in vivo mapping is misleading because it suggests that the precursor nature of the cells has been directly validated, when the reference is merely to in situ staining patterns of the genes.

RESPONSE:

We clarified that this refers to spatially mapping to the embryo.

p.9: the title refers to "posterior cell fate decisions" but the contents of this section are mainly about axial tissues. It is incorrect to consider the node and streak as "posterior fates". Even though they are located increasingly posteriorly during axial elongation, their descendant cells span almost the entire length of the body. So just "axial" should do for this title.

RESPONSE:

We altered the title of this section. See below regarding the posterior/ caudal adjective.

p. 10: "posterior neuruloids". Again these are not necessarily just posterior, but "neuruloids" is quite a good term. In fact the use of "gastruloids" and "gastrulation" in various parts of the paper is not strictly correct - what the study tries to generate is an in vitro model of neurulation, including the formation of axial (notochord) and a dorsoventrally patterned early neural plate which are the hallmarks of neurulation. "Neurulation" as a period begins when gastrulation ends, marked by the appearance of notochordal and neural tissues in front of the node (and the word gastrulation should not be a synonym for cell ingression or mesoderm formation because in many organisms the processes are uncoupled). Also, "mechanisms of trunk formation" seems a bit strong in the absence (at this point in the manuscript) of a description of cell morphologies/arrangements in notochord-like cells, or a clear description of elongation which is crucial for "trunk formation". This comes later in the manuscript and requires the retinoid precursor experiment. I could not find the movies in the submission but the description at this point only refers to the formation of a ring, not elongation.

RESPONSE:

We appreciate that the referee understood the reasoning behind the neuruloid naming! We choose to keep the “posterior” adjective to distinguish from Haremaki, Metzger, Rito et al. (2019) where the neuruloid micropattern models OTX2+PAX6+ anterior neural tissue, neural crest and placode. At

neurulation stages many processes are occurring simultaneously so it is important to highlight we are generating caudal CDX2+HOXB9+ cell populations *in vitro*. That said, we agree with the reviewer that it is not necessary to constantly use the phrasing throughout the manuscript and we have simplified the text. We have changed the ending of the section to be more in line with data presented at this point in the manuscript.

p. 12: " To investigate how YAP activation impairs TBXT induction we used LEF1 expression as readout of WNT signalling". This does not make logical sense. Irrespective of whether it has been previously suggested by others that YAP and Wnt are linked (in fact the references indicate both agonistic and antagonistic effects), the experimental design as stated does not follow the question.

RESPONSE:

We have clarified this sentence and moved the TBXT-WNT connection references closer in the text to aid the readers. Just to clarify, we agree with the reviewer, currently literature points to a complex relationship between YAP and WNT (we cite both agonistic and antagonistic), since TBXT is a known WNT target and is inhibited when we force nuclear YAP, we investigated whether WNT might be impaired by using LEF1 as its proxy, as is common in the literature (Kadigan 2012).

p 15: "To validate the assignment of cell identity, a classifier trained on chick and mouse cell clusters correctly placed the different neuruloid populations in our assigned cell fates" - this seems a circular argument. Using a AI or other predictor to validate another prediction is quite strange.

RESPONSE:

We appreciate the reviewer's comment on this analysis. The rationale for using this type of approach is to find whether a classifier trained on embryo data will correctly classify cell types *in vitro*. We use chicken and mouse embryo cell populations which we typically have assigned manually based on ~2-10 gene markers per cell type, these genes are extensively characterized in the literature (e.g. MESP1, TBX6, NOTO etc..) and we know their location in the embryo. Having these identities allowed us to use an unbiased approach with over an order of magnitude more genes to predict these same cells (0.77 accuracy) which we can use to validate our *in vitro* assignments. In short, yes both embryo and synthetic cultures are assigned but with differences in certainty and the increase in gene numbers make this analysis valuable as another layer of evidence.

p 15: "... prolonged signalling generates endodermal ..." this is very strange because definitive endoderm is believed to be generated only quite early, ending at about the time when notochord starts to form. So what does this prolonged period of signalling correspond to in the embryo?

RESPONSE:

The reviewer is correct in picking this "strange" behaviour from the differentiations. This relates with one of our conclusions that BMP and NODAL inhibition are critical for axial progenitors' generation and maintenance. Indeed, the culture condition that generates endoderm has the most delay in seeing inhibition and has no axial progenitors at day 3. The populations formed with this 48h delay, unlike the condition with inhibition from the beginning of induction ("neuruloid" pattern), have also no caudal character (HOXB9-, see Figure 5c). This most likely corresponds to an earlier, more anterior population which will be worth studying further.

Two previous papers have addressed the compartmentalization of the node in terms of fates at single cell

level: Selleck & Stern 1991 (<https://doi.org/10.1242/dev.112.2.615>) and Solovieva et al. 2022 (<https://doi.org/10.1073/pnas.2108935119>). Some of the statements here are supported by these two papers with direct cell lineage information so it would be useful to cite them. The latter paper also contains scRNAseq data of cells in different subregions of the chick node and at different stages (including CNTN2 expression at later stages) and it would be valuable to compare the chick (and human) transcriptomes generated here with those described in individually isolated cells from different regions of the node.

RESPONSE:

We agree with the referee. We already cited Solovieva et al. (2022) but now added Selleck & Stern (1991). Specifically, we include these references in the context of the new findings of the architecture of the node at 7 somites stage (Figure 2). Additionally, as per the reviewer suggestion, we plotted the expression of long-term resident cells found from HH3+/4 to HH8 and show that, consistent with our findings, cells at the posterior node have higher levels of CNTN2 and SPRY2. This new information was added in Figure S9c.

There are also a few grammatical/typographical errors throughout the manuscript. For example "posterior" is an adjective not a noun in this context (as a noun it means "bum"!) so should always be followed by a noun like "posterior part", etc. (see abstract for example). "Evolutionarily conserved" (not evolutionary conserved - in the Introduction). Media is plural so it should be "inducing medium". p. 6 uses both "median pit" and "medial pit". In the embryological literature this is named "primitive pit" (rightly or wrongly) but just "pit" will do. There is no lateral pit. "caudal node" makes no sense. Just node (as there is no cranial node). "posterior axial fate decisions" is embryologically incorrect - just "axial" - these tissues span most of the length of the embryo. On p 12, the first sentence under the heading makes no sense: "is" seems out of place, and "their" is ambiguous.

RESPONSE:

Thank you - we have corrected these. We agree regarding the naming of the primitive pit but choose to keep "median pit" to link with some of the literature e.g. Charrier et al. (1999).

In my opinion the Discussion is superfluous and could be deleted. Currently it is mainly another summary of the paper and does not add much, as well as reinforcing some of the extrapolations as if they were facts. A single concluding paragraph should be enough.

RESPONSE:

Given reviewer #1 's suggestion to discuss additional literature, we have kept the previous Discussion format but have streamlined the recast of our findings to make it shorter.

Referee #4 (Remarks to the Author):

In this paper, Rito et al investigate trunk tissue formation by using a combination of in vivo and in vitro models. First, they chart trunk formation in chick embryos by combining single-cell transcriptomic with HCR. This allows them to identify the signaling pathways that might lead to the formation of the trunk cell types. Based on these results, they developed a new and improved in vitro model of human trunk formation and characterize it to find the conditions under which specific cell types like notochord cells arise.

I find this study very elegant and interesting. The single-cell RNA-seq (scRNA-seq) experiments and the data analyses are overall well executed, but I have a few concerns and suggestions, as detailed below.

RESPONSE:

We thank the referee for the careful reading of our manuscript and finding it of interest for publication. Below we reply point-by-point to the concerns and suggestions.

1. The authors use the scRNA-seq data to estimate the cell type composition and how it changes, for instance, across developmental stages in chick embryos. However, they should add a caveat in the text warning that scRNA-seq can provide biased estimations of cell type compositions (see, e.g., <https://genomebiology.biomedcentral.com/articles/10.1186/s13059-020-02048-6>), and perform a statistical test to check whether the differences in cell type composition are statistically significant or not (by using methods like <https://www.nature.com/articles/s41467-021-27150-6>).

RESPONSE:

It was never our intention to use scRNA-seq data to estimate compositional changes. In fact, to our knowledge, most serious approaches that do this, such as scCODA the referee suggests, require a reference cell type to anchor the analyses which we would be hard pressed to find this during development or in vitro. For example, scCODA warns: "Thus, an interpretation of scCODA's effects should always be formulated like: "Using cell type xy as a reference, cell types (a, b, c) were found to credibly change in abundance"." Because we do not expect to have such populations, we have shortened this descriptive part of the chick single-cell data in the results so as to not inappropriately emphasize composition differences.

2. After scRNA-seq data analysis, they use HCR to map the spatial patterns of a few selected genes. It is not explained, though, why those genes were selected and not others. Was the choice informed by scRNA-seq data analysis? If so, it would be useful to explain the criteria that were used to choose the genes.

RESPONSE:

The choice was informed by both single-cell data and the existing literature. There are many available images and data in the literature for notochord markers such as CHR1, NOTO or LEFTY but not so much for other markers particularly at high resolution such as SPRY2, a readout of FGF which was important to relate to the on-going signalling in these progenitors (see also points 3 and 8 of reviewer 1). For other cell populations such as pre-neural, the gene marker CNTN2 was chosen solely based on the novel single-cell data. Here our main criteria were both localized and robust expression in the population clusters. But given the number of enriched genes found (e.g. Table S1) we made an arbitrary choice from those.

3. It seems like some cell sub-populations (e.g., the two sub-populations of NOTO+CHR1+ notochord cells described on page 4), were just identified "by eye" from 2D embeddings of the data. This is quite unreliable, as 2D embeddings can be affected by many artifacts. Instead, clustering algorithms should be used here.

RESPONSE:

We thank the referee for the opportunity to explain this further as it also pertains to our “fine-grained” Louvain clustering in minor point 1 below.

Scanpy’s Louvain clustering resolution parameter defaults to 1.0. This gives too broad clusters. In our pipeline we increase this parameter progressively in a biologically principled way, i.e. to capture known groups of cells that we clearly had captured and were also tightly clustered in the 2D UMAP. Namely, floor plate (TBXT-SOX2+FOXA2+SHH+) and neural crest (SOX10+) cells were used to set this parameter (resolution=3.5 as stated in Methods). These clusters are numbers 41 and 40 respectively (Figure S1a).

As we mention in the Results section, some clusters were merged for simplicity but also in the cases where we didn’t find meaningful differences to highlight (for the purposes of this project) which anyways wouldn’t be seen with default parameter settings. This was the case for the notochord. Figure S1 shows it is divided in 3 clusters: one which we call early progenitors (#38) and more mature notochord (merged #30 and #35). TIMP3 and SEMA3E shown in Figure 1h are more of a feature of cluster #35 but we chose not to focus on this difference since both #35 and #30 express SHH and considered them “mature notochord”. In short, the two sub-populations of CHR1+NOTO+ are based on Louvain clusters at res=3.5 and not simply by eye. This is now made clearer in the Methods section.

4. Similarly, on page 15, the authors claim that there's a population of cells (TBXT+FOXA2+CDX2+) that "occupied a region of gene expression space between NMPs and mature notochord" and then refer to a 2D embedding of the data in Figure S10a. This claim should be verified by a more rigorous analysis in the high dimensional space of gene expression.

RESPONSE:

We agree with the reviewer that this was one point of our analysis of looking across species and should be further backed-up. We now performed HCRs to look closer at this cell population (new Figure 2). Additionally, as per the reviewer’s suggestion, we performed a comparison of cell-cell distances across clusters that is independent of the 2D embedding (new Figure S4). We took 2000 random pairs of cells from two different clusters and calculated the average Jaccard distance between their binarized transcriptomes (considering only highly variable genes, n=1486-2327). We then report for chicken, macaque and mouse the mean distance (and 95% confidence interval) for all cluster pairs involving the notochord showing that the NMPs and notochord clusters display some of the shortest mean Jaccard distances across species (Figure S4).

5. Some details that are crucial to ensure the reproducibility of the analysis are missing from the Methods section:

RESPONSE:

We agree with the reviewer and have now added the information below to the methods.

a. As a validation of the assignments of cell identity, they ran a random forest classifier trained on mouse and chick data (page 15), as they specify in the legend of Figure S10c-d. However, the values of some parameters are missing (e.g., number of trees?); moreover, did they use only 1:1 homologous genes?

Our RF classifier was run on 5000 trees. Our strategy to identify homologous genes is now explained in the methods section.

b. On page 24: "counts were normalized to a target sum of 10000 excluding highly expressed genes". How many of the highly expressed genes were excluded?

We typically exclude the top 3% in our normalization.

c. About the clustering shown in Figure 2d, the authors say that they used hierarchical clustering with "K=5". Do they mean K-means clustering? what distance did they use?

This is now Figure S9a. We employed an agglomerative hierarchical clustering method (using Ward's Linkage) with Euclidean distance. Given the linkage matrix, flat clusters were determined by making the distance between members of a flat cluster no more than a minimum threshold and ensuring the number of clusters formed stays below a value K. In this case, we chose K to be 5.

6. It is very good that the authors created GitHub repositories to publish the scripts used for the analysis (these might also provide the answers to the above questions regarding the Methods section), but why are they still empty?

The github repositories are now populated:

<https://github.com/tiagu/gastrunet>

<https://github.com/tiagu/Nucleus>

https://github.com/tiagu/trunk_dev_scRNAseq

****Minor points****

- What do the authors mean by "fine-grained" Louvain clustering? it should either be clarified or removed
Please see major point 3 above.

- The sentence on page 4 "SEMA3C/E , a semaphorin implicated in the control of cell morphology and motility" lacks a citation.

This citation has been added.

- Figure S10C has very low resolution and the x-axis labels are cut

Thank you. This figure is now Figure S16 and has been redone to include human embryo data and new single-cell data for the 3D notoroid. We also made sure axes are now visible.

Reviewer Reports on the First Revision:

Referees' comments:

Referee #1 (Remarks to the Author):

Rito, Briscoe et al have extensively revised their work, which overall has substantially improved their manuscript by strengthening the claims. They should in particular be commended for the ultrastructural analysis of the in vitro generated notochord-like structure by EM, which is very well done and, together with the newly added scRNA-seq and stainings, provide unequivocal evidence that they have successfully generated a human embryo model with a notochord-like structure.

While I am very satisfied with this part of the revisions, I am still a bit underwhelmed by how other points of criticism were addressed. I will detail these remaining concerns below.

1. In general, in my opinion the manuscript still suffers from a patchwork structure. While I appreciate that - as Reviewer 3 indicates - the embryo model design is “informed by direct, primary transcriptomes from real embryos”, the current logical flow of the manuscript does not stress this enough. For me, this issue is rooted in a lack of focus - e.g. the whole part about cytoskeletal components is interesting from an atlas / resource point-of-view, but distracts from the main logical flow of the experiments. What is the added value for *this* paper? Similarly, the whole part about YAP-WNT-TBXT is an interesting observation but, as of yet, completely disconnected from the in vitro notochord part. The main problem with this is not only the apparent disconnect between the separate parts of the manuscript, but also that some parts of the manuscript remain very underdeveloped, even upon revisions (see more specific comments below).

2. A second general criticism, is that for many of the revisions, the authors rely on newly acquired scRNA-seq data. Many of the points would have been better addressed by further IFC/HCR and, especially, more detailed and quantitative analysis thereof (see more specific comments below).

3. The new data regarding YAP-WNT-TBXT is only partially addressing my concerns. In general, I do not agree that, as the authors rebuttal, genetic experiments [...] will require a full exploration [...] beyond the scope [...] and the conclusions. Yes, these are not straightforward experiments, but they are necessary to substantiate their conclusions. Moreover, there remain problems with the data provided. In particular:

3A. The YAP-S5A experiment provides important further evidence, but lacks a control OE.

3B. Figure S13b: why such a complicated analysis? As far as I can see, it only shows that at all analysed timepoints, there are more YAP-high/T-low and YAP-low/T-high cells than double-high. What it does not show is that *in individual cells* (nuclear) YAP is anti-correlated with TBXT levels. A straightforward analysis would be to plot all non-negative cells, and calculate Pearson R.

3C. The authors should perform similar quantifications for the in vivo data, and also analyse total YAP (nuclear/cytoplasmic).

3D. Mechanistically, it remains unclear how YAP activation would block TBXT induction by inhibiting WNT signalling, especially since as far as I can see all these experiments are conducted in the presence of CHIR.

4. For the deeper characterisation of the 2D model the authors largely rely on scRNA-seq data. Why do they not perform HCR for true notochord markers instead of relying on immunostaining for TBXT and FOXA2?

5. Proper quantifications, reporting of reproducibility in the patterns, cell type formed etc for the 2D model under different conditions is still lacking in Figure 4 and 5 (as e.g. provided in Figure 3e). This was requested in my previous Major point 7, but not addressed.

6. Overall the deeper characterisation of the notochord-like structure is very well done and extremely valuable. The human TNE line will be of great value for the field, and provides important orthogonal evidence. Thank you for that! Some remaining issues:

6A. The authors state that 75% of the aggregates show a notochord like structure (n=36), but the related figure shows a total n of 96? This is confusing but not my major issue, which instead is that this reporting suggests that this finding was based on a single experiment. Whereas the finding is clearly reproducible given the orthogonal evidence provided, the authors should provide a better quantification of the reproducibility across multiple experiments. The authors also did not address my previous comments, requesting statistics regarding reproducibility between experiments.

6B. Imaging analysis of the notochord-like structure: the authors claim that most notoroids have an uninterrupted signal for 30% of the length, but how can they conclude that the signal is uninterrupted since, as they report, the analysis was conducted on “estimates of 3D structure from bright-field images”? Such a conclusion can only be drawn from confocal imaging that is not subjected to Maximum Intensity Projection. A second issue with the imaging analysis becomes apparent from the Figure S22, where at least in 2 of the 3 structures it is clear that GastruNet does not extract the correct skeleton (given that the notoroid is partially folded on itself along the optical axis).

7. Despite great efforts, the characterisation of putative in vitro notochord induced patterning of neural tube and somites remains poor. The authors added extensive additional scRNA-seq analysis, whereas HCR for known DV markers would have been not only sufficient, but also way more informative. For the requested analysis of DV patterning of the somitic tissue no data beyond scRNA-seq (which does not show patterning) is provided, and for the neural tube the data is very sparse. On top, the data showing that “gene expression of the more ventral cell types was associated with the presence of nearby notochord-like cells” is not quantified across multiple samples. It would be very helpful if the authors

could analyse putative NT patterning in samples with a true notochord-like-structure (structures like in Fig 6c and 6g instead of the current Figure 6h) which only has scattered TNE+ cells, and provide statistics. I consider this critical to address before publication.

Minor comments:

1. Figure 1e, S1f: Based on which literature do the authors use MESP1, SNAI2 and MSGN1 as markers of the primitive streak?
2. Figure 3f: how sure are the authors about the annotations? The cluster annotated as PS is Cyp26a1 positive, suggesting that these are in fact NMPs. And the cluster annotated as NMPs could be neural biased NMPs, given that many of the cells lack TBXT. The authors should provide stronger evidence that these annotations are correct.
3. The inference of the NMP trajectories towards neural vs somitic decision making (Fig 2c and S6a) is very interesting. Are there commonalities and differences with the “neural vs somitic NMPs” trajectories recently published by Bolondi et al., (Dev Cell 2024)?
4. A rationale for adding a retinoid acid precursor in the 3D protocol should be provided.
5. I would highly recommend not to introduce the term “notoroid”. The embryo model literature is already cluttered enough.
6. At several occasions the authors use the term “organoid”. I would replace this with “embryoid” or “embryo model”.
7. “Total organoid area (p=0.32)” should be “R=0.32”.
8. The authors should cite and discuss the Thisse paper demonstrating the formation of a notochord-like structure with high efficiency in a mouse embryo model: DOI: 10.1038/s41467-021-23653-4
9. Scale bar is still missing in some figures, e.g. S17a,d.
10. Thank you for depositing the code. However, a data scientist in my lab tried to run the code and found that “The code in that nucleus repository is not finished at all by the way, I tried running it and it has quite some bugs and the model that he uses is not included anywhere”.
11. At multiple occasions the authors refer to analogous morphogenetic events between the embryo and the model without showing any proof:

“These cells constitute a delaminating population that ingresses [...] reminiscent of the behaviour [...] engrossing at the primitive streak”

“Strikingly, the different populations were organised [...] mechanical or cell adhesion cues driving morphogenetic behaviour.”

Without any data supporting this, these claims should be removed.

Further notes:

1. I agree with the authors' reasoning in the rebuttal regarding the transplantation experiment I suggested (apart from the ethical problems they rightfully note). I consider this fully resolved.
2. The authors provide good rationale for not being able to (fully) use the Tyser et al. & Xu et al. human data-sets as I previously requested. I consider this fully resolved.
3. The authors provide a good rationale in the rebuttal for their selection of a particular micropattern size. I consider this fully resolved.
4. The authors provide a good rationale in the rebuttal for not using the Pijuan-Sala scRNA-seq dataset. I consider this fully resolved.

Referee #2 (Remarks to the Author):

Thank you for addressing my suggestion to add clarity about ethics review and the status of the study within the most recent ISSCR guidelines.

Referee #3 (Remarks to the Author):

-

Referee #4 (Remarks to the Author):

The authors have addressed all the points I've raised in a satisfactory manner, making their analyses clearer and more robust.

Author Rebuttals to First Revision:

Referees' comments:

Referee #1 (Remarks to the Author):

Rito, Briscoe et al have extensively revised their work, which overall has substantially improved their manuscript by strengthening the claims. They should in particular be commended for the ultrastructural analysis of the in vitro generated notochord-like structure by EM, which is very well done and, together with the newly added scRNA-seq and stainings, provide unequivocal evidence that they have successfully generated a human embryo model with a notochord-like structure.

We thank the reviewer for carefully reading the paper and finding the manuscript much improved with new EM, scRNA-seq and stainings.

While I am very satisfied with this part of the revisions, I am still a bit underwhelmed by how other points of criticism were addressed. I will detail these remaining concerns below.

1. In general, in my opinion the manuscript still suffers from a patchwork structure. While I appreciate that - as Reviewer 3 indicates - the embryo model design is “informed by direct, primary transcriptomes from real embryos”, the current logical flow of the manuscript does not stress this enough. For me, this issue is rooted in a lack of focus - e.g. the whole part about cytoskeletal components is interesting from an atlas / resource point-of-view, but distracts from the main logical flow of the experiments. What is the added value for *this* paper? Similarly, the whole part about YAP-WNT-TBXT is an interesting observation but, as of yet, completely disconnected from the in vitro notochord part. The main problem with this is not only the apparent disconnect between the separate parts of the manuscript, but also that some parts of the manuscript remain very underdeveloped, even upon revisions (see more specific comments below).

The flow of the article, which was appreciated by the other referees, was to gain information from chicken embryos at elongation stages, for which genomic data is still scarce (e.g. compared to mouse), but functional studies and precise staging are available. Since much is already known in embryos, we highlight from this new atlas novel cytoskeletal components marking the relevant cell populations.

The *in vitro* platforms we propose are based on these *in vivo* observations, as indicated by Reviewer 3. Before the main focus of the article which indeed is the notochord, we believe it is important to show how the 2D micropatterned colonies are established, in particular how in the widely-used context of WNT and FGF signalling, BMP and NODAL inhibition leads to symmetry breaking and TBXT induction. Despite the interesting

observations of WNT and FGF in axial elongation in recent work (e.g. ref.12) much is already known (see Isabel Olivera-Martinez and Kate G. Storey Development 2017). In our view the YAP-WNT-TBXT connection is a novel and important discovery worth highlighting. These findings are confined to Figure 4, panels e-j and therefore we are careful not to overstate them in the text. We believe readers, including the reviewer, given his/her requests below, will find the results of interest.

2. A second general criticism, is that for many of the revisions, the authors rely on newly acquired scRNA-seq data. Many of the points would have been better addressed by further IFC/HCR and, especially, more detailed and quantitative analysis thereof (see more specific comments below).

We disagree. As previously mentioned by the reviewer, there is a range of orthogonal data to support our conclusions, including EM (scanning and transmission), new immunofluorescence antibody labelling and HCR, new *in vivo* chicken analyses of YAP and TBXT at a different stage, a new enhancer cell line and quantifications, cryo-sections asked by the reviewer and movies of the 3D structure.

3. The new data regarding YAP-WNT-TBXT is only partially addressing my concerns. In general, I do not agree that, as the authors rebuttal, genetic experiments [...] will require a full exploration [...] beyond the scope [...] and the conclusions. Yes, these are not straightforward experiments, but they are necessary to substantiate their conclusions. Moreover, there remain problems with the data provided. In particular:

3A. The YAP-S5A experiment provides important further evidence, but lacks a control OE.

We agree with the reviewer that a control experiment is needed however, as pointed out, these are not straightforward experiments. Although the YAP-S5A construct cannot be phosphorylated and hence will never reach the nucleus, the wild-type OE control will be able to translocate depending on LATS1/2 and up-stream kinases' activity and hence many scenarios are possible. That said, we performed the OE wtYAP control the reviewer requested (see new panel Fig.S13a). We examined the periphery (30% radius) of 3D segmented colonies where TBXT induction is concentrated. The wtYAP construct showed a small reduction in the number of TBXT expressing cells but non-significant (p-value 0.078 and a Cliff's delta close to 0.5). This compared to the striking reduction in TBXT+ observed previously in cells with the YAP-S5A construct (p-value=0.00011, Cliff's delta=0.98). This is best appreciated by comparing the observed TBXT+ in GFP cells with the expected value obtained for all cells at the periphery, which in the null scenario we expect to be similar. These observed over expected ratios are much closer to 1 in the

wtYAP than in the S5A case (p-value=0.00027, Cliff's delta= -0.975) supporting our hypothesis that active YAP is detrimental to differentiation and TBXT induction. Note that unlike YAP-S5A, where residues are mutated, in the wild-type YAP case, we do detect unphosphorylated active YAP with our active YAP antibody. The over-expression of both is detected using the total YAP antibody.

3B. Figure S13b: why such a complicated analysis? As far as I can see, it only shows that at all analysed timepoints, there are more YAP-high/T-low and YAP-low/T-high cells than double-high. What it does not show is that *in individual cells* (nuclear) YAP is anti-correlated with TBXT levels. A straightforward analysis would be to plot all non-negative cells, and calculate Pearson R.

This is incorrect, we show YAP-low/T-high are indistinguishable from double-high at 24 and 48h. Examining the levels of YAP and TBXT in double-positive single-cells does not reveal any meaningful correlation and indeed we would not necessarily expect that. Although we agree that a simpler analysis would be preferable, observing negative correlation in single cells implies assumptions about the mechanism that do not appear to hold, such as the dynamics of TBXT induction or the persistence of nuclear YAP. This highly dynamic process will have to be investigated further using synthetic reporters and live-imaging experiments, which is well beyond the scope of this project. Our data on fixed samples points to nuclear YAP (either by LATS1/2 chemical inhibition or over-expression of always active S5A form vs wild-type YAP) impeding timely TBXT induction.

3C. The authors should perform similar quantifications for the in vivo data, and also analyse total YAP (nuclear/cytoplasmic).

We agree with the reviewer that this would be a nice, although redundant, additional experiment and analysis. However, the mouse total YAP antibody we used in differentiating human colonies does not work well in chicken embryos yielding a high non-specific signal that we do not trust. Moreover, we have the quantification for the active, phosphoYAP antibody (Fig. S13c, bottom right) showing that TBXT+ cells are negative/ very low for active YAP.

3D. Mechanistically, it remains unclear how YAP activation would block TBXT induction by inhibiting WNT signalling, especially since as far as I can see all these experiments are conducted in the presence of CHIR.

We have revised the text to be more precise: we do not state that TBXT is permanently blocked by this inhibition but rather that is much reduced/absent at 24h. At day3 we detect TBXT+ cells (Fig.S12) but not more mature differentiation markers (e.g. TBX6/ FOXA2) so likely nuclear YAP results in a delay in activation that is eventually overcome

by persistent CHIR. This is not surprising given the existing indirect literature showing that TBXT expression is increased in YAP1 knockout human gastruloids induced by BMP4 stimulation (ref.80) and MST1/2 KO mouse ESC show impaired mesoderm formation and overall resistance to differentiation (refs.81,82).

4. For the deeper characterisation of the 2D model the authors largely rely on scRNA-seq data. Why do they not perform HCR for true notochord markers instead of relying on immunostaining for TBXT and FOXA2?

Our approach combines single-cell RNA-seq and immunostaining to characterize the 2D micropattern results. One major strength of our micropattern approach is that it yields stereotypical colonies that are highly reproducible and contain a small number of cell types which can be robustly captured in the single-cell data and identified by validated orthogonal markers (by literature and the scRNA-seq data) using immunofluorescence. In our view, HCR would not add to these analyses because we have good, available antibody combinations to uniquely identify all cell populations present at the protein level and unequivocally link them with their transcriptome.

5. Proper quantifications, reporting of reproducibility in the patterns, cell type formed etc for the 2D model under different conditions is still lacking in Figure 4 and 5 (as e.g. provided in Figure 3e). This was requested in my previous Major point 7, but not addressed.

The previous major point 7 requested further details on the small subpopulation of notochord cells found in the pattern with constant inhibition (Figure 3), the reviewer questioned: 1) the reproducibility of the induction of this population, 2) whether these could represent gut populations, and 3) whether this was indeed a node/notochord identity.

1) As previously discussed, during constant inhibition (neuruloid platform in Figure 3) this is a rare cell type (<1%) – we both expect and emphasize this in the text, despite being able to capture it and quantify with single-cell sequencing. These low numbers were our motivation for using whole-colony 3D segmentation of high-resolution images across experiments (Figure 3e).

2) We demonstrate this population identified by FOXA2+TBXT+ in IF also co-expressed NOTO together with CHRD in single-cell data (Figure S15c). This combination of markers is characteristic of notochord. In addition this population of cells is greatly expanded (to ~28% in single cell data) in the condition where BMP/Nodal inhibitor addition is delayed 24h delay inhibition. It is absent if addition of BMP/Nodal inhibits is delayed 48h (Figure S15c). The co-expression in individual cells of FOXA2+TBXT+NOTO+CHRD+ shows this is not a gut population. As requested by the reviewer, we have now added quantifications

of high-resolution colonies of these 24h delay colonies (Figure S15d). As expected from our previous data, TBXT+FOXA2+ increases significantly (to 44±8% out of 4991±907 segmented 3D nuclei per colony) in this assay and is robust across multiple experiments (Fig. S15e) – indeed this is the delay we employ in the 3D colonies.

3) This same FOXA2+TBXT+NOTO+CHRD+ cell population is captured in 3D and we use this 3D *in vitro* model to further characterize this *in vitro* generated notochord population by EM (Figure 6), which the referee finds convincing.

The micropattern experiments produce not just stereotypical colonies (e.g. see pERK1/2 at 12h in Figure S11 or Figure 5e) but also consistent results across experiments as illustrated by the multiple use of the platform throughout the paper.

To further accommodate the reviewer's concerns, we additionally show coverslip overviews for FOXA2+TBXT+ in the 24h delay condition across experiments in Figure S15e. We also report that several collaborating labs at UCL and University of Warwick are now successfully using our methods with identical results.

6. Overall the deeper characterisation of the notochord-like structure is very well done and extremely valuable. The human TNE line will be of great value for the field, and provides important orthogonal evidence. Thank you for that! Some remaining issues:

6A. The authors state that 75% of the aggregates show a notochord like structure (n=36), but the related figure shows a total n of 96? This is confusing but not my major issue, which instead is that this reporting suggests that this finding was based on a single experiment. Whereas the finding is clearly reproducible given the orthogonal evidence provided, the authors should provide a better quantification of the reproducibility across multiple experiments. The authors also did not address my previous comments, requesting statistics regarding reproducibility between experiments.

We are sorry this wasn't clear to the reviewer. Figure S17a showed a typical 96-well plate together with examples on the right of models positive and negative for what we considered a TBXT+ streak. We have clarified this now with new, separate panels for different cell lines. Additionally, we considered the reviewer's request for further details on reproducibility and added more information on separate experiments (Figure S17b) as well as these statistics for a different cell line (Figure S17c). Furthermore, we provide more examples of structures counted as streak or negative/poles of TBXT. Overall, the efficiency of robust TBXT streaks remains at around 70-75% depending on cell line, MShf4 or H9, respectively. We highlight that the remaining structures all had TBXT+ patches (just not as organized) as highlighted by our quantification of GFP area in over 576 preparations showing a non-zero bright GFP area (Figure 6k right).

6B. Imaging analysis of the notochord-like structure: the authors claim that most notoroids have an uninterrupted signal for 30% of the length, but how can they conclude that the signal is uninterrupted since, as they report, the analysis was conducted on “estimates of 3D structure from bright-field images”? Such a conclusion can only be drawn from confocal imaging that is not subjected to Maximum Intensity Projection. A second issue with the imaging analysis becomes apparent from the Figure S22, where at least in 2 of the 3 structures it is clear that GastruNet does not extract the correct skeleton (given that the notoroid is partially folded on itself along the optical axis).

We acknowledge, as we point out in the paper, the measurements taken from wide-field microscopy (not a confocal M.I.P.) are estimates from the real 3D structure. Figure S22a not only highlights this but clearly shows the estimates will always be an underestimation of the real length and GFP signal in the structures – we consider this an acceptable compromise as it allows the unbiased analyses of hundreds of structures (e.g. Figure 6k, n=576). Furthermore, it is also apparent from Figure S22a that the GFP+ signal considered (dark green outline) is an underestimation of the real signal, maybe necessarily so, as a fold along the optical axis will always result in a shorter measurement and a reduced GFP signal (which will interrupt the stretch as we quantify only the brightest signals).

We have now given additional evidence for our claims and the robustness of the signal by quantifying across cell lines and independent experimental replicates the TBXT streak inside the notoroids. In Figure S17b-c, we show the interior of notoroids where 70-75% of structures display a long TBXT streak, often spanning the entire structure (n_cell lines =2, n_experiments=5, n=89). Furthermore, to confirm our estimates aren't too far off from the actual 3D structures, we analysed the GFP and TBXT signal using lightsheet microscopy (n=4, see new Movie7 for an example). These volumetric measures largely corroborate our findings from high-throughput wide-field perspectives and point to structures slightly longer, as expected, with average length of 2.33 ± 0.13 mm. An independent replicate of Fig.6j showed an uninterrupted signal for 43% of the trunk models' lengths (Fig.S21c), yielding a $37.1 \pm 19.9\%$ overall value (n=192). In 3D, we found that the uninterrupted signal measure is consistent with what we previously reported: $37 \pm 5\%$ of their total 3D path length (Fig. S21c, table and Movie 7). This new information has been added to the manuscript.

7. Despite great efforts, the characterisation of putative in vitro notochord induced patterning of neural tube and somites remains poor. The authors added extensive additional scRNA-seq analysis, whereas HCR for known DV markers would have been not only sufficient, but also way more informative. For the requested analysis of DV patterning of the somitic tissue no data beyond scRNA-seq (which does not show

patterning) is provided, and for the neural tube the data is very sparse. On top, the data showing that “gene expression of the more ventral cell types was associated with the presence of nearby notochord-like cells” is not quantified across multiple samples. It would be very helpful if the authors could analyse putative NT patterning in samples with a true notochord-like-structure (structures like in Fig 6c and 6g instead of the current Figure 6h) which only has scattered TNE+ cells, and provide statistics. I consider this critical to address before publication.

We agree that an in-depth analysis of notochord patterning activity in our trunk embryo models is very interesting. However, this represents a major new project in itself. To begin this process, we provide the critical data and statistics requested by the reviewer to show that indeed there is a consistent relationship between NT patterning and notochord cells. This uses the transverse sections previously requested by the reviewer as. Sections allow immunostaining and measurement of the outer neural epithelium without illumination artifacts from lightsheet/ confocal microscopy. These new data document the physical distances of the 30 closest floor plate FOXA2+ cells and more dorsal neural PAX6+(SOX2+) cells for every notochord cell TBXT+(SOX2-) from 9 trunk structures. Altogether, we analysed over 663 individual notochord cells and 39,780 pairwise distances showing that floor plate cells are considerably closer to notochord cells than Pax6 expressing cells - mean distance of 66um vs 164um (pval=0.0 Wilcoxon test, Cliff's delta=-0.8). We show these new data and analysis in Fig. S20f.

Minor comments:

1. Figure 1e, S1f: Based on which literature do the authors use MESP1, SNAI2 and MSGN1 as markers of the primitive streak?

All three genes highlighted are co-expressed together in single cells close to the NMP cluster (Figs.S1e and S2a) and in an early population of cells that we believe are fated to mesoderm. Whereas MSGN1 appears less specific to this cluster (see Fig.1 I and Guillot et al. 2021), in chick, MESP1 has been associated with a primitive streak location (see GEISHA localization for the gene and Buchberger et al. 1998). Also in mouse *Mesp1*+ cells were found to accumulate in the primitive streak (Saga et al. 1999). We modified the text here to account that they reflect early nascent mesoderm.

2. Figure 3f: how sure are the authors about the annotations? The cluster annotated as PS is *Cyp26a1* positive, suggesting that these are in fact NMPs. And the cluster annotated as NMPs could be neural biased NMPs, given that many of the cells lack TBXT. The authors should provide stronger evidence that these annotations are correct.

We thank the reviewer for this question. We do see the two populations are very close in transcriptional state and probably reflect early biases towards the different trunk lineages. A few points to clarify our decision:

- *Cyp26a1* is not NMP-specific in chicken and is a broader marker towards the nascent mesoderm population. This appears to also be the case in vivo in the mouse data we analysed (Fig.S6a).
- *MIXL1* in Fig.S10d has a similar expression pattern. So does *Wnt3a*, a marker of PS (Takada et al. 1993).

3. The inference of the NMP trajectories towards neural vs somitic decision making (Fig 2c and S6a) is very interesting. Are there commonalities and differences with the “neural vs somitic NMPs” trajectories recently published by Bolondi et al., (Dev Cell 2024)?

Both our paper and Bolondi et al. analyses order the transcriptomes of single cells by their similarity in a Mesoderm-Neural axis. Bolondi et al. (2024) focus on the mouse TLS model system. While they detect (in their Fig.3E) neural- and meso-biased NMP populations that correspond to genes we also use as markers such as *Tbx6* (their module 14) and *Sox1* (module 7). It is less clear that an intermediate population is captured as their module 4 also appears to be neural-biased. We have not pursued this further, at this stage, as we do not think it will substantially increase our understanding.

4. A rationale for adding a retinoid acid precursor in the 3D protocol should be provided.

We now added a rationale for this treatment that comes from the opposing RA gradient occurring rostrally in the chicken embryo, see Diez del Corral et al. (2003) where they show expression of several RARs rostrally, after the first somite. The precursor is added to a RA-depleted media so that RA can be produced.

5. I would highly recommend not to introduce the term “notoroid”. The embryo model literature is already cluttered enough.

6. At several occasions the authors use the term “organoid”. I would replace this with “embryoid” or “embryo model”.

While we agree with the reviewer and refrain from using the word “notoroid” often or in the abstract, we also acknowledge that it is useful for people in the field to have a specific name to refer to the work such as “gastruloid” or “trunk-like structures”. As such we keep it in a few instances to make that distinction.

We modified the text to use the word trunk-organoid, which we think more accurately reflects the preparation we describe. We wish to avoid the use of “embryo model”

because this is being used by people constructing “synthetic embryos” and our preparations do not have any rostral cell populations.

7. “Total organoid area ($p=0.32$)” should be “ $R=0.32$ ”.

This was now changed. Thank you.

8. The authors should cite and discuss the Thisse paper demonstrating the formation of a notochord-like structure with high efficiency in a mouse embryo model: DOI: 10.1038/s41467-021-23653-4

Thank you. This is now cited in discussion.

9. Scale bar is still missing in some figures, e.g. S17a,d.

This was added. Thank you.

10. Thank you for depositing the code. However, a data scientist in my lab tried to run the code and found that “The code in that nucleus repository is not finished at all by the way, I tried running it and it has quite some bugs and the model that he uses is not included anywhere”.

The training data used and neural network weights (Pytorch models) have been deposited online, see link added on github:

https://zenodo.org/records/11388472/files/Nucleus_models.gz?download=1 . We also worked on a more detailed set of instructions so that the tool can be used more widely. It now successfully installs and runs by non-experts in our institute.

11. At multiple occasions the authors refer to analogous morphogenetic events between the embryo and the model without showing any proof:

“These cells constitute a delaminating population that ingresses [...] reminiscent of the behaviour [...] engrossing at the primitive streak”

“Strikingly, the different populations were organised [...] mechanical or cell adhesion cues driving morphogenetic behaviour.”

Without any data supporting this, these claims should be removed.

We modified the text to better reflect the evidence given.

“These cells constitute a **SNAIL+ cell population** found underneath the initially flat epithelial layer...” (Fig.4C)

and

“Strikingly, the different populations were organized in a stereotypical fashion indicating not only on-going cell fate specification but also **morphogenetic behaviours resulting in distinct morphologies (Figure 5e).**”

Further notes:

1. I agree with the authors' reasoning in the rebuttal regarding the transplantation experiment I suggested (apart from the ethical problems they rightfully note). I consider this fully resolved.
2. The authors provide good rationale for not being able to (fully) use the Tyser et al. & Xu et al. human data-sets as I previously requested. I consider this fully resolved.
3. The authors provide a good rationale in the rebuttal for their selection of a particular micropattern size. I consider this fully resolved.
4. The authors provide a good rationale in the rebuttal for not using the Pijuan-Sala scRNA-seq dataset. I consider this fully resolved.

Referee #2 (Remarks to the Author):

Thank you for addressing my suggestion to add clarity about ethics review and the status of the study within the most recent ISSCR guidelines.

Referee #3 (Remarks to the Author):

-

Referee #4 (Remarks to the Author):

The authors have addressed all the points I've raised in a satisfactory manner, making their analyses clearer and more robust.

Reviewer Reports on the Second Revision:

Referees' comments:

Referee #1:

First of all, I would like to thank the authors for the additional experiments and quantifications conducted. These strengthen the conclusions of the manuscript, and provide support for some conclusions previously drawn based on qualitative comparisons. I also thank the authors for the detailed additional clarifications that helped me to understand why some analyses were done in a particular way, or could not be performed for technical reasons. The textual revisions to clarify some points are helpful and important.

While these additional efforts do not change my opinion about the “patchwork” nature of the manuscript, this should not be a reason not to publish the manuscript in its current form, especially since I acknowledge that at least one other reviewer is of the opposite opinion, and very much appreciates the flow of the manuscript.

I still do have concerns about the ability of the imaging and imaging analysis pipeline to extract a true measure of uninterrupted signal (point 6), but while this might affect the exact numbers, it is unlikely to impact the conclusion that, indeed and remarkably, the authors succeeded in generating structures with a notochord-like-structure.

I very much appreciate the new analysis regarding the patterning activity of the in vitro notochord (response to point 7). However, this analysis (computing pairwise distance between notochord and floor plate cells) does not show the reproducibility of patterning in the nine analyzed structures. While I agree that a detailed analysis of notochord patterning activity could be a new project in itself, in order for the field to understand the utility of the model (which questions it can, and which questions it cannot, be used for), I think it is critical to demonstrate the reproducibility of in vitro (dorsoventral) NT patterning (or, if it's not reproducible, provide a measure of the variability). Given that the authors already analyzed nine structures this should not be an issue.

I do not agree about the rationale for not using “embryo model of the notochord” as a terminology, and do not think it is in the interest of the field to add yet another term to the already cluttered terminology, but in the end, this is not my call to make.

To conclude, although I still believe that some of my points raised are valid and could have been addressed with further experiments, I also acknowledge the extensive and thorough revisions and clarifications conducted by the authors. Hence, I think it is about time to approve this important work for publication in Nature, provided that an analysis of the reproducibility of patterning is added.

Author Rebuttals to Second Revision:

Referees' comments:

Referee #1:

First of all, I would like to thank the authors for the additional experiments and quantifications conducted. These strengthen the conclusions of the manuscript, and provide support for some conclusions previously drawn based on qualitative comparisons. I also thank the authors for the detailed additional clarifications that helped me to understand why some analyses were done in a particular way, or could not be performed for technical reasons. The textual revisions to clarify some points are helpful and important.

While these additional efforts do not change my opinion about the “patchwork” nature of the manuscript, this should not be a reason not to publish the manuscript in its current form, especially since I acknowledge that at least one other reviewer is of the opposite opinion, and very much appreciates the flow of the manuscript.

I still do have concerns about the ability of the imaging and imaging analysis pipeline to extract a true measure of uninterrupted signal (point 6), but while this might affect the exact numbers, it is unlikely to impact the conclusion that, indeed and remarkably, the authors succeeded in generating structures with a notochord-like-structure.

I very much appreciate the new analysis regarding the patterning activity of the in vitro notochord (response to point 7). However, this analysis (computing pairwise distance between notochord and floor plate cells) does not show the reproducibility of patterning in the nine analyzed structures. While I agree that a detailed analysis of notochord patterning activity could be a new project in itself, in order for the field to understand the utility of the model (which questions it can, and which questions it cannot, be used for), I think it is critical to demonstrate the reproducibility of in vitro (dorsoventral) NT patterning (or, if it's not reproducible, provide a measure of the variability). Given that the authors already analyzed nine structures this should not be an issue.

I do not agree about the rationale for not using “embryo model of the notochord” as a terminology, and do not think it is in the interest of the field to add yet another term to the already cluttered terminology, but in the end, this is not my call to make.

To conclude, although I still believe that some of my points raised are valid and could have been addressed with further experiments, I also acknowledge the extensive and thorough revisions and clarifications conducted by the authors. Hence, I think it is about time to approve this important work for publication in Nature, provided that an analysis of the reproducibility of patterning is added.

We thank the reviewer for carefully re-reading our manuscript and for finding it worthy of publication. We greatly value the time and effort spent to critique the work and believe the paper has been strengthened by it.

Our patterning analyses show that in different organoids we consistently see floorplate cells arising closer to the notochord than Pax6 expressing neural progenitors. The reviewer requests information on reproducibility of this observation between individual organoids. In fact, the thin lines in the plot of Fig.20g represented the distances found for each individual organoid, whereas the more prominent thick line is the averaged data for all organoids. We now make this clearer in the figure legend. We have added an extra panel in Fig. S20f (now new Extended Data Figure 9c) where we show for 3 independent experiments, sections of two different organoids stained for PAX6 (dorsal neural tube), OLIG2 (ventral neural tube) and TBXT (notochord). This shows that in all the individual organoids similar patterning occurs and allows the reader to better judge the patterning present in these structures.